# MAPSS: Manifold-based Assessment of Perceptual Source Separation

**Amir Ivry[1]   Samuele Cornell[2]   Shinji Watanabe[2]**
[1]Electrical and Computer Engineering, Technion - Israel Institute of Technology, Haifa, Israel
[2]Language Technologies Institute, Carnegie Mellon University, Pittsburgh, PA, USA
`aivry@ieee.org   samuele.cornell@ieee.org   swatanab@andrew.cmu.edu`

## Abstract

Objective assessment of audio source-separation systems still mismatches subjective human perception, especially when interference from competing talkers and distortion of the target signal interact. We introduce Perceptual Separation (PS) and Perceptual Match (PM), a complementary pair of measures that, by design, isolate these leakage and distortion factors. Our intrusive approach generates a set of fundamental distortions, e.g., clipping, notch filter, and pitch shift from each reference waveform signal in the mixture. Distortions, references, and system outputs from all sources are independently encoded by a pre-trained self-supervised model, then aggregated and embedded with a manifold learning technique called diffusion maps, which aligns Euclidean distances on the manifold with dissimilarities of the encoded waveform representations. On this manifold, PM captures the self-distortion of a source by measuring distances from its output to its reference and associated distortions, while PS captures leakage by also accounting for distances from the output to non-attributed references and distortions. Both measures are differentiable and operate at a resolution as high as 75 frames per second, allowing granular optimization and analysis. We further derive, for both measures, frame-level deterministic error radius and non-asymptotic, high-probability confidence intervals. Experiments on English, Spanish, and music mixtures show that, against 18 widely used measures, the PS and PM are almost always placed first or second in linear and rank correlations with subjective human mean-opinion scores[1].

## 1  Introduction

Reliable perceptual evaluation is critical for source-separation progress, yet gold-standard listening tests are costly and slow (ITU-T., 1996; 2003; 2018). Thus, research relies on objective metrics that blur two distinct failures, interference from competing talkers and target distortion. Disentangling these modes can better align with listener perception and accelerate trustworthy development.

Existing measures such as the signal-to-distortion ratio (SDR), signal-to-interference ratio (SIR), signal-to-artifacts ratio (SAR) (Vincent et al., 2006), scale-invariant SDR (SI-SDR) (Le Roux et al., 2019) and alike usually compute ratios between source to various disturbances in the waveform domain, offering low complexity and widespread adoption. However, even jointly, they mix leakage and self-distortion into global energy ratios and promote ambiguity to whether an error stems for leakage or self-distortion. Classical intrusive perceptual and intelligibility metrics like the PESQ (Rix et al., 2001), STOI (Taal et al., 2011) and ESTOI (Jensen & Taal, 2016) map an entire utterance to a scaled mean-opinion score (MOS) using hand-crafted auditory features. Designed preliminary for speech enhancement, they perform well for corrupted noisy-reverberant speech utterances but may not account for leakage, while also lacking to provide access to their inherent granular processing, i.e., at the frame level. Learned black-box metrics such as the DNSMOS family (Reddy et al., 2022) that are trained end-to-end to predict crowd-sourced MOS, as well as SpeechBERTscore (Saeki et al., 2024) and Sheet-SSQA (Huang et al., 2025), have shown promising results on various speech tasks, but do not offer confidence in their decisions. Spectral-distance metrics are interpretable but tend to mask where degradations occur, e.g., the popular Mel-Cepstral Distortion (MCD) (Fukada et al.,

---

[1]Code available at `https://github.com/Amir-Ivry/MAPSS-measures`

1992) collapses the spectral envelope into a global value. Another set of metrics that gained popularity in recent years include the non-intrusive VQScore built from variational auto encoder (Fu et al., 2024), the full-reference ViSQOL that uses gammatone "neurograms" (Chinen et al., 2020), NORESQA that learns a relative quality function between two non-matching recordings (Manocha et al., 2021), and SCOREQ that is trained to estimate utterance-level MOS on telephone and synthetic-speech degradations (Ragano et al., 2024). Even when taking into account a broader set of metrics, as available in recently developed speech quality assessment toolkits (Shi et al., 2025), no existing family of measures can simultaneously disentangle leakage from distortion, offer granular analysis, and provide error estimates for their decisions.

We introduce the Perceptual Separation (PS) and Perceptual Match (PM), the first measures for source separation that functionally disentangle leakage and self-distortion. Inspired by auditory theory (Gabrielsson & Sjögren, 1979; Jekosch, 2004; Wilson & Fazenda, 2014; Bannister et al., 2024), we apply a set of fundamental distortions to every reference waveform, intended to create a wide cover of perceptual auditory field around the reference. These distortions range from mildly-intrusive short-tailed reverberations to highly degrading hard clipping. A pretrained self-supervised model, e.g., wav2vec 2.0 (Baevski et al., 2020), is used to independently encode the waveforms of references, distortions, and system outputs across all sources, in a resolution as high as 75 frames-per-second. These representations are aggregated and projected via a manifold learning technique called diffusion maps (Coifman & Lafon, 2006) onto a low-dimensional manifold. A key property of diffusion maps aligns Euclidean distances between points on the manifold with dissimilarities between their encoded representations. On the manifold, PM quantifies self-distortion by measuring how far an output lies from its attributed reference and the distortions, whereas PS quantifies leakage by comparing these distances with the output proximity to non-attributed references and distortions.

Evaluations on the SEBASS database (Kastner & Herre, 2022) with mixtures of English, Spanish, and music, show that compared to 18 widely used measures, PS and PM almost always achieve first- or second-place rankings in both linear and rank correlations with human scores, with the exception of Spanish rank correlations, where they remain within the top third. We derive granular theoretical deterministic error radius and high-probability confidence intervals (CIs) for both measures, enabling frame-level guarantees on the reliability of the measures. In almost all scenarios, the worst-case error radius would not lower the PS and PM rankings. In addition, the normalized mutual information (NMI) (Danon et al., 2005) between the PS and PM values shows that they are highly complementary.

## 2 PROBLEM FORMULATION

**Notational remark.** Column vectors and matrices are written in bold and other symbols in non-bold.

Consider a source separation system performing inference on an audio mixture (Vincent et al., 2018). In a time frame $f$ that consists of $L$ samples, let $N_f \geq 2$ denote the number of active sources and $\mathcal{S}_f$ their index set. The observed mixture $\mathbf{z}_f \in \mathbb{R}^L$ is modeled as:

$$\mathbf{z}_f = \sum_{i \in \mathcal{S}_f} \mathbf{y}_{i,f} + \mathbf{v}_f. \tag{1}$$

For $i \in \mathcal{S}_f$, we denote $\mathbf{y}_{i,f} \in \mathbb{R}^L$ the reference signal of the $i$-th source in frame $f$, potentially including interference inherent to its original conditions. $\mathbf{v}_f$ represents system and environmental interference, assumed statistically independent of the sources. The estimation of $\mathbf{y}_{i,f}$ is denoted $\hat{\mathbf{y}}_{i,f}$.

Given source indices $i, j \in \mathcal{S}_f$ in time frame $f$, our goal is to introduce these two measures:

- The perceptual separation (PS) measure quantifies how well $\hat{\mathbf{y}}_{i,f}$ is perceptually separated from all interfering sources $\{\mathbf{y}_{j,f}\}_{j \neq i}$.
- The perceptual match (PM) measure quantifies how closely the estimated source $\hat{\mathbf{y}}_{i,f}$ perceptually aligns with its reference $\mathbf{y}_{i,f}$.

## 3 DIFFUSION MAPS: THEORETICAL FOUNDATIONS

**Notational remark.** Sections are denoted by §. Symbols are carried over from §2, except for indices $i, j$ that are repurposed, and the subscript $f$ that is dropped since we analyze a fixed time frame.

Diffusion maps is a manifold learning method that represents high-dimensional data in a low-dimensional space by capturing geometric and structural relationships (Coifman & Lafon, 2006). Consider the set $\mathcal{X} = \{\mathbf{x}_i\}_{i=1}^N$ with $\mathbf{x}_i \in \mathbb{R}^M$ for all $i$, e.g., feature vectors from wav2vec 2.0 (Baevski et al., 2020). An affinity matrix $\mathbf{K} \in \mathbb{R}^{N \times N}$ is calculated between the high-dimensional vectors:

$$\mathbf{K}_{i,j} = \exp\left(-\frac{\|\mathbf{x}_i - \mathbf{x}_j\|_2^2}{\sigma_{\mathbf{K}}^2}\right), \tag{2}$$

where $i, j \in \{1, \ldots, N\}$ and $\forall i, j : 0 \leq K_{i,j} \leq 1$, and $\sigma_{\mathbf{K}}^2 = \text{median}\left\{\|\mathbf{x}_i - \mathbf{x}_j\|_2^2 \,\middle|\, i \neq j\right\}$. To account for non-uniform sampling density of points, an $\alpha$-normalization replaces $\mathbf{K}$ by $\mathbf{K}^{(\alpha)}$:

$$\mathbf{K}_{i,j}^{(\alpha)} = \frac{\mathbf{K}_{i,j}}{(v_i v_j)^\alpha}, \tag{3}$$

where $\alpha \in [0, 1]$ and $v_i = \sum_{j=1}^N \mathbf{K}_{i,j}$. Then, we define the diagonal degree-matrix $\mathbf{D}^{(\alpha)}$, given by $\mathbf{D}^{(\alpha)} = \text{diag}\left(v_0^{(\alpha)}, \ldots, v_{N-1}^{(\alpha)}\right) \in \mathbb{R}^{N \times N}$, where $v_i^{(\alpha)} = \sum_{j=1}^N \mathbf{K}_{i,j}^{(\alpha)}$ and $\forall i : v_i^{(\alpha)} > 0$ by construction. We assume $\alpha$ is fixed and for readability we neglect the $\alpha$ notation from now on.

The probability transition operator $\mathbf{P}$ on $\mathbf{K}$ is defined with (3) as:

$$\mathbf{P} = \mathbf{D}^{-1}\mathbf{K} \in \mathbb{R}^{N \times N}. \tag{4}$$

Note $\mathbf{P}$ is row-stochastic, so $\forall i, j : \mathbf{P}_{ij} \geq 0, \ \sum_{j=1}^N \mathbf{P}_{ij} = 1$. Spectral decomposition on $\mathbf{P}$ reveals a trivial right eigenvector $\mathbf{u}_0 = \mathbf{1} \in \mathbb{R}^N$ with eigenvalue $\lambda_0 = 1$. Remaining eigenvectors $\{\mathbf{u}_\ell\}_{\ell=1}^{N-1}$ are associated with eigenvalues $\{\lambda_\ell\}_{\ell=1}^{N-1}$ and ordered as $1 > \lambda_1 \geq \lambda_2 \geq \cdots \geq \lambda_{N-1} > 0$, so that:

$$\mathbf{P}\mathbf{u}_\ell = \lambda_\ell \mathbf{u}_\ell. \tag{5}$$

Denoting $\mathbf{u}_j(i)$ the $i$-th element of the $j$-th eigenvector, then the embedding of $\mathbf{x}_i$ onto manifold $\mathcal{M}$ can be expressed with the eigenfunctions in (5), by the embedding operation $\Psi_t : \mathbb{R}^M \to \mathbb{R}^{N-1}$:

$$\boldsymbol{\Psi}_t(\mathbf{x}_i) = \left(\lambda_1^t \mathbf{u}_1(i), \lambda_2^t \mathbf{u}_2(i), \ldots, \lambda_{N-1}^t \mathbf{u}_{N-1}(i)\right)^T. \tag{6}$$

where $t$ is the number of Markov chain steps, controlling the diffusion scale of the embedding. The eigenvalues in $\{\mathbf{u}_\ell\}_{\ell=1}^{N-1}$ are orthonormal under the stationary measure $\boldsymbol{\pi} = [\pi_1, \pi_2, \ldots, \pi_N]^T$:

$$\pi_i = \frac{\mathbf{D}_{ii}}{\sum_{j=1}^N \mathbf{D}_{jj}}, \quad \pi_i \in (0, 1). \tag{7}$$

Let $D_t(i, j)$ be the diffusion distance at time step $t$ between two points $\mathbf{x}_i$ and $\mathbf{x}_j$:

$$D_t^2(i, j) = \sum_{m=1}^N \frac{\left(\mathbf{P}_{im}^t - \mathbf{P}_{jm}^t\right)^2}{\pi_m} \tag{8}$$

where $\mathbf{P}_{im}^t$ (4) denotes the probability of transitioning from node $i$ to node $m$ in $t$ time steps. Intuitively, the diffusion distance measures the similarity between the probability distributions of random walks starting from nodes $i$ and $j$. A key strength of diffusion maps is the equivalence (6):

$$D_t^2(i, j) = \left\|\boldsymbol{\Psi}_t(\mathbf{x}_i) - \boldsymbol{\Psi}_t(\mathbf{x}_j)\right\|_2^2, \tag{9}$$

which is fundamental to our approach, as it ensures that the Euclidean distances between every two points on the manifold, which we measure in §4.2 and §4.3, align with dissimilarities between their matching high-dimensional points, represented by the diffusion distance (8). The embedding in (6) is truncated to its first $d$ coordinates and discards the rest. This reduces noise sensitivity and retains the most meaningful geometric structures (Nadler et al., 2006). The mapping $\Psi_t^{(d)} : \mathbb{R}^M \to \mathbb{R}^d$ gives:

$$\boldsymbol{\Psi}_t^{(d)}(\mathbf{x}_i) = \left(\lambda_1^t \mathbf{u}_1(i), \lambda_2^t \mathbf{u}_2(i), \ldots, \lambda_d^t \mathbf{u}_d(i)\right)^T. \tag{10}$$

Consider $\tau \in [0, 1]$ as the minimal normalized retained sum of the eigenvalues, then $d$ is given by:

$$d = \min\left\{k \in \{1, \ldots, N\} : \frac{\sum_{\ell=1}^k \lambda_\ell}{\sum_{\ell=1}^N \lambda_\ell} \geq \tau\right\}. \tag{11}$$

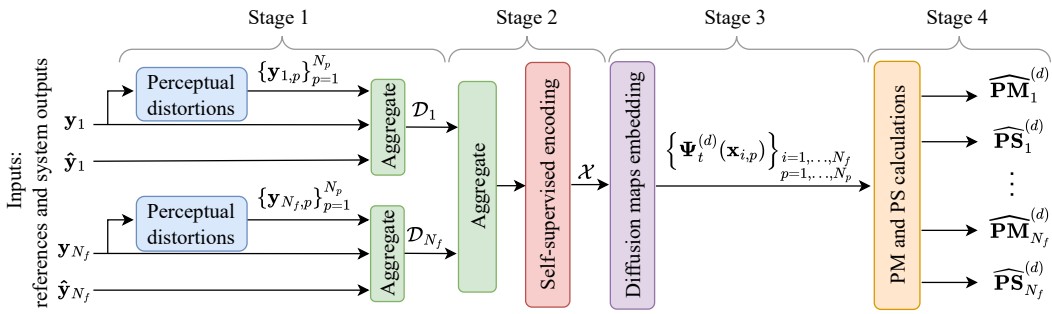

Figure 1: Overview of the proposed MAPSS pipeline. **Stage 1**. Each reference source among the $N_f$ sources in the mixture is independently augmented with a bank of perceptual distortions (§4.1). **Stage 2**. All distorted samples, references, and system outputs across the $N_f$ sources are aggregated and encoded into self-supervised representations (§4.1). **Stage 3**. Diffusion maps embed these representations into a low-dimensional perceptual manifold (§3, §4.1). **Stage 4**. The PS and PM measures are computed over this manifold to quantify self-distortion and leakage (§4.2, §4.3).

## 4 THE PERCEPTUAL SEPARATION AND PERCEPTUAL MATCH MEASURES

### 4.1 CONSTRUCTING PERCEPTUAL CLUSTERS ON THE MANIFOLD

The waveform reference signal of the $i$-th source, $\mathbf{y}_i$, undergoes $N_p$ perceptual distortions, e.g., noise gating in different thresholds, vibrato in various rates, and a comb filter with several delay-gain pairs. Typically, $N_p \in [60, 70]$. We define the $i$-th distortion set $\mathcal{D}_i$ as:

$$\mathcal{D}_i = \left\{ \hat{\mathbf{y}}_i, \mathbf{y}_i, \mathbf{y}_{i,1}, \ldots, \mathbf{y}_{i,N_p} \right\}, \quad \forall p \in \{1, \ldots, N_p\} : \mathbf{y}_{i,p} \in \mathbb{R}^L, \tag{12}$$

with $L$ from (1). Each waveform in $\mathcal{D}_i$ is independently encoded via a pre-trained self-supervised model, e.g., wav2vec 2.0 (Baevski et al., 2020). Let $\Phi : \mathbb{R}^L \to \mathbb{R}^M$ be this encoding operator, with $M$ from §3, so $\mathbf{x}_{i,p} = \Phi\left(\mathbf{y}_{i,p}\right)$, $\mathbf{x}_i = \Phi\left(\mathbf{y}_i\right)$, $\hat{\mathbf{x}}_i = \Phi\left(\hat{\mathbf{y}}_i\right)$. Applying (12) across all $N_f$ sources results in the high-dimensional set of representations:

$$\mathcal{X} = \left\{ \hat{\mathbf{x}}_i, \mathbf{x}_i, \mathbf{x}_{i,1}, \ldots, \mathbf{x}_{i,N_p} \mid i = 1, \ldots, N_f \right\}, \tag{13}$$

with $|\mathcal{X}| = N_f\left(N_p + 2\right) := N$. We define the $i$-th perceptual cluster $\mathcal{C}_i^{(d)}$ on manifold $\mathcal{M}^{(d)}$ (10):

$$\mathcal{C}_i^{(d)} = \left\{ \mathbf{\Psi}_t^{(d)}(\mathbf{x}_i), \mathbf{\Psi}_t^{(d)}(\mathbf{x}_{i,p}) \mid p = 1, \ldots, N_p \right\}. \tag{14}$$

where we exclude the embedding of the system output $\mathbf{\Psi}_t^{(d)}(\hat{\mathbf{x}}_i) \in \mathbb{R}^d$ (10) from $\mathcal{C}_i^{(d)}$, since this embedding will be measured against the cluster statistics to produce the PS and PM measures. Including $\mathbf{\Psi}_t^{(d)}(\hat{\mathbf{x}}_i)$ in the cluster would create a circular dependency that will bias the PS and PM. These distortions were hand-crafted to create a wide perceptual auditory coverage relative to the reference, e.g., by considering mildly-intrusive additive colored noise with signal-to-noise-ratios (SNRs) of 15 dB on one hand, and severely degrading heavy-tailed reverberations on the other hand.

Given $\mathbf{x}_i, \mathbf{x}_j \in \mathcal{X}$, the property in (9) guarantees that as the Euclidean distance between $\mathbf{\Psi}_t^{(d)}(\mathbf{x}_i)$ and $\mathbf{\Psi}_t^{(d)}(\mathbf{x}_j)$ lowers, so does the diffusion distance between $\mathbf{x}_i$ and $\mathbf{x}_j$. In §4.2 and §4.3, we define our PS and PM measures using Euclidean distances, based on our hypothesis that this diffusion distance also aligns with the perceptual alignment between the corresponding waveforms, $\mathbf{y}_i$ and $\mathbf{y}_j$. In §7, we explore if this perceptual-geometric hypothesis is valid by comparing our measures with human perception.

### 4.2 THE PERCEPTUAL SEPARATION (PS) MEASURE

For readability, we denote the elements of the clusters in (14) as $\boldsymbol{\psi}$ both here and in §4.3.

For source $i$, we aim to quantify the perceptual separation of $\hat{\mathbf{y}}_i$ from its non-attributed references $\{\mathbf{y}_j\}_{i \neq j}$ with the Mahalanobis distance (Evans et al., 2023). The empirical centroid and unbiased covariance matrix of the cluster $\mathcal{C}_j^{(d)}$ are:

$$\hat{\boldsymbol{\mu}}_j^{(d)} = \frac{1}{\left|\mathcal{C}_j^{(d)}\right|} \sum_{\boldsymbol{\psi} \in \mathcal{C}_j^{(d)}} \boldsymbol{\psi}, \quad \widehat{\boldsymbol{\Sigma}}_j^{(d)} = \frac{1}{\left|\mathcal{C}_j^{(d)}\right| - 1} \sum_{\boldsymbol{\psi} \in \mathcal{C}_j^{(d)}} \left(\boldsymbol{\psi} - \hat{\boldsymbol{\mu}}_j^{(d)}\right) \left(\boldsymbol{\psi} - \hat{\boldsymbol{\mu}}_j^{(d)}\right)^T, \quad (15)$$

where $\hat{\boldsymbol{\mu}}_j^{(d)} \in \mathbb{R}^d$, $\widehat{\boldsymbol{\Sigma}}_j^{(d)} \in \mathbb{R}^{d \times d}$. The squared Mahalanobis distance from the embedding of the $i$-th output $\boldsymbol{\Psi}_t^{(d)}(\hat{\mathbf{x}}_i)$ to $\mathcal{C}_j^{(d)}$ is given by:

$$d_M^2 \left(\boldsymbol{\Psi}_t^{(d)}(\hat{\mathbf{x}}_i); \hat{\boldsymbol{\mu}}_j^{(d)}, \widehat{\boldsymbol{\Sigma}}_j^{(d)}\right) = \left(\boldsymbol{\Psi}_t^{(d)}(\hat{\mathbf{x}}_i) - \hat{\boldsymbol{\mu}}_j^{(d)}\right)^T \left(\widehat{\boldsymbol{\Sigma}}_j^{(d)} + \epsilon I^{(d)}\right)^{-1} \left(\boldsymbol{\Psi}_t^{(d)}(\hat{\mathbf{x}}_i) - \hat{\boldsymbol{\mu}}_j^{(d)}\right), \quad (16)$$

where we use for regularization $\epsilon = 10^{-6}$ with the $d$-dimensional identity matrix $I^{(d)}$. We define the measured Mahalanobis distance from $\boldsymbol{\Psi}_t^{(d)}(\hat{\mathbf{x}}_i)$ to its attributed and closest non-attributed clusters as:

$$\hat{A}_i^{(d)} = d_M \left(\boldsymbol{\Psi}_t^{(d)}(\hat{\mathbf{x}}_i); \hat{\boldsymbol{\mu}}_i^{(d)}, \widehat{\boldsymbol{\Sigma}}_i^{(d)}\right), \quad \hat{B}_i^{(d)} = d_M \left(\boldsymbol{\Psi}_t^{(d)}(\hat{\mathbf{x}}_i); \hat{\boldsymbol{\mu}}_{j^*}^{(d)}, \widehat{\boldsymbol{\Sigma}}_{j^*}^{(d)}\right), \quad (17)$$

with $j^* = \arg\min_{j \in \{1, \dots, N_f\}, j \neq i} d_M \left(\boldsymbol{\Psi}_t^{(d)}(\hat{\mathbf{x}}_i); \boldsymbol{\mu}_j^{(d)}, \boldsymbol{\Sigma}_j^{(d)}\right)$. Notice that (17) resembles the source permutation minimization processing in source separation evaluations (Le Roux et al., 2019).

The measured PS score for $\hat{\mathbf{y}}_i$ in the truncated dimension $d$ is:

$$\widehat{\text{PS}}_i^{(d)} = 1 - \frac{\hat{A}_i^{(d)}}{\hat{A}_i^{(d)} + \hat{B}_i^{(d)}}, \quad \widehat{\text{PS}}_i^{(d)} \in [0, 1]. \quad (18)$$

where by design $\hat{A}_i^{(d)} + \hat{B}_i^{(d)} > 0$ and a higher score is better. Functionally, when $\hat{A}_i^{(d)} \ll \hat{B}_i^{(d)}$ then the $i$-th output perceptually resembles its cluster members significantly more than competing cluster members and $\widehat{\text{PS}}_i^{(d)}$ approaches 1. $\hat{B}_i^{(d)} \ll \hat{A}_i^{(d)}$ indicates the opposite, and $\widehat{\text{PS}}_i^{(d)}$ drops towards 0.

## 4.3 THE PERCEPTUAL MATCH (PM) MEASURE

The PM measure aims to quantify how perceptually aligned the estimated output $\hat{\mathbf{y}}_i$ is with its reference $\mathbf{y}_i$. Let $\tilde{\mathcal{C}}_i^{(d)} = \mathcal{C}_i^{(d)} \setminus \Psi_t^{(d)}(\mathbf{x}_i)$ denote the reference-free $i$-th cluster. Unlike Equation (15), we compute the unbiased empirical covariance matrix of $\tilde{\mathcal{C}}_i^{(d)}$ relative to its reference embedding:

$$\widehat{\tilde{\boldsymbol{\Sigma}}}_i^{(d)} = \frac{1}{\left|\tilde{\mathcal{C}}_i^{(d)}\right| - 1} \sum_{\boldsymbol{\psi} \in \tilde{\mathcal{C}}_i^{(d)}} \left(\boldsymbol{\psi} - \boldsymbol{\Psi}_t^{(d)}(\mathbf{x}_i)\right) \left(\boldsymbol{\psi} - \boldsymbol{\Psi}_t^{(d)}(\mathbf{x}_i)\right)^T. \quad (19)$$

Then, for $p \in \{1, \dots, N_p\}$, the squared Mahalanobis distance from the $p$-th distortion to its attributed reference in the $i$-th cluster, is given by $d_M^2 \left(\boldsymbol{\Psi}_t^{(d)}(\mathbf{x}_{i,p}); \boldsymbol{\Psi}_t^{(d)}(\mathbf{x}_i), \widehat{\tilde{\boldsymbol{\Sigma}}}_i^{(d)}\right)$, following the definition in Equation (16). Let us define the set of distances:

$$\hat{\mathcal{G}}_i^{(d)} = \left\{ d_M^2 \left(\boldsymbol{\Psi}_t^{(d)}(\mathbf{x}_{i,p}); \boldsymbol{\Psi}_t^{(d)}(\mathbf{x}_i), \widehat{\tilde{\boldsymbol{\Sigma}}}_i^{(d)}\right) \mid p = 1, \dots, N_p \right\}. \quad (20)$$

Empirically, we validated that nearly always these distances are well-approximated by a Gamma distribution, using Kolmogorov-Smirnov goodness-of-fit tests (Smirnov, 1948; Kolmogorov, 1986). The sample mean and unbiased variance of $\hat{\mathcal{G}}_i^{(d)}$ are estimated by:

$$\hat{\mu}_{\mathcal{G}_i^{(d)}} = \frac{1}{\left|\hat{\mathcal{G}}_i^{(d)}\right|} \sum_{g \in \hat{\mathcal{G}}_i^{(d)}} g, \quad \hat{\sigma}_{\mathcal{G}_i^{(d)}}^2 = \frac{1}{\left|\hat{\mathcal{G}}_i^{(d)}\right| - 1} \sum_{g \in \hat{\mathcal{G}}_i^{(d)}} \left(g - \hat{\mu}_{\mathcal{G}_i^{(d)}}\right)^2, \quad (21)$$

and can be moment-matched with a Gamma distribution, assuming $\hat{\mu}_{\mathcal{G}_i^{(d)}}, \hat{\sigma}_{\mathcal{G}_i^{(d)}}^2 > 0$, with parameters:

$$\hat{k}_i^{(d)} = \frac{\hat{\mu}_{\mathcal{G}_i^{(d)}}^2}{\hat{\sigma}_{\mathcal{G}_i^{(d)}}^2}, \quad \hat{\theta}_i^{(d)} = \frac{\hat{\sigma}_{\mathcal{G}_i^{(d)}}^2}{\hat{\mu}_{\mathcal{G}_i^{(d)}}}. \quad (22)$$

Similarly, the squared Mahalanobis distance from the output embedding to its attributed reference is $\hat{a}_i^{(d)} = d_M^2 \left( \boldsymbol{\Psi}_t^{(d)}(\hat{\mathbf{x}}_i); \boldsymbol{\Psi}_t^{(d)}(\mathbf{x}_i), \widehat{\boldsymbol{\widetilde{\Sigma}}}_i^{(d)} \right)$. Consider $Q(k, x) = \Gamma(k, x)/\Gamma(k)$ as the regularized upper incomplete Gamma function (of Mathematical Functions, 2024). Then, the PM score for $\hat{\mathbf{y}}_i$ in dimension $d$ is:

$$\widehat{\text{PM}}_i^{(d)} = Q\left( \hat{k}_i^{(d)}, \frac{\hat{a}_i^{(d)}}{\hat{\theta}_i^{(d)}} \right), \quad \widehat{\text{PM}}_i^{(d)} \in [0, 1], \tag{23}$$

where $\hat{k}_i^{(d)}, \hat{\theta}_i^{(d)}$ are well-defined by design for $N_p \geq 1$ and a higher score is better. If the output $\hat{a}_i^{(d)}$ lies well within the bulk of its distortion cluster, the Gamma-tail probability is near 1, which may indicate a strong perceptual match. As $\hat{a}_i^{(d)}$ drifts away, the score decays smoothly toward zero, reflecting degradation. When distortions are tightly concentrated and $\hat{k}_i^{(d)}$ or $\hat{\theta}_i^{(d)}$ lower, even small mismatches in $\hat{a}_i^{(d)}$ lower PM sharply. As $\hat{k}_i^{(d)}$ and $\hat{\theta}_i^{(d)}$ grow, the PM tolerates larger $\hat{a}_i^{(d)}$ deviations.

## 5    ERROR GUARANTEES FOR THE PS AND PM MEASURES

**Standing notation and assumptions.** We fix frame $f$ with generally $N_f \geq 2$ (1). For this proof, consider the specific case of $N_f = 2$ with indices $i, j \in S_f$. Consider source index $j$ and set $m = N - 1 - d$ as the dimension of the omitted space in the diffusion maps process, so the embedding notations in the retained, omitted, and complete $N - 1$-dimensional spaces are respectively $\boldsymbol{\Psi}_t^{(d)}(\mathbf{x}_j), \boldsymbol{\Psi}_t^{(m)}(\mathbf{x}_j), \boldsymbol{\Psi}_t(\mathbf{x}_j)$ (10). Similarly, clusters $\mathcal{C}_j^{(d)}, \mathcal{C}_j^{(m)}$, and $\mathcal{C}_j$ are formed as in §4.1, with means and covariances pairs $\left( \boldsymbol{\mu}_j^{(d)}, \boldsymbol{\Sigma}_j^{(d)} \right)$, $\left( \boldsymbol{\mu}_j^{(m)}, \boldsymbol{\Sigma}_j^{(m)} \right)$, and $(\boldsymbol{\mu}_j, \boldsymbol{\Sigma}_j)$, and cross-covariance $\boldsymbol{C}_j \in \mathbb{R}^{d \times m}$. We have empirically prevented ill-conditioning, as all matrix inversions are Tikhonov-regularized (Tikhonov & Arsenin, 1977) with $\epsilon I$, where $\epsilon = 10^{-6}$ and $I$ the identity matrix, with context-dependent dimension. When we quantify sampling uncertainty, we use dependence-adjusted effective sample sizes via Bartlett method (Bartlett, 1946), and sub-Gaussian tails for quadratic forms via the dependent Hanson–Wright inequalities (Adamczak, 2014; Vershynin, 2024).

**Schur decomposition of full versus truncated Mahalanobis distances.** For radius error calculations, from (24) to (36), we assume access to clusters statistics. For the output embedding of source $i$ against cluster $j$, define $\boldsymbol{\Delta}_{i,j}^{(d)} = \boldsymbol{\Psi}_t^{(d)}(\hat{\mathbf{x}}_i) - \boldsymbol{\mu}_j^{(d)}$ and $\boldsymbol{\Delta}_{i,j}^{(m)} = \boldsymbol{\Psi}_t^{(m)}(\hat{\mathbf{x}}_i) - \boldsymbol{\mu}_j^{(m)}$. The full cluster statistics aggregate as:

$$\boldsymbol{\mu}_j = \begin{bmatrix} \boldsymbol{\mu}_j^{(d)} \\ \boldsymbol{\mu}_j^{(m)} \end{bmatrix}, \quad \boldsymbol{\Delta}_{i,j} = \begin{bmatrix} \boldsymbol{\Delta}_{i,j}^{(d)} \\ \boldsymbol{\Delta}_{i,j}^{(m)} \end{bmatrix}, \quad \boldsymbol{\Sigma}_j = \begin{bmatrix} \boldsymbol{\Sigma}_j^{(d)} & \boldsymbol{C}_j \\ \boldsymbol{C}_j^T & \boldsymbol{\Sigma}_j^{(m)} \end{bmatrix}. \tag{24}$$

Block inversion to (16) via the Schur complement (Horn & Johnson, 2012) yields:

$$d_M^2(\boldsymbol{\Psi}_t(\hat{\mathbf{x}}_i); \boldsymbol{\mu}_j, \boldsymbol{\Sigma}_j) = \underbrace{\left( \boldsymbol{\Delta}_{i,j}^{(d)} \right)^T \left( \boldsymbol{\Sigma}_j^{(d)} + \epsilon I^{(d)} \right)^{-1} \boldsymbol{\Delta}_{i,j}^{(d)}}_{:= a} + \underbrace{\boldsymbol{r}_{i,j}^T \boldsymbol{S}_j^{-1} \boldsymbol{r}_{i,j}}_{:= b}, \tag{25}$$

$$\boldsymbol{r}_{i,j} = \boldsymbol{\Delta}_{i,j}^{(m)} - \boldsymbol{C}_j^T \left( \boldsymbol{\Sigma}_j^{(d)} + \epsilon I^{(d)} \right)^{-1} \boldsymbol{\Delta}_{i,j}^{(d)}, \quad \boldsymbol{S}_j = \boldsymbol{\Sigma}_j^{(m)} - \boldsymbol{C}_j^T \left( \boldsymbol{\Sigma}_j^{(d)} + \epsilon I^{(d)} \right)^{-1} \boldsymbol{C}_j. \tag{26}$$

Since $\forall a, b \geq 0 : \left| \sqrt{a + b} - \sqrt{a} \right| \leq \sqrt{b}$ (Rudin, 1976, Ch. 5), we bound truncation error to (25):

$$|\delta_{i,j}| := \left| d_M(\boldsymbol{\Psi}_t(\hat{\mathbf{x}}_i); \boldsymbol{\mu}_j, \boldsymbol{\Sigma}_j) - d_M^2 \left( \boldsymbol{\Psi}_t^{(d)}(\hat{\mathbf{x}}_i); \boldsymbol{\mu}_j^{(d)}, \boldsymbol{\Sigma}_j^{(d)} \right) \right| \leq \sqrt{\boldsymbol{r}_{i,j}^T \boldsymbol{S}_j^{-1} \boldsymbol{r}_{i,j}}. \tag{27}$$

**PS radius.** Let $A_i, B_i$ be the full-space versions of $A_i^{(d)}, B_i^{(d)}$ in (17). Set $|\delta_{i,i}| := |A_i - A_i^{(d)}|$ and $|\delta_{i,j^*}| := |B_i - B_i^{(d)}|$, with $j^*$ as in (17). We empirically confirmed that truncation introduces only mild changes, i.e., $|\delta_{i,i}|, |\delta_{i,j^*}| \ll A_i^{(d)} + B_i^{(d)}$. Thus, a first-order Taylor expansion of $\text{PS}_i$, the full-space version of $\text{PS}_i^{(d)}$, around $\left( A_i^{(d)}, B_i^{(d)} \right)$, is valid. Ultimately, we drop quadratic components that were found negligible, and use $|\delta_{i,i}|$ and $|\delta_{i,j^*}|$ inside the Taylor expansion, to yield:

$$\left| \text{PS}_i - \text{PS}_i^{(d)} \right| \leq \frac{B_i^{(d)} |\delta_{i,i}| + A_i^{(d)} |\delta_{i,j^*}|}{\left( A_i^{(d)} + B_i^{(d)} \right)^2}. \tag{28}$$

Combining with (27), the deterministic PS error radius is:

$$\left| \mathrm{PS}_i - \mathrm{PS}_i^{(d)} \right| \leq \frac{B_i^{(d)} \sqrt{\boldsymbol{r}_{i,i}^T \boldsymbol{S}_i^{-1} \boldsymbol{r}_{i,i}} + A_i^{(d)} \sqrt{\boldsymbol{r}_{i,j^*}^T \boldsymbol{S}_{j^*}^{-1} \boldsymbol{r}_{i,j^*}}}{\left( A_i^{(d)} + B_i^{(d)} \right)^2}. \tag{29}$$

We notice that large residual spread $\boldsymbol{\Sigma}_j^{(m)}$ or cross-block coupling $\boldsymbol{C}_j$ inflate (29) through $\boldsymbol{S}_j^{-1}$.

**PM radius.** For source $i$ and every distortion index $p \in \{1, \dots, N_p\}$, we center cluster coordinates at the reference $\mathbf{x}_i$, so $\boldsymbol{\Delta}_{i,p}^{(d)} = \boldsymbol{\Psi}_t^{(d)}(\mathbf{x}_{i,p}) - \boldsymbol{\Psi}_t^{(d)}(\mathbf{x}_i)$ and $\boldsymbol{\Delta}_{i,p}^{(m)} = \boldsymbol{\Psi}_t^{(m)}(\mathbf{x}_{i,p}) - \boldsymbol{\Psi}_t^{(m)}(\mathbf{x}_i)$. By repeating (24) for the reference-free clusters in the $d$, $m$, and $N-1$ spaces, we obtain:

$$\widetilde{\boldsymbol{\Sigma}}_i = \begin{bmatrix} \widetilde{\boldsymbol{\Sigma}}_i^{(d)} & \widetilde{\boldsymbol{C}}_i \\ \widetilde{\boldsymbol{C}}_i^T & \widetilde{\boldsymbol{\Sigma}}_i^{(m)} \end{bmatrix}, \tag{30}$$

where $\widetilde{\boldsymbol{\Sigma}}_i^{(d)}$ is defined in (19). Exactly as in (25)-(26), we use Schur complement:

$$d_M^2 \left( \boldsymbol{\Psi}_t(\mathbf{x}_{i,p}); \boldsymbol{\Psi}_t(\mathbf{x}_i), \widetilde{\boldsymbol{\Sigma}}_i \right) = \left( \boldsymbol{\Delta}_{i,p}^{(d)} \right)^T \left( \widetilde{\boldsymbol{\Sigma}}_i^{(d)} + \epsilon I)^{-1} \right) \boldsymbol{\Delta}_{i,p}^{(d)} + \boldsymbol{r}_{i,p}^T \boldsymbol{S}_i^{-1} \boldsymbol{r}_{i,p}, \tag{31}$$

$$\boldsymbol{r}_{i,p} = \boldsymbol{\Delta}_{i,p}^{(m)} - \widetilde{\boldsymbol{C}}_i^T \left( \widetilde{\boldsymbol{\Sigma}}_i^{(d)} + \epsilon I \right)^{-1} \boldsymbol{\Delta}_{i,p}^{(d)}, \quad \boldsymbol{S} = \widetilde{\boldsymbol{\Sigma}}_i^{(m)} - \widetilde{\boldsymbol{C}}_i^T \left( \widetilde{\boldsymbol{\Sigma}}_i^{(d)} + \epsilon I \right)^{-1} \widetilde{\boldsymbol{C}}_i. \tag{32}$$

Let $\mathcal{G}_i$ be the set of the squared distances in (31) over $p$ and $\mathcal{G}_i^{(d)}$ its $d$-dimensional analogue (20). Define per-sample truncation gaps $\delta_{\mathcal{G}_i,p} := \boldsymbol{r}_{i,p}^T \boldsymbol{S}_i^{-1} \boldsymbol{r}_{i,p} \geq 0$ and $\delta_{\max} = \max_p \delta_{\mathcal{G}_i,p}$. Employing elementary algebra and Cauchy–Schwarz inequality (Vershynin, 2024), we obtain the relations (21):

$$\left| \mu_{\mathcal{G}_i} - \mu_{\mathcal{G}_i^{(d)}} \right| = \frac{1}{N_p} \sum_{p=1}^{N_p} \delta_{\mathcal{G}_i,p}, \quad \left| \sigma_{\mathcal{G}_i}^2 - \sigma_{\mathcal{G}_i^{(d)}}^2 \right| \leq \frac{N_p}{N_p - 1} \left( 2\delta_{\max} \left( \sigma_{\mathcal{G}_i} + \sigma_{\mathcal{G}_i^{(d)}} \right) + \delta_{\max}^2 \right). \tag{33}$$

Again, simple algebra bounds the Gamma-matching parameters (22), with constants $C_1, C_2 > 0$:

$$\left| k_i - k_i^{(d)} \right| \leq C_1 \, \delta_{\max} \frac{N_p}{N_p - 1} \frac{\mu_{\mathcal{G}_i} + \mu_{\mathcal{G}_i^{(d)}}}{\sigma_{\mathcal{G}_i^{(d)}}^2}, \quad \left| \theta_i - \theta_i^{(d)} \right| \leq C_2 \, \delta_{\max} \frac{N_p}{N_p - 1} \frac{\sigma_{\mathcal{G}_i}^2 + \sigma_{\mathcal{G}_i^{(d)}}^2}{\mu_{\mathcal{G}_i^{(d)}}^2}. \tag{34}$$

Recalling the distance of the output from its cluster, denoted $a_i^{(d)}$ (23), we can define using (32):

$$d_M^2 \left( \boldsymbol{\Psi}_t(\hat{\mathbf{x}}_i); \boldsymbol{\Psi}_t(\mathbf{x}_i), \tilde{\boldsymbol{\Sigma}}_i \right) - d_M^2 \left( \boldsymbol{\Psi}_t^{(d)}(\hat{\mathbf{x}}_i); \boldsymbol{\Psi}_t^{(d)}(\mathbf{x}_i), \tilde{\boldsymbol{\Sigma}}_i^{(d)} \right) = \boldsymbol{r}_{i,a}^T \boldsymbol{S}_i^{-1} \boldsymbol{r}_{i,a} := \delta_{\mathcal{G}_i,a}. \tag{35}$$

In the full space, $\mathrm{PM}_i = Q(k_i, a_i/\theta_i)$ (23). Its derivatives with respect to its variables are standard and bounded on compact sets (of Mathematical Functions, 2024). Let the truncation ellipsoid $\mathcal{B}_i$ be such set, so that $\mathcal{B}_i = \left\{ (k_i', \theta_i', a_i') : \left| k_i' - k_i^{(d)} \right| \leq \delta_{\mathcal{G}_i,k}, \left| \theta_i' - \theta_i^{(d)} \right| \leq \delta_{\mathcal{G}_i,\theta}, \left| a_i' - a_i^{(d)} \right| \leq \delta_{\mathcal{G}_i,a} \right\}$, with $\delta_{\mathcal{G}_i,k}, \delta_{\mathcal{G}_i,\theta}$ denoting the bounds in (34). Since $Q$ increases in $k$ and decreases in $x = a/\theta$ for $k, x > 0$ (of Mathematical Functions, 2024), the maximum deviation over $\mathcal{B}_i$ occurs at a corner, and the radius can be obtained by:

$$\left| \mathrm{PM}_i - \mathrm{PM}_i^{(d)} \right| \leq \max_{(k_c, \theta_c, a_c) \in \partial \mathcal{B}_i} \left| Q(k_c, a_c/\theta_c) - Q(k_i^{(d)}, a_i^{(d)}/\theta_i^{(d)}) \right|. \tag{36}$$

**Dependence-adjusted sample size.** For any cluster $\mathcal{C}_j^{(d)}$ with $n_j$ dependent points, we use $n_{j,\mathrm{eff}} := n_j \left( 1 + 2 \sum_{\ell=1}^{L_j} \hat{\rho}_{j,\ell} \right)^{-1}$ with $L_j = \min\{\ell : |\hat{\rho}_{j,\ell}| < z_{0.975}/\sqrt{n_j - \ell}\}$ (Bartlett, 1946).

**PS tail bound.** We now resort to the retained $d$-dimensional space, and estimate cluster statistics due the finite number of $n_{j,\mathrm{eff}}$ samples. Let $\widehat{\boldsymbol{\mu}}_j^{(d)}$ and $\widehat{\boldsymbol{\Sigma}}_j^{(d)}$ be the empirical cluster statistics. Vector and matrix-Bernstein and dependent Hanson–Wright theories (Vershynin, 2024, Props. 2.8.1, 4.7.1), (Adamczak, 2014, Thm. 2.5) give for $\delta_{j,\mu}^{\mathrm{PS}}, \delta_{j,\Sigma}^{\mathrm{PS}} \in (0, 1/2)$, with least probabilities $1 - \delta_{j,\mu}^{\mathrm{PS}}, 1 - \delta_{j,\Sigma}^{\mathrm{PS}}$:

$$\left\| \boldsymbol{\mu}_j^{(d)} - \widehat{\boldsymbol{\mu}}_j^{(d)} \right\|_2 \leq \sqrt{2 \lambda_{\max} \left( \widehat{\boldsymbol{\Sigma}}_j^{(d)} \right) \ln(2/\delta_{j,\mu}^{\mathrm{PS}})/n_{j,\mathrm{eff}}} =: \Delta_{j,\mu}, \tag{37}$$

$$\left\| \boldsymbol{\Sigma}_j^{(d)} - \widehat{\boldsymbol{\Sigma}}_j^{(d)} \right\|_2 \leq C \lambda_{\max} \left( \widehat{\boldsymbol{\Sigma}}_j^{(d)} \right) r_j + \ln(2/\delta_{j,\Sigma}^{\mathrm{PS}})/n_{j,\mathrm{eff}} =: \Delta_{j,\Sigma}, \tag{38}$$

with $r_j = \text{trace}(\widehat{\boldsymbol{\Sigma}}_j^{(d)})/\lambda_{\max}\left(\widehat{\boldsymbol{\Sigma}}_j^{(d)}\right)$. A first-order perturbation of $A_i^{(d)}, B_i^{(d)}$ (17) yields the bounds:

$$\varepsilon^{\text{PS}}\left(\widehat{A}_i^{(d)}\right) \leq 2\sqrt{\widehat{A}_i^{(d)}}\Delta_{i,\boldsymbol{\mu}}\sqrt{\lambda_{\max}\left(\widehat{\boldsymbol{\Sigma}}_i^{(d)}\right)/\tilde{\lambda}_{\min}\left(\widehat{\boldsymbol{\Sigma}}_i^{(d)}\right)} + \widehat{A}_i^{(d)}\Delta_{i,\boldsymbol{\Sigma}}/\lambda_{\max}\left(\widehat{\boldsymbol{\Sigma}}_i^{(d)}\right), \quad (39)$$

$$\varepsilon^{\text{PS}}\left(\widehat{B}_i^{(d)}\right) \leq 2\sqrt{\widehat{B}_i^{(d)}}\Delta_{j^*,\boldsymbol{\mu}}\sqrt{\lambda_{\max}\left(\widehat{\boldsymbol{\Sigma}}_{j^*}^{(d)}\right)/\tilde{\lambda}_{\min}\left(\widehat{\boldsymbol{\Sigma}}_{j^*}^{(d)}\right)} + \widehat{B}_i^{(d)}\Delta_{j^*,\boldsymbol{\Sigma}}/\lambda_{\max}\left(\widehat{\boldsymbol{\Sigma}}_{j^*}^{(d)}\right). \quad (40)$$

With $L_i^{\text{PS}}$ being the Euclidean gradient norm of $\text{PS}_i^{(d)}$ at $(A_i^{(d)}, B_i^{(d)})$, for $\delta_i^{\text{PS}} = \delta_{i,\mu}^{\text{PS}} + \delta_{i,\Sigma}^{\text{PS}} \in (0,1)$

$$\left|\widehat{\text{PS}}_i^{(d)} - \text{PS}_i^{(d)}\right| \leq L_i^{\text{PS}}\sqrt{\varepsilon^{\text{PS}}\left(\widehat{A}_i^{(d)}\right) + \varepsilon^{\text{PS}}(\widehat{B}_i^{(d)})} \quad \text{w.p.} \geq 1 - \delta_i^{\text{PS}}. \quad (41)$$

**PM tail bound.** Let $R_i = \max_{g \in \mathcal{G}_i} g$ and choose confidence levels $\delta_{i,\mu}^{\text{PM}}, \delta_{i,\sigma}^{\text{PM}}, \delta_{i,a}^{\text{PM}} \in (0, 1/3)$. Concentration bounds for the output quadratic form via Hanson–Wright (Vershynin, 2024, Props. 2.8.1, 4.7.1) yields

$$\left|\mu_{\mathcal{G}_i^{(d)}} - \widehat{\mu}_{\mathcal{G}_i^{(d)}}\right| \leq \sqrt{\frac{2\widehat{\sigma}_{\mathcal{G}_i^{(d)}}^2 \ln(2/\delta_{i,\mu}^{\text{PM}})}{N_p}} + \frac{3R_i \ln(2/\delta_{i,\mu}^{\text{PM}})}{N_p}, \quad (42)$$

$$\left|\sigma_{\mathcal{G}_i^{(d)}} - \widehat{\sigma}_{\mathcal{G}_i^{(d)}}\right| \leq \sqrt{\frac{2R_i^2 \ln(2/\delta_{i,\sigma}^{\text{PM}})}{N_p}} + \frac{3R_i^2 \ln(2/\delta_{i,\sigma}^{\text{PM}})}{N_p}, \quad \left|a_i^{(d)} - \widehat{a}_i^{(d)}\right| \leq R_i\sqrt{\frac{\ln(2/\delta_{i,a}^{\text{PM}})}{N_p}}. \quad (43)$$

Just like in (39) and (40), these are mapped to bounds on the parameters $(k, \theta, a)$, yielding $\Delta_{i,k}, \Delta_{i,\theta}, \Delta_{i,a}$. Consider $\delta_i^{\text{PM}} = \delta_{i,\mu}^{\text{PM}} + \delta_{i,\sigma}^{\text{PM}} + \delta_{i,a}^{\text{PM}} \in (0,1)$, then for the local box $\mathcal{B}_i^{\text{loc}} = \{|k - k_i^{(d)}| \leq \Delta_{i,k}, |\theta - \theta_i^{(d)}| \leq \Delta_{i,\theta}, |a - a_i^{(d)}| \leq \Delta_{i,a}\}$:

$$\left|\widehat{\text{PM}}_i^{(d)} - \text{PM}_i^{(d)}\right| \leq \max_{(k_c, \theta_c, a_c) \in \partial\mathcal{B}_i^{\text{loc}}} \left| Q(k_c, a_c/\theta_c) - Q(\widehat{k}_i^{(d)}, \widehat{a}_i^{(d)}/\widehat{\theta}_i^{(d)}) \right| \quad \text{w.p.} \geq 1 - \delta_i^{\text{PM}}. \quad (44)$$

# 6 EXPERIMENTAL SETUP

## 6.1 DATABASE

We use the Subjective Evaluation of Blind Audio Source Separation (SEBASS) database (Kastner & Herre, 2022), a public collection of expertly curated listening tests that aggregates $11,000$ ratings for more than $900$ separated signals across five evaluation campaigns. SEBASS covers speech mixtures of 4 male or 4 female speakers, each consisting of English and Spanish pairs. As realistic conversations are monolingual, we separate each mixture into English and Spanish speakers pairs. Also included are music mixtures with drums and without drums, each with 3 sources. Namely, $N_f \in \{2, 3\}$ (1). The split between drum and no-drum mixtures is crucial, as percussion transients create perceptual and algorithmic masking distinct from harmonic content. Each mixture was processed by 32 source separation systems, ranging from classic approaches to deep-learning models. Outputs with 10 s duration, sampled at 16 kHz, were judged by 15 certified raters under the MUSHRA standard (Schoeffler et al., 2018), which grades output quality between 0 and 100 relative to a reference. These MOS ratings are provided directly by SEBASS and no new human listening tests were conducted by the authors. All experiments in this paper rely solely on the existing SEBASS ratings.

## 6.2 PRE-PROCESSING AND PERFORMANCE EVALUATION

All parameters in this study are selected based on internal properties of diffusion maps and the underlying self-supervised models, or on prior work, and are neither data-driven nor tuned using SEBASS labels. SEBASS provides MOS values at the utterance level. Since our PM and PS measures operate at much finer temporal resolutions, with frame sizes of $L = 400$ for speech and $L = 324$ samples for music (1), aggregation from the frame-level to the utterance-level is required to enable comparison with human MOS. PM values are aggregated using a simple average, while PS values are aggregated with a perceptually-weighted scheme inspired by PESQ. For performance evaluation, we

Table 1: SRCC and PCC of the PS and PM measures (underlined), their waveform counterparts, and 14 comparative measures, across scenarios. The top-3 results in every column are in bold.

| Measure | English | | Spanish | | Music (Drums) | | Music (No Drums) | |
|---|---|---|---|---|---|---|---|---|
| | SRCC | PCC | SRCC | PCC | SRCC | PCC | SRCC | PCC |
| PS | **84.12**% | **83.74**% | 82.33% | **85.01**% | **72.87**% | **77.38**% | 87.23% | **87.81**% |
| PM | **84.69**% | **86.36**% | 83.41% | **85.30**% | **75.18**% | 69.88% | **88.12**% | **85.26**% |
| PS (waveform) | 73.42% | 71.04% | 74.69% | 75.05% | 51.75% | 61.83% | 78.88% | 78.95% |
| PM (waveform) | 69.30% | 66.62% | 68.27% | 67.35% | 49.52% | 51.77% | 74.37% | 75.51% |
| STOI | 80.85% | 78.40% | 78.79% | 82.56% | 67.29% | **71.27**% | 75.64% | 78.13% |
| eSTOI | 82.14% | 82.28% | 79.20% | 82.68% | 54.68% | 57.35% | 70.06% | 74.45% |
| PESQ | **85.56**% | **84.05**% | **86.06**% | **84.98**% | 61.60% | 53.87% | 61.26% | 60.24% |
| SI-SDR | 78.11% | 76.96% | 84.07% | 81.38% | 42.08% | 56.98% | 70.42% | 71.96% |
| SDR | 77.72% | 73.13% | **84.29**% | 76.07% | 44.78% | 54.33% | 74.51% | 75.35% |
| SIR | 51.28% | 56.20% | 45.67% | 55.19% | 18.64% | 35.76% | 51.00% | 55.12% |
| SAR | 75.54% | 72.98% | 78.21% | 73.29% | 36.98% | 40.81% | 66.15% | 68.96% |
| CI-SDR | 78.66% | 77.41% | **84.32**% | 81.48% | 45.02% | 55.42% | 74.25% | 75.11% |
| DNSMOS-OVRL | 63.70% | 67.77% | 35.34% | 43.57% | 21.79% | 34.27% | 13.81% | 19.47% |
| MCD | 43.05% | 33.86% | 45.90% | 37.97% | 30.27% | 42.23% | 33.49% | 32.19% |
| SpeechBERTscore | 68.58% | 67.44% | 69.55% | 70.48% | 52.33% | 59.71% | 75.60% | **81.13**% |
| Sheet-SSQA | 41.17% | 51.38% | 61.06% | 73.01% | 39.40% | 29.03% | 14.19% | 5.17% |
| UTMOS | 55.53% | 55.43% | 52.22% | 55.75% | -9.24% | -8.25% | 12.59% | 7.72% |
| NISQA | 60.78% | 67.62% | 63.37% | 66.58% | 27.27% | 41.73% | 42.33% | 48.07% |
| SCOREQ | 77.02% | 81.85% | 82.32% | 83.13% | 62.73% | 68.56% | 75.35% | 75.84% |
| NORESQA | 55.82% | 41.53% | 58.56% | 47.24% | -2.34% | -0.80% | 61.21% | 65.45% |
| VQScore | 36.08% | 37.31% | 36.93% | 40.65% | 9.75% | -7.68% | 31.14% | 33.83% |
| ViSQOL | 72.59% | 74.55% | 76.48% | 75.48% | **71.94**% | 65.48% | 78.75% | 70.31% |

correlate the aggregated PM and PS values with the utterance-level MOS values using the Pearson product-moment correlation coefficient (PCC) (Benesty et al., 2009) and the Spearman rank-order correlation coefficient (SRCC) (Sedgwick, 2014). We set $\alpha = 1$ (3) to eliminate density-dependent bias from the embedding, and $t = 1$ (6) to keep the diffusion operator focused on local structures. The retained dimension $d$ is in $[20, 40]$ (10), using $\tau = 0.99$ (11), as done on (Fjellström & Nyström, 2022). Although diffusion maps and Laplacian Eigenmaps (LE) Belkin & Niyogi (2003) are spectrally related, our setting with $\alpha = 1$ and $t = 1$ does not reduce to LE. The parameter $\alpha = 1$ introduces a density-normalized kernel that yields an operator approximating intrinsic Laplace-Beltrami geometry rather than the sampling density Coifman & Lafon (2006). Furthermore, diffusion maps embed points via eigenvalue-weighted coordinates, which is essential for preserving the diffusion distance in (9), whereas LE uses unweighted eigenvectors of a graph Laplacian.

# 7 EXPERIMENTAL RESULTS

Results are from zero-shot SEBASS inference, without training or data-driven parameter tuning.

Table 1 benchmarks the proposed PS and PM measures against 14 widely-used metrics for audio quality and also versus its waveform-only version, denoted PS (waveform) and PM (waveform). In this variant, the raw waveforms are passed directly through the diffusion-maps process, skipping the self-supervised representations. This allows us to isolate and quantify the contribution of self-supervised embeddings to the effectiveness of the PS and PM measures. For speech, we used a wav2vec 2.0-based (Baevski et al., 2020) model with features of dimension $M = 1024$ (§3) and 24 transformer layers, and for music we use the MERT model (Li et al., 2024) with $M = 768$ and 12 transformer layers. Previous work has shown that earlier layers of self-supervised speech models are often more perceptually stable (Pasad et al., 2021; 2023). In our experiments, layers 1-3 produce nearly identical performance, with correlation coefficients that differ by less than 1% absolute, as shown in Figure 7 in Appendix C. For concreteness, we illustrate results using layer 2 for speech, layer 1 with drums, and layer 3 without drums. Our conclusions hold uniformly across these layers. PS and PM consistently achieve top PCC values, aside from minor advantages by PESQ and STOI. For SRCC, our measures dominate in music, but trail PESQ in English and SDR-based metrics in

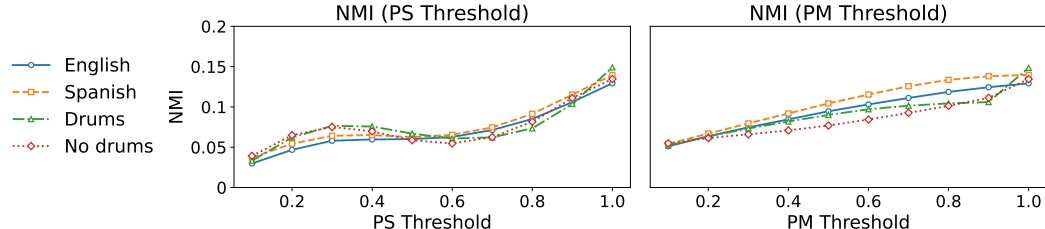

Figure 2: NMI between the PS and PM measures across their thresholded values.

Table 2: Deterministic error radius and probabilistic 95% CI of the SRCC and PCC across scenarios.

| Measure | English | | Spanish | | Music (Drums) | | Music (No Drums) | |
|---|---|---|---|---|---|---|---|---|
| | SRCC | PCC | SRCC | PCC | SRCC | PCC | SRCC | PCC |
| PS radius | 0.16% | 0.21% | 0.10% | 0.14% | 0.40% | 0.72% | 0.14% | 0.11% |
| PS CIs (95%) | 30.03% | 10.29% | 26.39% | 8.85% | 28.71% | 12.21% | 12.69% | 4.11% |
| PM radius | 0.11% | 0.99% | 0.18% | 1.23% | 0.29% | 1.39% | 0.02% | 1.04% |
| PM CIs (95%) | 7.23% | 3.83% | 8.98% | 4.28% | 6.25% | 4.15% | 4.75% | 1.77% |

Spanish. These results position the PS and PM as valid measures for leakage and self-distortion for source separation systems. Encoding proves essential, as waveform-only variants perform worse. Finally, PS and PM outperform SpeechBERTScore, showing the benefit of diffusion maps over cosine similarity.

We examine the complementary relationship between the PS and PM using NMI (Danon et al., 2005), which captures statistical dependence beyond linear effects, with lower NMI indicating less shared information. Each measure is normalized per utterance to $[0, 1]$. For thresholds $\{0.1, 0.2, \ldots, 1\}$, we retain frames with PS below the threshold and compute the NMI between aligned PS-PM pairs. The procedure is repeated with thresholding on the PM. Figure 2 shows the NMI decreases toward zero as thresholds tighten, suggesting that the PS and PM become more complementary when separation quality is poor. At the loosest thresholds, NMI rises up to 0.15, yet full redundancy corresponds to 1. These results support reporting both PS and PM, as each captures failure modes missed by the other.

Frame-level error bounds of the measures were derived in §5. Table 2 presents their propagation into PCC and SRCC error bounds. The error radius never exceeds $1.39\%$, a bias that rarely affects the performance ranking in Table 1. The $95\%$ CIs highlight that the PS carries higher statistical uncertainty, whereas the PM is statistically more robust. This positions the PS as a complementary diagnostic, capturing perceptual leakage that the PM misses, at the cost of greater variability.

Additional experiments that are presented in the Appendices include analyzing the PCC and SRCC error bounds of the measures under different self-supervised models, introducing failing points of the PS and PM relative to PESQ as our strongest competitor, testing the generalization of our measures under out of distribution distortions, demonstrating how the PS and PM each disentangles leakage and self-distortion in audio source separation and emphasizing specifically their advantage relative to the SDR family of measures, and examining our method under multilingual information of both English and Spanish.

## 8 CONCLUSIONS

We introduced the PS and PM, frame-level measures that showed competitive correlations with human MOS for source separation evaluation by operating on diffusion map embeddings of self-supervised audio representations. We derived a deterministic truncation bias and non-asymptotic CIs for both measures, making scores interpretable under quantified uncertainty. Looking forward, PS and PM can serve as diagnostic tools to localize whether errors stem from target distortion or cross-talk, while their differentiability enables use as loss terms or curriculum triggers to balance fidelity and separation under confidence monitoring. Finally, their uncertainty bounds offer a principled layer for benchmarking, supporting fairer hyper-parameter sweeps and reporting standards.

**Ethics Statement.** This work does not involve human subjects or personally identifiable information. We use only existing datasets under their respective licenses and terms of use, and we do not redistribute any data where licenses restrict sharing. Our study complies with the ICLR Code of Ethics. We assessed foreseeable risks and did not identify specific, material harms arising from the methods or results presented here. We will release implementation details and scripts to support responsible reuse and verification.

**Reproducibility Statement.** We provide complete code as an anonymous supplementary material in a separate .zip file. It contains the complete inference pipeline, including the frame-level calculation of the PS and PM measures and their determinstic and probabilistic error bounds.

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

# A PERCEPTUAL DISTORTIONS APPLIED IN THE PS AND PM CALCULATIONS

Table 3: Distortions applied to the references when calculating the PS and PM measures (§4.1). $f_s$ is the sampling frequency, and $A_{95}$ and $A_{RMS}$ mark the 95th-percentile and RMS absolute amplitudes.

| Distortion | PS | PM |
|---|---|---|
| Notch Filter | **Center frequencies:**
500, 1000, 2000, 4000, 8000 Hz | **Number of notches:**
$\leq 20$
**Operating band:**
80 Hz - $0.45 f_s$
**Notch spacing:**
$\geq 300$ Hz
**Bandwidth:**
$\pm 60$ Hz |
| Comb Filter | **Delay:**
2.5-15 ms
**Feedback gain:**
0.4-0.9 | **Delay-gain pairs:**
(2.5 ms, 0.4), (5 ms, 0.5),
(7.5 ms, 0.6), (10 ms, 0.7),
(12.5 ms, 0.9) |
| Tremolo | **Rate:**
1, 2, 4, 6 Hz
**Depth:**
0.3-1.0 | **Rate:**
1, 2, 4, 6 Hz
**Depth:**
1 |
| Additive Noise | **SNR:**
-15, -10, -5, 0, 5, 10, 15 dB
**Noise color:**
white, pink, brown | **SNR:**
-15, -10, -5, 0, 5, 10, 15 dB
**Noise color:**
white, pink, brown |
| Additive Harmonic Tone | **Tone frequency:**
100, 500, 1000, 4000 Hz
**Amplitude:**
0.02-0.08 (absolute) | **Tone frequency:**
100, 500, 1000, 4000 Hz
**Amplitude:**
$\{0.4, 0.6, 0.8, 1\} \times A_{RMS}$ |
| Reverberation | **RT$_{60}$ (Schroeder, 1965):**
0.3-1.1 s
**Early tail length:**
5, 10, 15, 20 ms | **Exponential tail length:**
50, 100, 200, 400 ms
**Decay scaling:**
0.3, 0.5, 0.7, 0.9 |
| Noise Gate | **Threshold:**
0.005, 0.01, 0.02, 0.04 (absolute) | **Threshold:**
$\{0.05, 0.1, 0.2, 0.4\} \times A_{95}$ |
| Pitch Shift | **Offsets:**
-4, -2, +2, +4 semitones | **Offsets:**
-4, -2, +2, +4 semitones |
| Low-Pass Filter | **Cutoff:**
2000, 3000, 4000, 6000 Hz | **Cutoff rule:**
spectral-energy quintiles:
50, 70, 85, 95%
**Rounding:**
nearest 100 Hz |
| High-Pass Filter | **Cutoff:**
100, 300, 500, 800 Hz | **Cutoff rule:**
spectral-energy quintiles:
5, 15, 30, 50%
**Rounding:**
nearest 100 Hz |
| Echo | **Delay:**
5-20 ms
**Gain:**
0.3-0.7 | **Delay:**
50, 100, 150 ms
**Gain:**
0.4, 0.5, 0.7 |
| Hard Clipping | **Threshold:**
0.3, 0.5, 0.7 (absolute) | **Threshold:**
$\{0.3, 0.5, 0.7\} \times A_{95}$ |
| Vibrato | **Rate:**
3, 5, 7 Hz
**Depth:**
0.001-0.003 (fractional stretch) | **Rate:**
3, 5, 7 Hz
**Depth:**
adaptive, clipped to 0.01-0.05 |

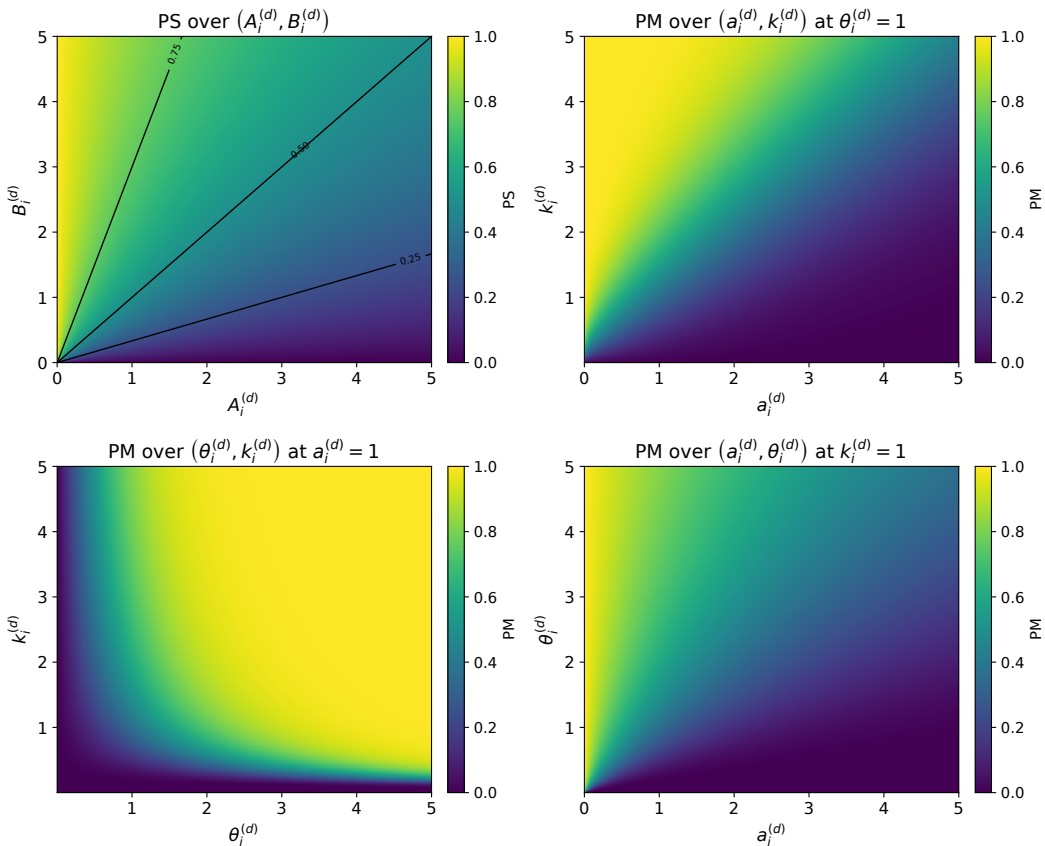

Figure 3: Functional behavior of the PS measure with $0.25, 0.5, 0.75$ contour lines, and of the PM measure in three different setups of $\theta_i^{(d)} = 1, a_i^{(d)} = 1, k_i^{(d)} = 1$.

# B   ADDITIONAL EXPREIMENTAL SETUP DETAILS

## B.1   THE PS AND PM MEASURES

The functionality of the measures is demonstrated in Figure 3 and illustrates the behavior explained in §4.2 and §4.3.

The empirical distributions of the frame-level values of the measures are shown in Figure 4. The PM and PS metrics exhibit contrasting distribution patterns. PM values cluster predominantly around zero with minimal density near one, while PS concentrate near one with virtually no occurrence near zero. Although frame-level human speech quality ratings are not publicly available for direct comparison, these patterns raise comparisons to how humans might perceive audio disturbances. The PM distribution aligns intuitively with human perception, as listeners typically penalize speech quality severely when disturbances occur, making ratings near the scale minimum unsurprising. However, real granular human ratings would likely show less extreme clustering around zero due to perceptual and rating scale complexities. The PS behavior presents a more complex interpretative challenge. Previous research suggests that humans perceive leakage as more quality-degrading than self-distortions, particularly in acoustic echo cancellation contexts (Khanagha et al., 2024), yet our findings here do not support this hypothesis. Whether this discrepancy stems from dataset characteristics, limitations of the PS measure itself, or the mismatch between granular PS values and aggregated human ratings remains unclear and warrants future investigation beyond the scope of this study.

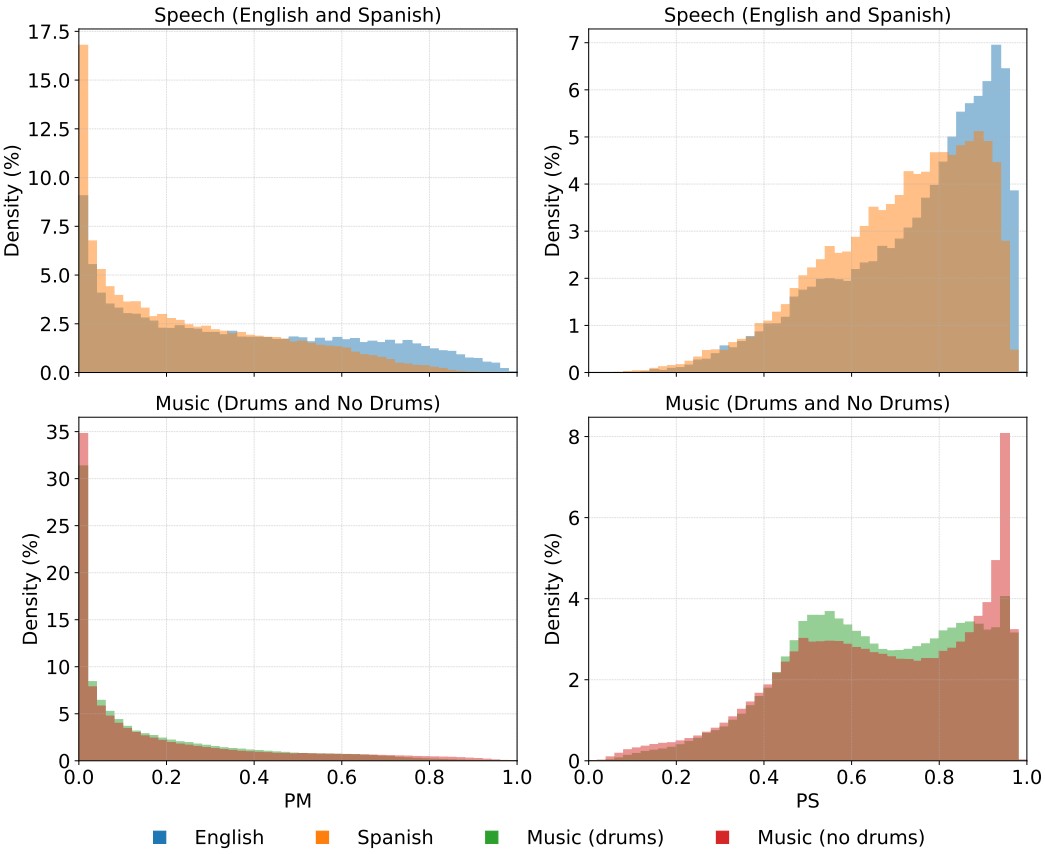

Figure 4: The distribution of PM and PS values across speech and music scenarios from the SEBASS database.

## B.2 THE SEBASS DATABASE

The SEBASS dataset suits this study for several reasons. Multilingual coverage of English and Spanish validates language-agnostic behavior, while music tests robustness to highly transient material. Large algorithmic spread creates rich output clusters that stress-test our methodology, and the dense sampling of raters allows for a more reliable estimation of the true mean-opinion score of subjective human opinion. Figure 5 shows that the speech reference signals have been recorded in a relatively clean environment with SNRs between 3.9 dB to 41.7 dB, with an average of 25 dB.

## B.3 PRE-PROCESSING

We recognize that English and Spanish speakers rarely participate in the same conversation in real-life scenarios. To emulate realistic scenarios, we separate each 4-speaker mixture into their English and Spanish speakers, creating for each language two mixtures where the one has a pair of male speakers and the other a pair of female speakers. We acknowledge the uncertainty this step induces, as residuals of English may be present in the output signal of a Spanish speaker, and vice versa. It should be mentioned that listening tests have rendered this cross-language leakage extremely negligible. This may be since, as expected, source separation systems are able to leverage languages as a meaningful feature to recognize leakage and remove it.

Every waveform, including references, distortions, and outputs from all sources of the mixture, undergoes independent loudness normalization. We use the EBU Recommendation R-128 (European Broadcasting Union (EBU), 2011) and set the target level of each waveform to loudness units relative to full scale (LUFS) of -23. If the peak magnitude of the scaled waveform exceeds one, we attenuate it to avoid digital clipping. This step removes loudness bias, known to wrongly affect both human

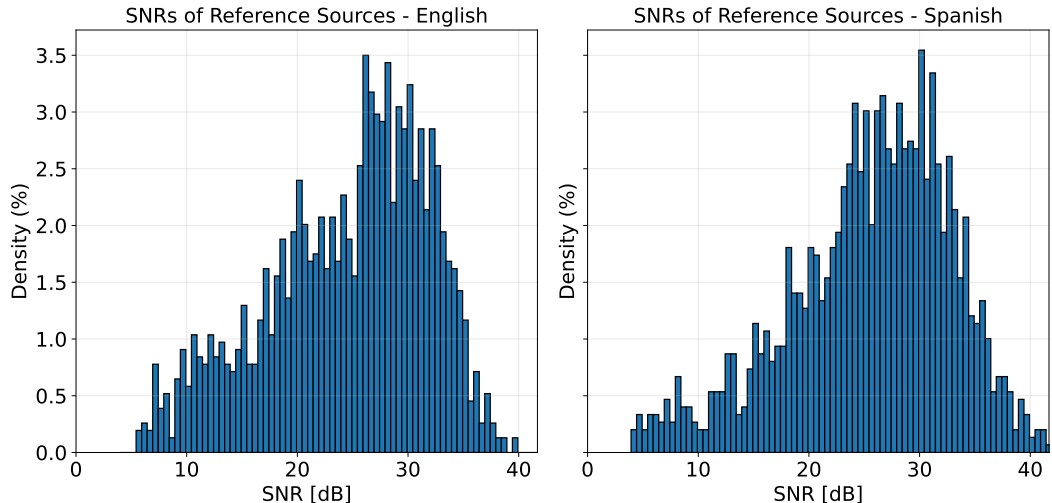

Figure 5: Frame-level SNR estimations for English and Spanish references in the SEBASS database.

and algorithmic quality judgments, while preserving inter-speaker level relations across the outputs. Since the PS and PM measures address source separation, we also filter out any frames in which there are not at least two active sources using energy-based thresholding.

When applying diffusion maps, we set $\alpha = 1$ in (3) to eliminate density-dependent bias from the embedding. This choice ensures that the PS and PM measures reflect the intrinsic geometric structure of the manifold rather than sampling density variations, which would introduce instead artificial distortions into the representation. We set $t = 1$ in (6), to keep the diffusion operator focused on local neighborhoods and not being blurred by multi-step mixing.

## B.4 FRAME-LEVEL TO UTTERANCE-LEVEL AGGREGATION OF THE MEASURES

At trial $l$, let us denote the PM value of the $i$-th output of source-separation system $q$ in time frame $f$ as $\mathrm{PM}_{i,f}^{q,l}$. Let $\mathcal{F}^l$ holds the time-frame indices with at least two active sources, and let $\mathcal{F}_i^l \in \mathcal{F}^l$ be its subset of time-frame indices in which the $i$-th source is active. Then, the utterance-level PM measure after average aggregation is given by:

$$\mathrm{PM}_{i,\mathrm{utt}}^{q,l} = \frac{1}{\left|\mathcal{F}_i^l\right|} \sum_{f \in \mathcal{F}_i^l} \mathrm{PM}_{i,f}^{q,l}. \tag{45}$$

Although average aggregation assumes that human listeners perceive global audio quality by weighing local events equally, which is evidently not the case (Rix et al., 2001), we chose to carry it for the PM since its behavior already exhibits strong and frequent granular penalties where the score drops to around zero. Thus, it is assumed that standard human behavior that weighs negative experience heavily in the utterance-level score is implicitly carried out by the nature of the PM measure itself.

However, this is not the case for the PS measure. Here, the aggregation we applied is inspired by the window-based pooling and logistic mapping used inside PESQ (Rix et al., 2001). Again, considering only time frame indices in $\mathcal{F}_i^l$ and dropping the rest, let us consider a window of size $W$ frames that slides across the PS measure with a hop size of $H$ frames. Using the $p$-norm, we define the following:

$$\ell_{i,m}^{q,l} = \left( \frac{1}{W} \sum_{w=1}^{W} \left| \mathrm{PS}_{i,(m-1)H+w}^{q,l} \right|^p \right)^{1/p}, \tag{46}$$

where $m \in \{1, \ldots, M_i^l\}$ and $M_i^l$ is the number of possible windows:

$$M_i^l = \max\left( 1, \left\lfloor \frac{\left|\mathcal{F}_i^l\right| - W}{H} \right\rfloor \right). \tag{47}$$

We then calculate the following root mean square expression:

$$\ell_i^{q,l} = \sqrt{\frac{1}{M_i^l} \sum_{m=1}^{M_i^l} \left(\ell_{i,m}^{q,l}\right)^2}, \tag{48}$$

and eventually the aggregated PS measure is given by:

$$PS_{i,\text{utt}}^{q,l} = 0.999 + \frac{4}{1 + \exp\left(-1.3669\,\ell_i^{q,l} + 3.8224\right)}, \tag{49}$$

where the constants were chosen according to (ITU-T, 2007). Here, we penalize lower scores explicitly using the $p$-norm to better match human perceptual aggregation.

## B.5 CORRELATION COEFFICIENTS BETWEEN AGGREGATED MEASURES AND MOS

At trial $l$, let the utterance-level MOS of the $i$-th output from separation system $q$ be $v_i^{q,l}$. Given $Q$ independent source separation systems such that $q \in \{1, \dots, Q\}$, consider the $Q$-dimensional vectors:

$$\mathbf{PS}_{i,\text{utt}}^l = \left(PS_{i,\text{utt}}^{1,l}, \dots, PS_{i,\text{utt}}^{Q,l}\right)^T, \tag{50}$$

$$\mathbf{PM}_{i,\text{utt}}^l = \left(PM_{i,\text{utt}}^{1,l}, \dots, PM_{i,\text{utt}}^{Q,l}\right)^T, \tag{51}$$

$$\mathbf{v}_i^l = \left(v_i^{1,l}, \dots, v_i^{Q,l}\right)^T. \tag{52}$$

The PCC (Benesty et al., 2009) is measured twice, for the PS and the PM, as follows:

$$r_i^{\text{pcc},l}\left(\mathbf{PS}_{i,\text{utt}}^l, \mathbf{v}_i^l\right) = \frac{\left(\overline{\mathbf{PS}}_{i,\text{utt}}^l\right)^T \overline{\mathbf{v}}_i^l}{\left\|\overline{\mathbf{PS}}_{i,\text{utt}}^l\right\|_2 \left\|\overline{\mathbf{v}}_i^l\right\|_2}, \tag{53}$$

$$r_i^{\text{pcc},l}\left(\mathbf{PM}_{i,\text{utt}}^l, \mathbf{v}_i^l\right) = \frac{\left(\overline{\mathbf{PM}}_{i,\text{utt}}^l\right)^T \overline{\mathbf{v}}_i^l}{\left\|\overline{\mathbf{PM}}_{i,\text{utt}}^l\right\|_2 \left\|\overline{\mathbf{v}}_i^l\right\|_2}, \tag{54}$$

where $\overline{\mathbf{PS}}_{i,\text{utt}}^l, \overline{\mathbf{PM}}_{i,\text{utt}}^l$ and $\overline{\mathbf{v}}_i^l$ are the centered versions of $\mathbf{PS}_{i,\text{utt}}^l, \mathbf{PM}_{i,\text{utt}}^l$ and $\mathbf{v}_i^l$, respectively.

Let $\mathcal{R} : \mathbb{R}^Q \to R^Q$ be the ranking operator, which in the presence of ties assigns the average ranks. The SRCC (Sedgwick, 2014) is measured for the PS and the PM:

$$\rho_i^{\text{srcc},l}\left(\mathbf{PS}_{i,\text{utt}}^l, \mathbf{v}_i^l\right) = r_i^{\text{pcc},l}\left(\mathcal{R}\left(\mathbf{PS}_{i,\text{utt}}^l\right), \mathcal{R}\left(\mathbf{v}_i^l\right)\right), \tag{55}$$

$$\rho_i^{\text{srcc},l}\left(\mathbf{PM}_{i,\text{utt}}^l, \mathbf{v}_i^l\right) = r_i^{\text{pcc},l}\left(\mathcal{R}\left(\mathbf{PM}_{i,\text{utt}}^l\right), \mathcal{R}\left(\mathbf{v}_i^l\right)\right). \tag{56}$$

We report these correlation coefficients per English, Spanish, and music mixtures scenarios separately. Let us denote $N_f^l$ the number of active sources in trial $l$ during frame $f$. Then, given a scenario with $\mathcal{L}$ independent trials such that $l \in \{1, \dots, \mathcal{L}\}$, we mark the maximal number of sources in trail $l$ with $N_{\max}^l$:

$$N_{\max}^l = \max_{f \in \mathcal{F}^l} N_f^l. \tag{57}$$

Then, for the PS and PM measures, the PCC and SRCC we report per scenario are given by:

$$PS^{pcc} = \frac{1}{\sum_{l=1}^{\mathcal{L}} N_{\max}^l} \sum_{l=1}^{\mathcal{L}} \sum_{i=1}^{N_{\max}^l} r_i^{pcc,l} \left( \mathbf{PS}_{i,\text{utt}}^l, \mathbf{v}_i^l \right), \tag{58}$$

$$PM^{pcc} = \frac{1}{\sum_{l=1}^{\mathcal{L}} N_{\max}^l} \sum_{l=1}^{\mathcal{L}} \sum_{i=1}^{N_{\max}^l} r_i^{pcc,l} \left( \mathbf{PM}_{i,\text{utt}}^l, \mathbf{v}_i^l \right), \tag{59}$$

$$PS^{srcc} = \frac{1}{\sum_{l=1}^{\mathcal{L}} N_{\max}^l} \sum_{l=1}^{\mathcal{L}} \sum_{i=1}^{N_{\max}^l} \rho_i^{srcc,l} \left( \mathbf{PS}_{i,\text{utt}}^l, \mathbf{v}_i^l \right), \tag{60}$$

$$PM^{srcc} = \frac{1}{\sum_{l=1}^{\mathcal{L}} N_{\max}^l} \sum_{l=1}^{\mathcal{L}} \sum_{i=1}^{N_{\max}^l} \rho_i^{srcc,l} \left( \mathbf{PM}_{i,\text{utt}}^l, \mathbf{v}_i^l \right). \tag{61}$$

## C  ADDITIONAL EXPERIMENTAL RESULTS

Table 4: Self-supervised architectures, their pre-trained checkpoints, scenarios, and number of transformer layers.

| Architecture | Checkpoint | Scenario | Transformer Layers |
|---|---|---|---|
| WavLM Large | microsoft/wavlm-large | English | 24 |
| WavLM Base | microsoft/wavlm-base | English | 12 |
| wav2vec 2.0 Large | facebook/wav2vec2-large-lv60 | English | 24 |
| wav2vec 2.0 Base | facebook/wav2vec2-base | English | 12 |
| HuBERT Large | facebook/hubert-large-ll60k | English | 24 |
| HuBERT Base | facebook/hubert-base-ls960 | English | 12 |
| wav2vec 2.0 Large | facebook/wav2vec2-large-xlsr-53 | Spanish | 24 |
| MERT | m-a-p/MERT-v1-95M | Music | 12 |

We begin by analyzing how performance depends on the choice of the pre-trained self-supervised model, the purpose of which is encoding waveforms into perceptual representations before they are fed into the diffusion maps. Table 4 lists the models we examine in this study. We consider six different models for English mixtures, based on the wav2vec 2.0 (Baevski et al., 2020), WavLM (Chen et al., 2022), and HuBERT (Hsu et al., 2021) backbones, with Figure 6 demonstrating their layer-wise performance. When using "Large" versions of the models, for both PCC and SRCC values, earlier layers frequently produce representations that allow superior results that gradually decline toward deeper layers, showing approximately 10% average absolute degradation between extremes. Existing layer-wise analysis already reported that acoustic and phonetic content is richly represented in intermediate layers, while deeper layers shift toward semantic abstraction (Pasad et al., 2023; Vaidya et al., 2022). Additional work confirms that distortion sensitivity peaks in the lower or middle layers and diminishes in deeper ones (Hung et al., 2022), and that pretrained models tend to lose low-level signal fidelity in their deepest layers (Moussa & Toneva, 2025). A notable data point appears in the final layers of wav2vec 2.0 with a sharp drop in performance, especially for PS. This is likely due to its contrastive learning pretraining objective, which drives later layers to specialize in predicting quantized latent codes rather than preserving acoustic detail. For the "Base" versions of the models, we observe a somewhat different behavior. At low and middle layers, their performance is often quite competitive with the "Large" variants, and in several cases the former even outperforms the latter in deeper layers. However, for WavLM, the gap widens toward the final layers, with the "Large" version consistently outperforming. Interestingly, wav2vec 2.0 Base does not exhibit the sharp degradation observed in its counterpart and instead its deeper layers remain stable and even show improvements for PS, suggesting that the absence of over-specialization to quantized prediction in the "Base" model preserves sensitivity to perceptual distortions.

Table 5 narrows these models down to their top performing layer, chosen by the max-min criteria of the PCC and SRCC values, across all layers. A no-encoding option is also reported, where

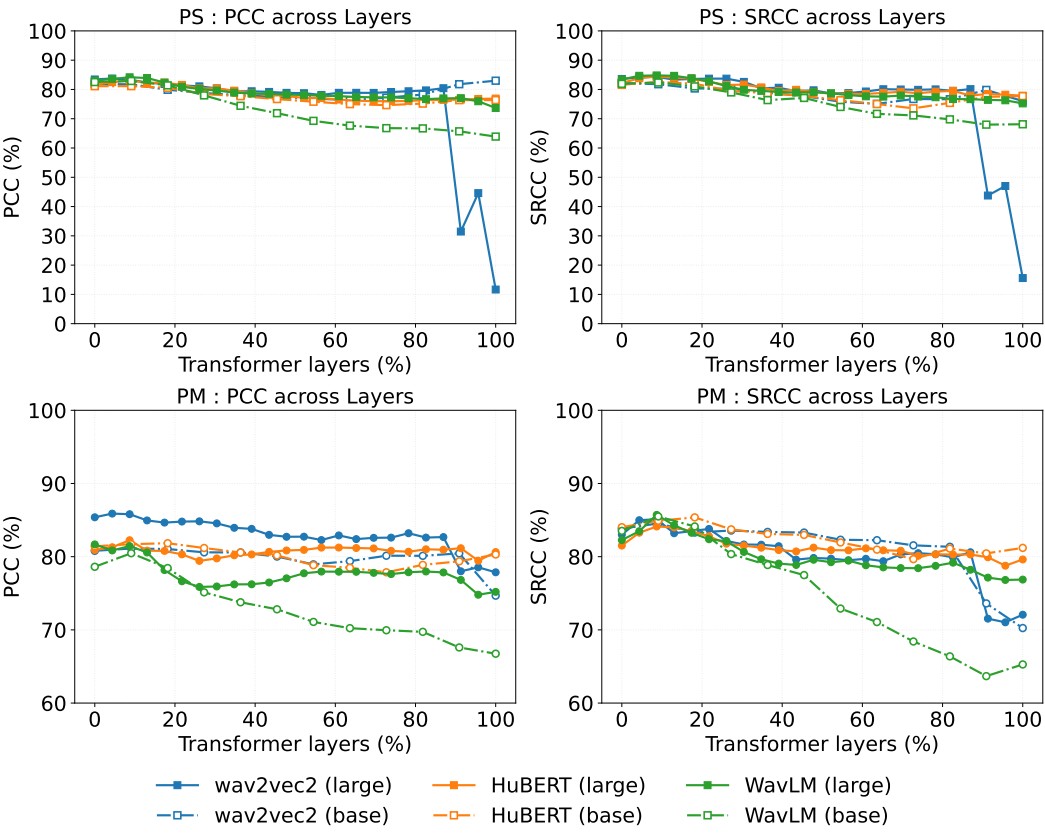

Figure 6: For English mixtures, the effect of transformer layers in different pretrained self-supervised models on the PCC and SRCC values for the PS and PM measures.

waveforms are skip-connected directly into the diffusion maps, which under-performs compared to encoded modes and emphasizes the effectiveness of the waveform encoding in the proposed pipeline. These results reaffirm that shallow layers achieve optimal performance. Although point-by-point comparisons show that "Base" models perform comparably to, or occasionally slightly exceed "Large" models, applying the max-min criteria across the models in the table reveals that 'Large' models are preferable when jointly optimizing for PS and PM. For once, wav2vec 2.0 Large achieves for the PM a PCC and SRCC differences from its "Base" counterpart of absolute 6% and 2%, respectively, even when the PS case shows a negligible gap. Among the "Large" model variants, wav2vec 2.0 Large with transformer layer 2 emerges as the ideal configuration and we carry it forward as a case study we investigate. It should be noted that among "Large" models, the PS very slightly changes with roughly 1% and 0.5% gaps between extremes for the PCC and SRCC, respectively, while the PM gaps are more meaningful. This suggests that the choice of model may mainly affect the PM scores.

We also investigate into how the SRCC and PCC deterministic and probabilistic error bounds behave under change of models, as shown in Table 6. Three main observations emerge:

(i) Deterministic errors are uniformly small across all models. The deterministic radii remain small for every model (less than 0.2% for SRCC and 0.21% for PCC under PS, and less than 0.26% for SRCC and 1.38% for PCC under PM). This sustain the observation made in the paper, which that the deterministic uncertainty around the PS and PM correlations is small enough so that the competitive ranking of the measures is sustained and is not driven by unstable estimates.

(ii) Uncertainty is highly comparable across models. In the PS case, the CIs for SRCC vary only from 26.47-30.03% and for PCC from 9.10-10.29% across all models. Similarly, in the PM case, the CIs are in a narrow band for most models (e.g., SRCC is 7.23-11.33% for five of the six models, PCC is 3.83-5.36% for four of the six). No family of models out of wav2vec2, hubert, and wavlm

| Measure | Representation | Transformer Layer | SRCC | PCC |
|---------|----------------|-------------------|------|-----|
| PS | wav2vec 2.0 (Large) | 2 | 84.12% | 83.74% |
| PS | wav2vec 2.0 (Base) | 2 | 84.25% | 83.23% |
| PM | wav2vec 2.0 (Large) | 2 | 84.69% | **86.36%** |
| PM | wav2vec 2.0 (Base) | 2 | 82.79% | 80.07% |
| PS | WavLM (Large) | 3 | 84.80% | 84.16% |
| PS | WavLM (Base) | 2 | **84.84%** | **84.19%** |
| PM | WavLM (Large) | 3 | **85.71%** | 81.44% |
| PM | WavLM (Base) | 2 | 82.82% | 77.51% |
| PS | HuBERT (Large) | 3 | 84.48% | 83.09% |
| PS | HuBERT (Base) | 2 | 84.83% | 82.73% |
| PM | HuBERT (Large) | 3 | 84.12% | 82.24% |
| PM | HuBERT (Base) | 2 | 81.37% | 79.47% |
| PS | Waveform (raw) | - | 73.42% | 71.04% |
| PM | Waveform (raw) | - | 69.30% | 66.62% |

Table 5: For English mixtures, comparing PCC and SRCC values between best-layer performance of "Large" and "Base" models. A raw waveform option, i.e. no encoding, is also reported. The highest SRCC and PCC are in bold per PS and PM.

| Representation | PS Radius | PS CIs (95%) | PM Radius | PM CIs (95%) |
|----------------|-----------|--------------|-----------|--------------|
| wav2vec 2.0 (Large) | 0.16 / 0.21% | 30.03 / 10.29% | 0.11 / 0.99% | 7.23 / 3.83% |
| wav2vec 2.0 (Base) | 0.16 / 0.16% | 28.35 / 9.91% | 0.01 / 1.10% | 11.33 / 5.36% |
| HuBERT (Large) | 0.07 / 0.19% | 26.47 / 9.36% | 0.15 / 1.38% | 8.44 / 4.30% |
| HuBERT (Base) | 0.20 / 0.16% | 28.34 / 9.71% | 0.07 / 1.24% | 11.08 / 5.16% |
| WavLM (Large) | 0.09 / 0.20% | 26.94 / 9.41% | 0.26 / 1.24% | 9.99 / 4.81% |
| WavLM (Base) | 0.00 / 0.10% | 26.66 / 9.10% | 0.25 / 1.38% | 16.13 / 7.22% |

Table 6: PS and PM radius values and corresponding 95% confidence intervals for different self-supervised representations (Large vs. Base). Values are reported as SRCC / PCC %.

systematically exhibits larger error bars, indicating that the different front-ends are estimated with essentially the same level of reliability.

(iii) Larger variants tend to be slightly more stable than their base counterparts. Within each family of models, the "large" variants have consistently smaller PM CIs than the corresponding base model (e.g., for CIs in the PM case - wav2vec2 large gives 7.23/3.83% and wav2vec2 base gives 11.33/5.36%, and similar patterns hold for hubert and wavlm). This suggests that increased model capacity improves stability rather than inflating uncertainty.

Next, we analyze all scenarios with wav2vec 2.0 "Large" encoders for speech and the MERT encoder for music representations, and analyze the effect of their transformer layer on performance. The results are shown in Figure 7. English and Spanish mixtures, both evaluated with wav2vec 2.0 backbones, show broadly similar trends across layers, with Spanish exhibiting a sharper decline in deeper layers. This can be explained by the XLSR pretraining data being relatively scarcer in Spanish than in English, leading later layers to emphasize cross-lingual abstractions over fine acoustic detail (Conneau et al., 2021). Music mixtures with drums show the lowest performance among scenarios, which we attribute to the dominance of strong percussive transients. Self-supervised models have demonstrated less stability in these highly non-stationary regions, reducing the ability of PS and PM to capture perceptual degradations (Zeghidour et al., 2021). In contrast, music mixtures without drums demonstrate consistently high performance, in most layers even surpassing speech mixtures. This likely stems from the MERT backbone being particularly suited in capturing harmonic and timbral structure, allowing the measures to remain faithful to perceptual cues such as instrument texture and vocal clarity (Li et al., 2024). Interestingly, whether drums are present or not, MERT-based performance demonstrates a steady behavior across all layers, suggesting the MERT representations are not vulnerable to degradation across processing stages. The max-min criteria across all layers, per scenario, shows that the ideal layers for English, Spanish, drums, and no-drums music mixtures are layers 2, 2, 1, and 3, respectively.

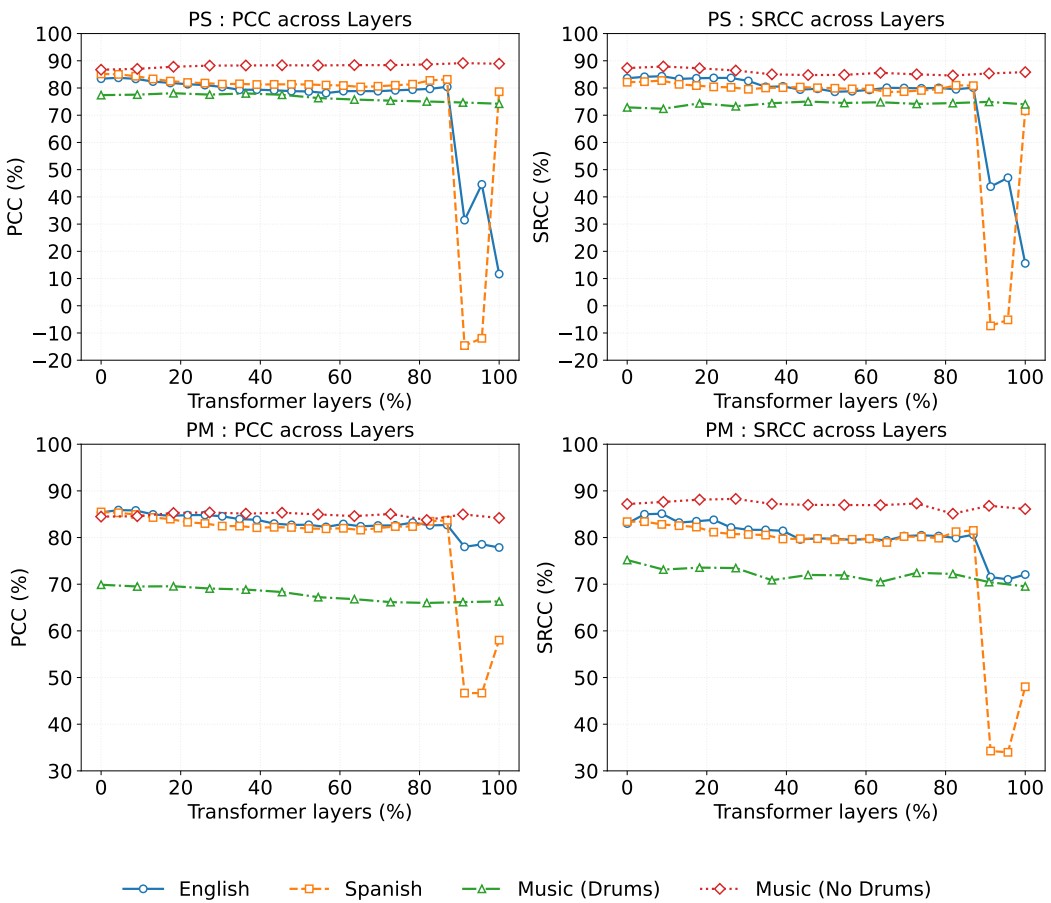

Figure 7: For all scenarios, the effect of transformer layers in their respective pretrained self-supervised architectures on the PCC and SRCC values for the PS and PM measures.

We employed these layers to construct Table 1 in the main text and now we extend the discussion on it. The advantage of PESQ can be attributed to its long-standing perceptual model, which explicitly encodes aspects of loudness perception, asymmetry, and time-alignment penalties, features that directly penalize separation artifacts. In Spanish mixtures, the PS and PM are most performant in terms of PCC, but fall behind PESQ and SDR-based metrics in SRCC. One possible explanation is that the syllable-timed rhythm and steady vowels of Spanish make fidelity-driven metrics such as SI-SDR, CI-SDR, and SDR more predictive of listener rankings, as these metrics emphasize reconstruction accuracy at the waveform level. For music mixtures, PS and PM achieve the strongest overall correlations across both drums and no-drums conditions for both PCC and SRCC. Even though SpeechBERTscore has shown impressive results and is also based on a self-supervised backbone, it is mostly not competitive with our measures, and notably even performing worse than our raw waveform version at times, which projects on the importance of the diffusion maps in the pipeline. We emphasize that unlike English, we only inspected one backbone model for Spanish or music mixtures. In addition, the aggregation strategies we applied were not data-driven but a heuristic and reasoning-based choice. Consequently, while the proposed measures already demonstrate strong alignment with human perception, these low-hanging fruits may potentially boost performance. Quite surprisingly, the first group of STOI, PESQ, and SDR-based measures is consistently preferable to the second group consisting of DNSMOS, speechBERTscore, UTMOS, and others, which rarely achieve more than 70% in performance. One crucial conclusion this table suggests is that measures originally developed for a certain audio application, should not be zero-shot adapted into other applications, and in that case into source separation evaluation. Otherwise, values that drift from human opinion may be reported, which may spiral the development of audio technologies instead of accelerating it.

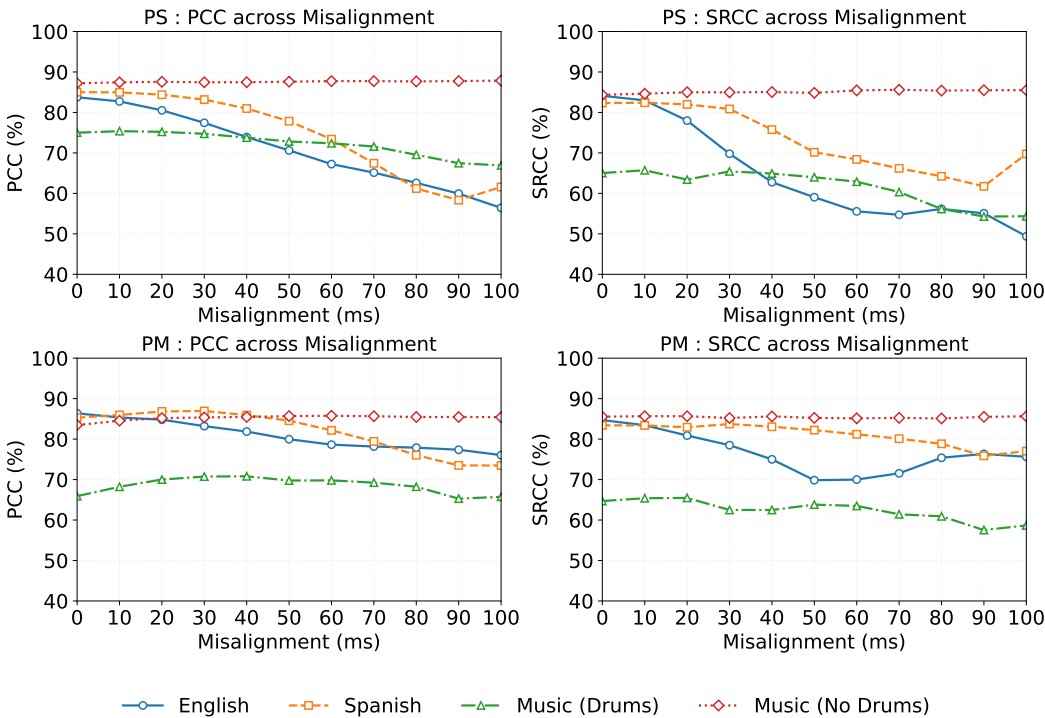

Figure 8: The effect of temporal misalignment between references and outputs of the separation system on the PS and PM measures.

An additional stress test for our measures concerns their robustness to temporal misalignment between the input and output streams of the separator, a phenomenon commonly introduced by modern communication systems or, e.g., when dealing with references obtained from different, per-speaker microphones, such as in meeting datasets (Carletta et al., 2005; Vinnikov et al., 2024). Figure 8 illustrates the effect of artificial delays ranging from 0 ms to 100 ms across English, Spanish, and music scenarios. While performance gradually degrades for speech scenarios as misalignment grows, as expected, a 20 ms delay or less still preserves coefficients higher than 80%. Surpassing this threshold, however, often causes a pronounced drop that underscores this weakness in our measures, since human ratings are insensitive to these short latencies. Music mixtures exhibit a different pattern, as performance remains largely stable across delays, with the presence of drums introducing more variability than its counterpart.

Table 7 provides complementary information to the NMI results in Figure 2 but listing the frame counts used for the PS and PM measures for every examined threshold. Even the lowest threshold of 0.1, had a minimum of 481 for calculations, rendering its results statistically reliable. An interesting observation in the music scenario, without drums, is that it exhibits significantly more time frames in which there are at least two active sources, compared to all other scenarios.

In the next phase, we investigate the deterministic error radius and probabilistic CIs derived for the PS and PM measures in Appendix E. Figure 9 shows histograms of the frame-level error distributions for speech and music mixtures. As expected, the radius caused by the spectral truncation in the diffusion maps process is typically an order of magnitude smaller than the 95% probabilistic width, which is originated from finite-sample clusters on the manifold. The error radius is also concentrated mostly near zero, which further confirms its negligibility. CIs typically span 10-50% of the dynamic range of the measures at the frame level, but surprisingly in the PM, between 10-15% of frame-level instances have probabilistic tails that approach zero across scenarios. The immediate contribution of these results are by making development of source separation systems more reliable and informed at the frame-level.

| Threshold | English | | Spanish | | Music (Drums) | | Music (No Drums) | |
|---|---|---|---|---|---|---|---|---|
| | PS≤th | PM≤th | PS≤th | PM≤th | PS≤th | PM≤th | PS≤th | PM≤th |
| 0.1 | 583 | 7426 | 622 | 10721 | 481 | 13987 | 622 | 22859 |
| 0.2 | 1546 | 12086 | 1591 | 15756 | 1536 | 16126 | 1832 | 28828 |
| 0.3 | 3191 | 16350 | 3350 | 19714 | 3327 | 17846 | 4226 | 33231 |
| 0.4 | 5753 | 20118 | 6091 | 23054 | 5861 | 19492 | 8821 | 36953 |
| 0.5 | 9115 | 23697 | 9725 | 25964 | 8927 | 21033 | 15748 | 40426 |
| 0.6 | 13364 | 27232 | 13904 | 28627 | 12037 | 22522 | 24958 | 43769 |
| 0.7 | 18465 | 30758 | 18592 | 30885 | 15373 | 23864 | 35434 | 47186 |
| 0.8 | 24477 | 34076 | 23871 | 32703 | 19498 | 25168 | 46078 | 50682 |
| 0.9 | 31572 | 36507 | 29589 | 33902 | 25748 | 26614 | 57795 | 54939 |
| 1.0 | 37888 | 37888 | 34496 | 34496 | 34688 | 34688 | 66528 | 66528 |

Table 7: Frame counts used for NMI computation at each threshold, denoted 'th' in the table. Columns show counts of frames per scenario, split by PS and PM subsets.

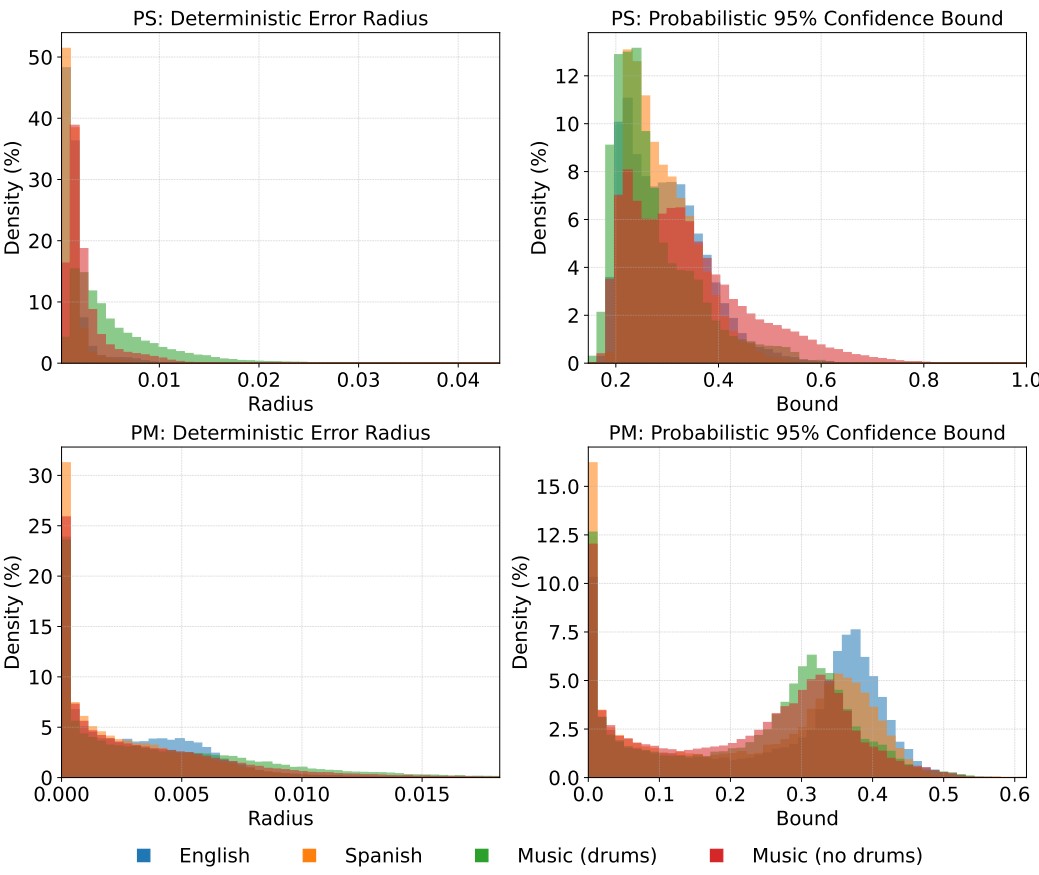

Figure 9: An histogram view of the frame-level deterministic error radius and the 95% probabilistic tail in the PS and PM measures across scenarios.

For illustration, Figure 10 shows reference and output spectrograms from an English mixture, time-aligned to corresponding PM and PS values over a 10-second utterance using the "Large" models, with layers specified in Table 5. While a single example cannot be over-interpreted, the latest observation about the PS gaps across layers is visually supported here, with very similar behavior of all models. The PM shows highly correlated behavior, but with noticeable different values by wav2vec 2.0, which exhibits the highest PM value for PCC. An interesting visual example is shown at approximately the 9 seconds mark, when both speakers exhibit visible self-distortion artifacts

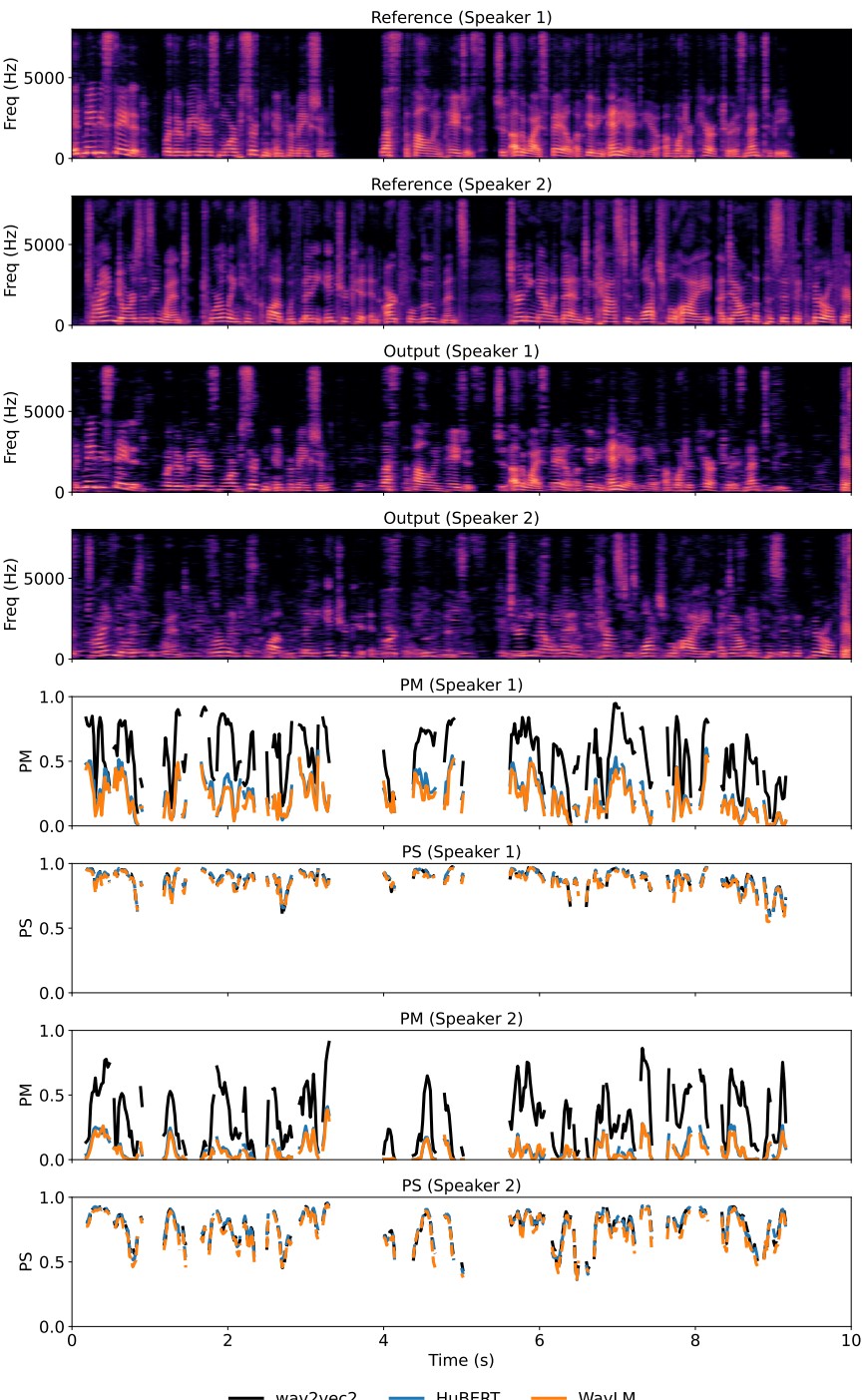

Figure 10: For an English mixture with two speakers, a spectral view of the system signals and aligned with a time-series view of the PS and PM measures of each speaker across different self-supervised architectures. Blank time intervals remain whenever speech does not overlap.

accompanied by sharp drops in their PM measures. Listening tests confirmed that leakage is indeed more present in "Speaker 2" than in "Speaker 1", as supported by the PS plot.

Finally, to give the reader an intuitive grasp of how the two error terms evolve in a time-aligned manner with the PS and PM measures, Figure 11 illustrates a representative example.

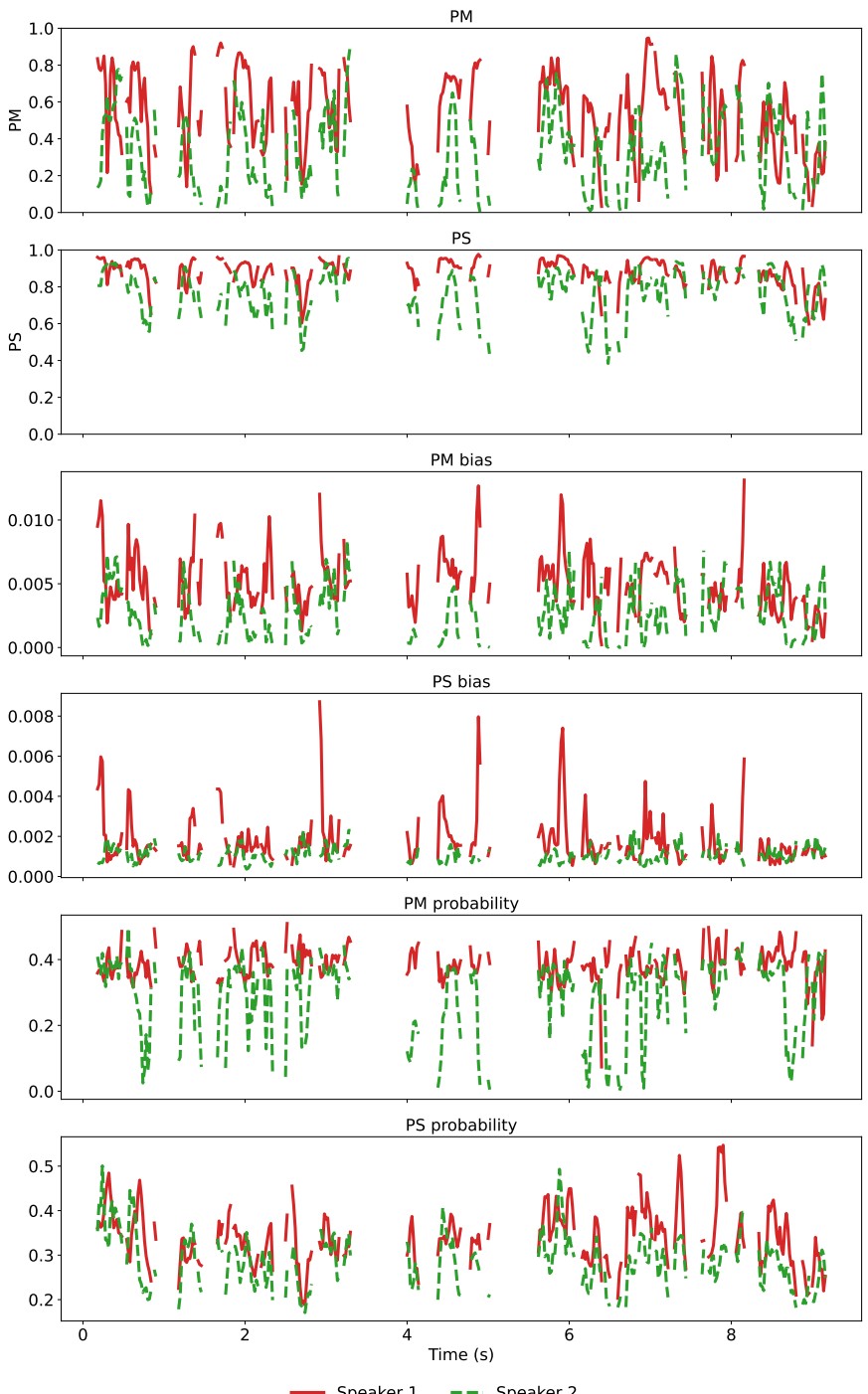

Figure 11: Time-aligned view of the PM and PS measures and their deterministic error radius and probabilistic tail with 95%, of two English speakers. Time indices where speech does not overlap remain blank.

| PM vs. PESQ - Temporal Misalignment (ms) | | | | | | | | | | | |
|---|---|---|---|---|---|---|---|---|---|---|---|
| Lang. | Metric | 0 | 10 | 20 | 30 | 40 | 50 | 60 | 70 | 80 | 90 | 100 |
| EN | PM | 1.00 | 0.91 | 0.84 | 0.81 | 0.80 | 0.79 | 0.79 | 0.79 | 0.79 | 0.79 | 0.78 |
| EN | PESQ | 4.64 | 4.49 | 4.43 | 4.40 | 4.41 | 4.36 | 4.35 | 4.33 | 4.33 | 4.32 | 4.29 |
| SP | PM | 1.00 | 0.93 | 0.89 | 0.87 | 0.86 | 0.86 | 0.86 | 0.86 | 0.86 | 0.86 | 0.86 |
| SP | PESQ | 4.64 | 4.59 | 4.50 | 4.47 | 4.46 | 4.39 | 4.37 | 4.33 | 4.31 | 4.27 | 4.25 |

| PS vs. PESQ - Packet Loss (%) | | | | | | | | | | | |
|---|---|---|---|---|---|---|---|---|---|---|---|
| Lang. | Metric | 0 | 2.2 | 4.4 | 6.6 | 8.8 | 11.11 | 13.33 | 15.5 | 17.8 | 20 | |
| EN | PS | 1.00 | 0.93 | 0.94 | 0.93 | 0.93 | 0.93 | 0.97 | 0.96 | 0.95 | 0.94 | |
| EN | PESQ | 4.64 | 2.45 | 1.39 | 1.17 | 1.14 | 1.08 | 1.07 | 1.05 | 1.04 | 1.04 | |
| SP | PS | 1.00 | 0.93 | 0.94 | 0.93 | 0.93 | 0.93 | 0.97 | 0.96 | 0.95 | 0.94 | |
| SP | PESQ | 4.64 | 2.91 | 1.53 | 1.22 | 1.17 | 1.08 | 1.05 | 1.04 | 1.04 | 1.04 | |

Table 8: PM and PS behavior compared to PESQ under temporal misalignment and packet loss, for English (EN) and Spanish (ES) trials.

We now compare PS and PM to PESQ - our strongest comparator as of Table 1. PS and PM operate on diffusion-map distances computed from self-supervised embeddings, using a bank of synthetic distortions around each reference. They are explicitly designed to (i) disentangle leakage from self-distortion, and (ii) function at the frame level. PESQ, in contrast, produces a single utterance-level MOS-aligned score using engineered auditory features and an internal time-alignment stage (Rix et al., 2001). As shown in our misalignment stress-test in Fig 8, even moderate delays (20 ms or more) between reference and output reduce PM because it compares frame-synchronous embeddings without an alignment stage. PESQ, in contrast, performs time alignment and is largely insensitive to such shifts. Thus, a separator that outputs a clean signal with 30-100 ms latency, common in practice and imperceptible to listeners, receives high MOS and high PESQ, but PM drops because each frame is matched to the wrong reference frame, inflating Mahalanobis distances. Our experiment models this exactly: a pure time-shifted reference with no added leakage or distortion, consistent with real-time communication standards that allow up to 40 ms latency. PESQ is explicitly built for telephony/VoIP and includes mechanisms for packet loss, time alignment, and "bad-frame" handling. PS, by design, targets continuous distortions and does not model deletions, insertions, or non-uniform time warping. In our test, the system output is generated by randomly deleting 20 ms frames from the reference and then time-compressing and resampling to the original length, without any leakage or distortion.

In Table 8, we report average PM and PESQ values for delays from 0-100 ms, and average PS and PESQ values for packet-loss rates from 0-20%, averaged across all trials in each SEBASS speech scenario (English and Spanish). Under misalignment, PESQ decreases modestly from 4.64 to 4.29 (English) and 4.25 (Spanish) at 100 ms-only 5% of its dynamic range. PM, bounded in $[0, 1]$, degrades far more: 22% (English) and 14% (Spanish), showing substantially higher sensitivity to timing shifts, whereas PESQ remains closer to human judgments. For packet loss, PESQ collapses by more than half its dynamic range even below 5% loss and approaches 1 at 20% loss, aligning with the severe perceptual degradation reported by listeners. PS, by contrast, changes only modestly, indicating that it does not fully capture the perceptual impact of packet loss.

We now ask: How well does MAPSS generalize when one distortion group is entirely left out, and another is augmented with an excessively strong out of distribution variant? We therefore conducted the following experiment. MAPSS uses 13 distortion groups, each with several parameterized variants - about 70 total distortions per reference. For each of the 13 groups, we repeat the same two-step procedure: (i) remove the entire group from the distortion bank, and (ii) augment a different remaining group with one new, deliberately excessive parameter whose effect is perceptually far from the reference. For example, we omit the "tremolo" group and introduce an additional high-severity noise parameter (e.g., additive noise at -20 dB SNR) into the "noise" group. In Table 9, each row specifies which group was omitted ("O.") and which group received the new excessive parameter ("E."). As in the main paper, each entry reports the SRCC/PCC values (in %) obtained by averaging the aggregated PS and PM scores across all trials for that scenario.

| Scenario | PM (SRCC / PCC %) | | | |
|---|---|---|---|---|
| | English | Spanish | Music w/ Drums | Music w/o Drums |
| Original | 84.69/86.36 | 83.41/85.30 | 75.18/69.88 | 88.12/85.26 |
| O. notch, E. comb | 82.13/81.62 | 79.29/80.16 | 74.57/69.83 | 88.59/85.26 |
| O. comb, E. tremolo | 81.75/81.11 | 79.09/79.61 | 74.89/69.92 | 88.47/84.98 |
| O. tremolo, E. noise | 81.57/80.85 | 78.82/78.89 | 74.34/68.34 | 88.27/84.60 |
| O. noise, E. harmonic | 82.20/77.83 | 80.14/76.66 | 75.06/67.40 | 90.48/80.65 |
| O. harmonic, E. reverb | 81.72/81.27 | 79.08/79.75 | 75.16/70.05 | 88.81/85.05 |
| O. reverb, E. noisegate | 82.18/81.74 | 79.45/80.32 | 75.06/69.88 | 88.50/85.27 |
| O. noisegate, E. pitch | 82.13/81.64 | 79.13/79.69 | 74.97/69.77 | 88.48/85.06 |
| O. pitch, E. lowpass | 81.66/81.20 | 79.33/79.38 | 75.05/70.06 | 88.44/84.70 |
| O. lowpass, E. highpass | 81.46/80.54 | 78.06/78.73 | 75.00/69.79 | 88.82/85.19 |
| O. highpass, E. echo | 81.44/81.02 | 79.05/78.45 | 76.22/70.05 | 88.59/85.00 |
| O. echo, E. clipping | 82.19/81.74 | 79.47/80.31 | 75.47/69.94 | 88.40/85.26 |
| O. clipping, E. vibrato | 82.04/81.53 | 79.32/80.00 | 75.14/69.85 | 88.47/85.13 |
| O. vibrato, E. notch | 81.84/81.43 | 79.34/79.86 | 75.29/69.89 | 88.54/85.02 |

| Scenario | PS (SRCC / PCC %) | | | |
|---|---|---|---|---|
| | English | Spanish | Music w/ Drums | Music w/o Drums |
| Original | 84.12/83.74 | 82.33/85.01 | 72.87/77.38 | 87.23/87.81 |
| O. notch, E. comb | 84.09/83.52 | 82.67/85.08 | 70.65/76.38 | 85.56/86.73 |
| O. comb, E. tremolo | 84.27/83.52 | 82.45/85.09 | 70.86/76.62 | 86.02/86.94 |
| O. tremolo, E. noise | 84.30/83.41 | 82.52/84.97 | 68.96/75.68 | 85.81/86.87 |
| O. noise, E. harmonic | 84.51/83.95 | 83.49/85.08 | 72.12/77.41 | 87.17/87.97 |
| O. harmonic, E. reverb | 84.23/83.56 | 82.60/85.13 | 69.80/76.60 | 85.89/87.05 |
| O. reverb, E. noisegate | 83.95/83.34 | 82.51/84.98 | 70.20/76.43 | 85.68/86.69 |
| O. noisegate, E. pitch | 84.38/84.43 | 82.81/85.74 | 70.54/76.63 | 86.05/86.82 |
| O. pitch, E. lowpass | 82.64/82.32 | 81.22/83.70 | 70.87/76.61 | 85.52/86.98 |
| O. lowpass, E. highpass | 84.34/83.55 | 82.85/85.04 | 70.76/76.56 | 85.48/86.77 |
| O. highpass, E. echo | 84.58/83.49 | 82.52/85.21 | 71.39/76.24 | 86.01/87.01 |
| O. echo, E. clipping | 84.20/83.46 | 82.33/85.02 | 70.50/76.40 | 85.70/86.68 |
| O. clipping, E. vibrato | 84.32/83.50 | 82.43/85.04 | 70.66/76.50 | 85.53/86.81 |
| O. vibrato, E. notch | 84.11/83.48 | 82.60/85.05 | 70.16/76.50 | 85.91/86.77 |

Table 9: PM and PS performance (SRCC/PCC) across all distortion pairs and content types.

This experiment directly probes the generalization of MAPSS when the distortion bank itself is made strongly out-of-distribution with respect to the artifacts produced by the separation systems. For each of the 13 distortion groups in the paper, we (i) remove that entire family from the bank, so MAPSS never "sees" this type of distortion when constructing its manifold, and (ii) add an excessively strong parameter to a different family. The SEBASS mixtures, separator outputs, and human scores remain fixed and only the perceptual model used by PS and PM is perturbed. Deviations in SRCC/PCC therefore quantify how much MAPSS relies on a correctly specified distortion bank versus how well it extrapolates to unseen or mis-calibrated distortions. Across all 13 omit/extend configurations and four SEBASS scenarios, PS is well-generalized. Relative to the original configuration, PS SRCC/PCC vary by at most 1.5/1.4 points for English and Spanish speech, 3.9/1.7 points for music with drums, and 2/1 points for music without drums. In all cases, PS keeps the same qualitative ranking against competing metrics as in Table 1 of the paper: it remains among the best-performing measures in terms of correlation with human mean opinion score. This indicates that the separation score generalizes well even when the manifold is built from a substantially misspecified distortion bank and when one distortion family is pushed far beyond the severity range used in the original setup. PM shows a clearer limit to out of distribution generalization, which is expected because it explicitly models the empirical distribution of self-distortions within each cluster. When we remove a perceptually dominant family for speech, most notably additive noise, and simultaneously inject a very severe parameter into another family, the largest degradation is about 3-5 SRCC points and 8-9 PCC points

for English and Spanish. For other omit/extend pairs the effect is smaller (typically 1-4 points), and for music PM is essentially stable: with drums, SRCC changes by less than 1 point and PCC by 2.5 points, and without drums, SRCC never decreases relative to the original configuration and only one condition reduces PCC by 4.6 points. These results suggest that PM extrapolates gracefully to much stronger distortions within the families represented in the bank, but its performance degrades when the bank completely omits a distortion family that strongly influences human ratings in that domain (e.g., noise in speech). We argue that the main practical limit we observe is coverage of perceptually salient distortion types, not sensitivity to the exact parameter ranges.

| $\alpha$ | English | Spanish | Music no drums | Music drums |
|---|---|---|---|---|
| 0 | 0.96/1.00 | 0.96/1.00 | 0.96/1.00 | 0.94/1.00 |
| 0.11 | 0.91/0.91 | 0.88/0.90 | 0.70/0.83 | 0.85/0.90 |
| 0.22 | 0.83/0.88 | 0.79/0.87 | 0.61/0.83 | 0.76/0.87 |
| 0.33 | 0.76/0.87 | 0.72/0.85 | 0.55/0.83 | 0.70/0.85 |
| 0.44 | 0.69/0.85 | 0.65/0.84 | 0.50/0.83 | 0.65/0.85 |
| 0.55 | 0.64/0.85 | 0.59/0.84 | 0.45/0.83 | 0.61/0.84 |
| 0.66 | 0.60/0.84 | 0.54/0.83 | 0.42/0.83 | 0.58/0.84 |
| 0.77 | 0.56/0.84 | 0.50/0.83 | 0.39/0.83 | 0.55/0.83 |
| 0.88 | 0.53/0.84 | 0.47/0.83 | 0.36/0.83 | 0.53/0.83 |
| 1 | 0.50/0.83 | 0.44/0.83 | 0.34/0.83 | 0.51/0.83 |

| $\lambda$ | English | Spanish | Music no drums | Music drums |
|---|---|---|---|---|
| 0 | 1.00/0.99 | 1.00/1.00 | 1.00/1.00 | 1.00/0.98 |
| 0.11 | 1.00/0.94 | 1.00/0.95 | 1.00/0.98 | 1.00/0.91 |
| 0.22 | 1.00/0.90 | 1.00/0.90 | 1.00/0.96 | 1.00/0.86 |
| 0.33 | 1.00/0.85 | 1.00/0.84 | 1.00/0.94 | 1.00/0.79 |
| 0.44 | 1.00/0.79 | 1.00/0.78 | 1.00/0.91 | 1.00/0.72 |
| 0.55 | 1.00/0.73 | 1.00/0.71 | 1.00/0.86 | 1.00/0.63 |
| 0.66 | 1.00/0.66 | 1.00/0.64 | 1.00/0.79 | 1.00/0.54 |
| 0.77 | 0.99/0.58 | 0.99/0.56 | 0.99/0.69 | 0.99/0.42 |
| 0.88 | 0.99/0.48 | 0.99/0.46 | 0.99/0.56 | 0.99/0.30 |
| 1 | 0.98/0.36 | 0.98/0.34 | 0.98/0.40 | 0.98/0.17 |

Table 10: Sensitivity analysis over $\alpha$ and $\lambda$ parameters (SRCC/PCC in %).

By design, PM only measures proximity to the target reference and its distortion cloud, while PS also compares to all non-attributed references. To show that this construction disentangles leakage from self-distortion, we include two new controlled experiments where leakage and self-distortion vary independently. For each mixture $z = y_1 + y_2$, we construct outputs $\hat{y}_1(\alpha) = y_1 + \alpha y_2$ for $\alpha \in [0, 1]$. Here only leakage increases with $\alpha$. In a second experiment, we select a new distortion not present in the MAPSS distortion bank and apply it to $y_1$ with continuously increasing strength $\lambda \in [0, 1]$. Here only self-distortion increases with $\lambda$. Table 10 provides a compact summary table. Each cell is the aggregated PS/PM, averaged across trials in each scenario, over the corresponding $\alpha$ or $\lambda$ values. We observe that PS responds selectively to leakage and PM responds selectively to self-distortion, with minor false reaction to leakage.

| | English Only | Spanish Only | Multilingual |
|---|---|---|---|
| PS (SRCC / PCC %) | 84.12/83.74 | 82.33/85.01 | 86.27/85.60 |

| | English Only | Spanish Only | Multilingual |
|---|---|---|---|
| PS (SRCC / PCC %) | 84.69/86.36 | 83.41/85.30 | 84.41/84.07 |

Table 11: Comparison between monolingual and multilingual construction of the MAPSS pipeline for PS and PM (SRCC/PCC in %).

Real-world scenarios are often multilingual and that purely monolingual experiments limit MAPSS's practical relevance. To address this, we evaluated MAPSS on multilingual mixtures from SEBASS: original 4-speaker mixtures in two configurations, (i) four female speakers (two English, two Spanish) and (ii) four male speakers (two English, two Spanish). Complementing the monolingual results of Table 1, we now report Table 11. PS slightly improves in the multilingual case, likely because clusters are formed over all sources, yielding a richer manifold on which to assess target-interference relations (e.g., fine-grained confusions between voices and languages). PM remains roughly unchanged, as it depends only on the target cluster and not on cross-cluster geometry. Together, these results suggest that with respect to the SEBASS database, MAPSS is at least as stable, and in PS's case, slightly more informative, in multilingual mixtures than in monolingual ones.

# D EXPECTATION AND PROBABILISTIC CONFIDENCE BOUND OF THE TRUNCATION ERROR

Truncating the spectrum to $d$ dimensions breaks the equality in Equation (9), and leads to a truncation error. Here, we derive the expectation and probabilistic tail bound for this truncation error. Assume a point $\mathbf{x}_i \in \mathcal{X}$ is drawn from the stationary distribution $\boldsymbol{\pi}$ 7 of the diffusion process, where $i \in \{1, \ldots, N\}$. This assumption is supported by (Hein et al., 2005, Lem. 1), and by showing empirically on $5,000$ graphs that the corresponding eigenvector matches the theoretical stationary distribution up to statistical fluctuations. Given that the $N-1$-dimensional embedding of $\mathbf{x}_i$ is truncated to dimension $d$, then the truncation error is expressed as (10):

$$E(\mathbf{x}_i) = \left( \sum_{\ell=d+1}^{N-1} \lambda_\ell^{2t} \mathbf{u}_\ell^2(i) \right)^{1/2}. \tag{62}$$

We define the squared truncation error and analyze it:

$$T(\mathbf{x}_i) = E^2(\mathbf{x}_i) = \sum_{\ell=d+1}^{N-1} \lambda_\ell^{2t} \mathbf{u}_\ell^2(i). \tag{63}$$

Since the eigenvectors $\{\mathbf{u}_\ell\}_{\ell=0}^{N-1}$ are orthonormal under $\boldsymbol{\pi}$, then:

$$\mathbb{E}_{\boldsymbol{\pi}} \left[ \mathbf{u}_\ell^2(i) \right] = \sum_{i=1}^{N} \boldsymbol{\pi}_i \mathbf{u}_\ell^2(i) = 1, \tag{64}$$

from which we derive the expectation of $T(\mathbf{x}_i)$ under $\boldsymbol{\pi}$:

$$\mathbb{E}_{\boldsymbol{\pi}} \left[ T(\mathbf{x}_i) \right] = \mathbb{E}_{\boldsymbol{\pi}} \left( \sum_{\ell=d+1}^{N-1} \lambda_\ell^{2t} u_\ell^2(i) \right) = \sum_{\ell=d+1}^{N-1} \lambda_\ell^{2t}. \tag{65}$$

Thus, the expectation of the truncation error is given directly by:

$$\mathbb{E}_{\boldsymbol{\pi}} \left[ E(\mathbf{x}_i) \right] = \left( \sum_{\ell=d+1}^{N-1} \lambda_\ell^{2t} \right)^{1/2}. \tag{66}$$

This term decays monotonically as $d$ grows and is typically lower than $10^{-3}$. To obtain a non-asymptotic and high-probability confidence bound on the truncation error, we derive $\forall \ell, i$ (64):

$$\left| \mathbf{u}_\ell(i) \right| \leq \pi_{\min}^{-1/2}, \quad \pi_{\min} = \min_{i \in \{1, \ldots, N\}} \boldsymbol{\pi}. \tag{67}$$

Any bounded variable is sub-Gaussian, and its $\psi_2$-norm is at most the bound divided by $\sqrt{\ln 2}$ (Vershynin, 2024, Example 2.6.5):

$$\|\mathbf{u}_\ell(i)\|_{\psi_2, \boldsymbol{\pi}} \leq \frac{\pi_{\min}^{-1/2}}{\sqrt{\ln 2}} := K. \tag{68}$$

Let $m = N - 1 - d$, so we define $\mathbf{z}_i \in \mathbb{R}^m$ as:

$$\mathbf{z}_i = \left( \mathbf{u}_{d+1}(i), \ldots, \mathbf{u}_{N-1}(i) \right)^T, \tag{69}$$

and the diagonal matrix of weights $\mathbf{D} \in \mathbb{R}^{m \times m}$ as:

$$\mathbf{D} = \text{diag}\big(\lambda_{d+1}^t, \ldots, \lambda_{N-1}^t\big). \tag{70}$$

Then $E(\mathbf{x}_i)$ and $T(\mathbf{x}_i)$ can be rewritten as:

$$T(\mathbf{x}_i) = \big\|\mathbf{D}\mathbf{z}_i\big\|_2^2, \tag{71}$$

$$E(\mathbf{x}_i) = \big\|\mathbf{D}\mathbf{z}_i\big\|_2. \tag{72}$$

For $\ell > 0$, $\mathbf{u}_\ell(i)$ is zero-mean under $\boldsymbol{\pi}$. Consequently, the vector $\mathbf{z}_i$ is zero-mean and by definition satisfies $\|\mathbf{z}_i\|_{\psi_2,\boldsymbol{\pi}} \leq K\sqrt{m}$. We also notice that multiplication by a fixed matrix scales the sub-Gaussian norm linearly, and since $\mathbf{D}$ is symmetric and positive:

$$\|\mathbf{D}\mathbf{z}(i)\|_{\psi_2,\boldsymbol{\pi}} \leq K\sqrt{m}\|\mathbf{D}\|_2 = K\sqrt{m}\max_{\ell>d}\lambda_\ell^t = K\sqrt{m}\lambda_{d+1}^t. \tag{73}$$

According to (Vershynin, 2024, Prop. 6.2.1), for an $m$-dimensional, zero-mean and sub-Gaussian vector $\mathbf{Y}$ with $\|\mathbf{Y}\|_{\psi_2,\boldsymbol{\pi}} \leq \kappa$, it holds:

$$\mathbb{P}_{\boldsymbol{\pi}}\big\{\|\mathbf{Y}\|_2 \geq C\kappa(\sqrt{m}+t)\big\} \leq e^{-t^2}, \tag{74}$$

where $t \geq 0$ and $C > 0$ is a constant. Setting $\mathbf{Y} = \mathbf{D}\mathbf{z}_i$ and $\kappa = K\sqrt{m}\lambda_{d+1}^t$ gives:

$$\mathbb{P}_{\boldsymbol{\pi}}\Big\{T(\mathbf{x}_i) > C^2\lambda_{d+1}^{2t}K^2m\big(\sqrt{m}+t\big)^2\Big\} \leq e^{-t^2}. \tag{75}$$

Let $\delta \in (0,1)$ and set $t = \sqrt{\ln(1/\delta)}$. We can rewrite (75) as:

$$\mathbb{P}_{\boldsymbol{\pi}}\left\{T(\mathbf{x}_i) \leq C^2\lambda_{d+1}^{2t}K^2m\left(\sqrt{m}+\sqrt{\ln\frac{1}{\delta}}\right)^2\right\} \geq 1-\delta. \tag{76}$$

Thus, the desired confidence bound on the truncation error is:

$$\mathbb{P}_{\boldsymbol{\pi}}\left\{E(\mathbf{x}_i) \leq C\lambda_{d+1}^tK\left(m+\sqrt{m\ln\frac{1}{\delta}}\right)\right\} \geq 1-\delta. \tag{77}$$

The choice of $d$ dimensions affects both $m$ that shrinks linearly with $d$ and $\lambda_{d+1}^t$ that falls monotonically with $d$. $K$ is affected by the minimal stationary probability $\pi_{\min}$, so if the graph contains rare points then $\pi_{\min}$ may be tiny, while a well-balanced graph derives $K \sim \sqrt{N}$ and tightens the bound.

# E  DETERMINISTIC ERROR RADIUS AND PROBABILISTIC TAIL BOUND OF THE MEASURES

We derive a deterministic error radius and a high-probability confidence bound on the frame-level PS and PM measures by considering: (i) spectral truncation error due to retaining $d$ diffusion coordinates, which is separately developed in Appendix D; (ii) finite-sample uncertainty in estimating the cluster centroid and covariance. We then combine these via union bounds. In this section, we consider a fixed trail $l$, separation system $q$, and time frame $f$.

## E.1  THE PS MEASURE

Considering source indices $i, j \in \{1, \ldots, N_f\}$ (§2), we begin by analyzing the effect of the truncation error, assuming access to cluster statistics. The difference between the embedding of $\hat{\mathbf{x}}_i$ and the centroid of cluster $j$ can be expressed in the truncated subspace $\mathbb{R}^d$ and in its complement subspace $\mathbb{R}^m$, respectively denoted $\boldsymbol{\Delta}_{i,j}^{(d)}$ and $\boldsymbol{\Delta}_{i,j}^{(m)}$. Using (10), (15):

$$\boldsymbol{\Delta}_{i,j}^{(d)} = \boldsymbol{\Psi}_t^{(d)}(\hat{\mathbf{x}}_i) - \boldsymbol{\mu}_j^{(d)} \in \mathbb{R}^d, \tag{78}$$

$$\boldsymbol{\Delta}_{i,j}^{(m)} = \boldsymbol{\Psi}_t^{(m)}(\hat{\mathbf{x}}_i) - \boldsymbol{\mu}_j^{(m)} \in \mathbb{R}^m, \tag{79}$$

where $m = N - d - 1$. For completion, for every $\mathbf{x} \in \mathcal{X}$ and its global index $k \in \{1, \ldots, N\}$:

$$\boldsymbol{\mu}_j^{(m)} = \frac{1}{\left|\mathcal{C}_j^{(m)}\right|} \sum_{\boldsymbol{\psi} \in \mathcal{C}_j^{(m)}} \boldsymbol{\psi} \tag{80}$$

$$\mathcal{C}_j^{(m)} = \left\{ \boldsymbol{\Psi}_t^{(m)}(\mathbf{x}_j), \boldsymbol{\Psi}_t^{(m)}(\mathbf{x}_{j,p}) \ \middle| \ p = 1, \ldots, N_p \right\} \tag{81}$$

$$\boldsymbol{\Psi}_t^{(m)}(\mathbf{x}) = \left( \lambda_{d+1}^t \mathbf{u}_{d+1}(k), \ldots, \lambda_{N-1}^t \mathbf{u}_{N-1}(k) \right). \tag{82}$$

In the full, $N - 1$-dimensional space, the cluster $\mathcal{C}_j$ is given by:

$$\mathcal{C}_j = \left\{ \boldsymbol{\Psi}_t(\mathbf{x}_j), \boldsymbol{\Psi}_t(\mathbf{x}_{j,p}) \ \middle| \ p = 1, \ldots, N_p \right\}, \tag{83}$$

with mean $\boldsymbol{\mu} \in \mathbb{R}^{N-1}$, difference $\boldsymbol{\Delta}_{i,j} \in \mathbb{R}^{N-1}$ and covariance $\boldsymbol{\Sigma}_j \in \mathbb{R}^{(N-1) \times (N-1)}$ that hold:

$$\boldsymbol{\mu}_j = \begin{bmatrix} \boldsymbol{\mu}_j^{(d)} \\ \boldsymbol{\mu}_j^{(m)} \end{bmatrix}, \quad \boldsymbol{\Delta}_{i,j} = \begin{bmatrix} \boldsymbol{\Delta}_{i,j}^{(d)} \\ \boldsymbol{\Delta}_{i,j}^{(m)} \end{bmatrix}, \quad \boldsymbol{\Sigma}_j = \begin{bmatrix} \boldsymbol{\Sigma}_j^{(d)} & \boldsymbol{C}_j \\ \boldsymbol{C}_j^T & \boldsymbol{\Sigma}_j^{(m)} \end{bmatrix}, \tag{84}$$

where $\boldsymbol{\Sigma}_j^{(m)} \in \mathbb{R}^{m \times m}$ and $\boldsymbol{C}_j \in \mathbb{R}^{d \times m}$ are:

$$\boldsymbol{\Sigma}_j^{(m)} = \frac{1}{\left|\mathcal{C}_j^{(m)}\right| - 1} \sum_{\boldsymbol{\psi} \in \mathcal{C}_j^{(m)}} \left( \boldsymbol{\psi} - \boldsymbol{\mu}_j^{(m)} \right) \left( \boldsymbol{\psi} - \boldsymbol{\mu}_j^{(m)} \right)^T, \tag{85}$$

$$\boldsymbol{C}_j = \frac{1}{\left|\mathcal{C}_j^{(m)}\right| - 1} \sum_{p=0}^{N_p} \left( \boldsymbol{\Psi}_t^{(d)}(\mathbf{x}_{j,p}) - \boldsymbol{\mu}_j^{(d)} \right) \left( \boldsymbol{\Psi}_t^{(m)}(\mathbf{x}_{j,p}) - \boldsymbol{\mu}_j^{(m)} \right)^T. \tag{86}$$

According to 16, the squared Mahalanobis distance from $\boldsymbol{\Psi}_t(\hat{\mathbf{x}}_i)$ to $\mathcal{C}_j$ is:

$$d_M^2 \left( \boldsymbol{\Psi}_t(\hat{\mathbf{x}}_i); \boldsymbol{\mu}_j, \boldsymbol{\Sigma}_j \right) = \boldsymbol{\Delta}_{i,j}^T \left( \boldsymbol{\Sigma}_j + \epsilon I^{(N-1)} \right)^{-1} \boldsymbol{\Delta}_{i,j}, \tag{87}$$

where inversion is empirically obtained by taking $\epsilon = 10^{-6}$ with $I^{(N-1)}$ being the $N - 1$-dimensional identity matrix. To evaluate the truncation effect, we perform blockwise inversion on (87) via the Schur complement (Horn & Johnson, 2012):

$$d_M^2 \left( \boldsymbol{\Psi}_t(\hat{\mathbf{x}}_i); \boldsymbol{\mu}_j, \boldsymbol{\Sigma}_j \right) = \left( \boldsymbol{\Delta}_{i,j}^{(d)} \right)^T \left( \boldsymbol{\Sigma}_j^{(d)} + \epsilon I^{(d)} \right)^{-1} \boldsymbol{\Delta}_{i,j}^{(d)} + \boldsymbol{r}_{i,j}^T \boldsymbol{S}_j^{-1} \boldsymbol{r}_{i,j}, \tag{88}$$

where $\boldsymbol{r}_{i,j} \in \mathbb{R}^m$ and the Schur complement $\boldsymbol{S}_j \in \mathbb{R}^{m \times m}$ hold:

$$\boldsymbol{r}_{i,j} = \boldsymbol{\Delta}_{i,j}^{(m)} - \boldsymbol{C}_j^T \left( \boldsymbol{\Sigma}_j^{(d)} + \epsilon I^{(d)} \right)^{-1} \boldsymbol{\Delta}_{i,j}^{(d)}, \tag{89}$$

$$\boldsymbol{S}_j = \boldsymbol{\Sigma}_j^{(m)} - \boldsymbol{C}_j^T \left( \boldsymbol{\Sigma}_j^{(d)} + \epsilon I^{(d)} \right)^{-1} \boldsymbol{C}_j. \tag{90}$$

We now utilize the inequality:

$$\forall a, b \geq 0 : \quad \left| \sqrt{a + b} - \sqrt{a} \right| \leq \sqrt{b}, \tag{91}$$

obtained by the mean-value theorem for $f(\cdot) = \sqrt{\cdot}$ (Rudin, 1976, Ch. 5). Let us set:

$$a = \left( \boldsymbol{\Delta}_{i,j}^{(d)} \right)^T \left( \boldsymbol{\Sigma}_j^{(d)} + \epsilon I^{(d)} \right)^{-1} \boldsymbol{\Delta}_{i,j}^{(d)}, \tag{92}$$

$$b = \boldsymbol{r}_{i,j}^T \boldsymbol{S}_j^{-1} \boldsymbol{r}_{i,j}, \tag{93}$$

to obtain:

$$|\delta_{i,j}| = \left| d_M \left( \boldsymbol{\Psi}_t(\hat{\mathbf{x}}_i); \boldsymbol{\mu}_j, \boldsymbol{\Sigma}_j \right) - d_M \left( \boldsymbol{\Psi}_t^{(d)}(\hat{\mathbf{x}}_i); \boldsymbol{\mu}_j^{(d)}, \boldsymbol{\Sigma}_j^{(d)} \right) \right| \leq \sqrt{\boldsymbol{r}_{i,j}^T \boldsymbol{S}_j^{-1} \boldsymbol{r}_{i,j}}. \tag{94}$$

Namely, $|\delta_{i,j}|$ is the truncation error of this Mahalanobis distance. From (17), it holds that:

$$\delta_{i,i} = A_i - A_i^{(d)}, \quad \delta_{i,j^*} = B_i - B_i^{(d)}, \tag{95}$$

where $j^*$ is defined in (17) and:

$$A_i = d_M \left( \boldsymbol{\Psi}_t(\hat{\mathbf{x}}_i); \boldsymbol{\mu}_i, \boldsymbol{\Sigma}_i \right), \tag{96}$$

$$B_i = d_M \left( \boldsymbol{\Psi}_t(\hat{\mathbf{x}}_i); \boldsymbol{\mu}_{j^*}, \boldsymbol{\Sigma}_{j^*} \right). \tag{97}$$

Consider the $N-1$-dimensional representation of $\text{PS}_i^{(d)}$, i.e. $\text{PS}_i$ (18):

$$\text{PS}_i = 1 - \frac{A_i}{A_i + B_i}, \tag{98}$$

which is smooth and differentiable in $A_i$ and $B_i$, since by definition $A_i + B_i > 0$. We assume and have empirically validated that truncation introduces a small relative change, i.e.:

$$|\delta_{i,i}| \ll A_i^{(d)} + B_i^{(d)}, \tag{99}$$

$$|\delta_{i,j^*}| \ll A_i^{(d)} + B_i^{(d)}, \tag{100}$$

making the first-order Taylor expansion of $\text{PS}_i$ around $\left( A_i^{(d)}, B_i^{(d)} \right)$ valid. We can therefore write:

$$\text{PS}_i - \text{PS}_i^{(d)} \simeq \frac{\partial \text{PS}_i}{\partial A_i} \left( A_i^{(d)}, B_i^{(d)} \right) \delta_{i,i} + \frac{\partial \text{PS}_i}{\partial B_i} \left( A_i^{(d)}, B_i^{(d)} \right) \delta_{i,j^*} \tag{101}$$

$$= -\frac{B_i^{(d)}}{\left( A_i^{(d)} + B_i^{(d)} \right)^2} \delta_{i,i} + \frac{A_i^{(d)}}{\left( A_i^{(d)} + B_i^{(d)} \right)^2} \delta_{i,j^*},$$

where the quadratic remainder in the expansion is empirically one order smaller than the first-order term and can be safely dropped. Applying the triangle inequality and (94) yields the deterministic error radius in the PS measure:

$$\left| \text{PS}_i - \text{PS}_i^{(d)} \right| \le \frac{B_i^{(d)} |\delta_{i,i}| + A_i^{(d)} |\delta_{i,j^*}|}{\left( A_i^{(d)} + B_i^{(d)} \right)^2} = \frac{B_i^{(d)} \sqrt{\boldsymbol{r}_{i,i}^T \boldsymbol{S}_i^{-1} \boldsymbol{r}_{i,i}} + A_i^{(d)} \sqrt{\boldsymbol{r}_{i,j^*}^T \boldsymbol{S}_{j^*}^{-1} \boldsymbol{r}_{i,j^*}}}{\left( A_i^{(d)} + B_i^{(d)} \right)^2}. \tag{102}$$

We now quantify the uncertainty in $\widehat{\text{PS}}_i^{(d)}$ due to finite-sample cluster statistics. Empirically, we observe that cluster coordinates exhibit weak dependence between one another and derive from (Bartlett, 1946) the following cut-off rule for the effective sample size of cluster $\mathcal{C}_j^{(d)}$:

$$n_{j,\text{eff}} = \frac{n_j}{1 + 2 \sum_{\ell=1}^{L_j} \hat{\rho}_{j,\ell}}, \quad n_j = \left| \mathcal{C}_j^{(d)} \right| \tag{103}$$

where $\hat{\rho}_{j,\ell}$ is the empirical average Pearson auto-correlation of coordinates at lag $\ell$, and:

$$L_j = \arg\min_{\ell} \left\{ |\hat{\rho}_{j,\ell}| < \frac{z_{0.975}}{\sqrt{n_j - \ell}} \right\}. \tag{104}$$

Empirical evidence across $5,000$ graphs suggest that on average $\sum_{\ell} \hat{\rho}_{j,\ell} \simeq 0.2$, and so we set $n_{j,\text{eff}} = 0.7 \, n_j$ for all clusters.

To bound the deviation between the estimated and true cluster mean and covariance, we employ the vector and matrix Bernstein (Vershynin, 2024, Props. 2.8.1, 4.7.1) and the dependent Hanson-Wright inequalities (Adamczak, 2014, Thm. 2.5). For every $\delta_{j,\boldsymbol{\mu}}^{\text{PS}}, \delta_{j,\boldsymbol{\Sigma}}^{\text{PS}} \in (0, 1/2)$, with respective least probabilities $1 - \delta_{j,\boldsymbol{\mu}}^{\text{PS}}$ and $1 - \delta_{j,\boldsymbol{\Sigma}}^{\text{PS}}$:

$$\left| \boldsymbol{\mu}_j - \hat{\boldsymbol{\mu}}_j \right| \le \sqrt{\frac{2 \lambda_{\max} \left( \widehat{\boldsymbol{\Sigma}}_j^{(d)} \right) \ln \left( 2/\delta_{j,\boldsymbol{\mu}}^{\text{PS}} \right)}{n_{j,\text{eff}}}} := \Delta_{j,\boldsymbol{\mu}} \tag{105}$$

$$\left\| \boldsymbol{\Sigma}_j^{(d)} - \widehat{\boldsymbol{\Sigma}}_j^{(d)} \right\|_2 \le C \lambda_{\max} \left( \widehat{\boldsymbol{\Sigma}}_j^{(d)} \right) \left( \frac{r_j}{n_{j,\text{eff}}} + \frac{r_j + \ln \left( 2/\delta_{j,\boldsymbol{\Sigma}}^{\text{PS}} \right)}{n_{j,\text{eff}}} \right) := \Delta_{j,\boldsymbol{\Sigma}}, \tag{106}$$

with an absolute constant $C > 0$ and the ratio:

$$r_j = \frac{\text{tr}\left(\widehat{\mathbf{\Sigma}}_j^{(d)}\right)}{\lambda_{\max}\left(\widehat{\mathbf{\Sigma}}_j^{(d)}\right)}. \tag{107}$$

Let us integrate the definitions of $\widehat{A}_i^{(d)}$ and $\widehat{B}_i^{(d)}$ (17) with (105)-(106). Then, with probability of at least $1 - \delta_{i,\boldsymbol{\mu}}^{\text{PS}} - \delta_{i,\boldsymbol{\Sigma}}^{\text{PS}}$, $\widehat{A}_i^{(d)}$ and $\widehat{B}_i^{(d)}$ deviate from their true versions by $\varepsilon^{\text{PS}}\left(\widehat{A}_i^{(d)}\right)$ and $\varepsilon^{\text{PS}}\left(\widehat{B}_i^{(d)}\right)$, bounded by:

$$\varepsilon^{\text{PS}}\left(\widehat{A}_i^{(d)}\right) \leq 2\sqrt{\widehat{A}_i^{(d)}}\,\Delta_{i,\boldsymbol{\mu}}\sqrt{\frac{\lambda_{\max}\left(\widehat{\mathbf{\Sigma}}_i^{(d)}\right)}{\tilde{\lambda}_{\min}\left(\widehat{\mathbf{\Sigma}}_i^{(d)}\right)}} + \widehat{A}_i^{(d)}\frac{\Delta_{i,\boldsymbol{\Sigma}}}{\lambda_{\max}\left(\widehat{\mathbf{\Sigma}}_i^{(d)}\right)}, \tag{108}$$

$$\varepsilon^{\text{PS}}\left(\widehat{B}_i^{(d)}\right) \leq 2\sqrt{\widehat{B}_i^{(d)}}\,\Delta_{j^*,\boldsymbol{\mu}}\sqrt{\frac{\lambda_{\max}\left(\widehat{\mathbf{\Sigma}}_{j^*}^{(d)}\right)}{\tilde{\lambda}_{\min}\left(\widehat{\mathbf{\Sigma}}_{j^*}^{(d)}\right)}} + \widehat{B}_i^{(d)}\frac{\Delta_{j^*,\boldsymbol{\Sigma}}}{\lambda_{\max}\left(\widehat{\mathbf{\Sigma}}_{j^*}^{(d)}\right)}. \tag{109}$$

We avoid extremely loose bounds by replacing tiny, rarely observable eigenvalues, by a robust floor eigenvalue. Given a matrix $\mathbf{A} \in \mathbb{R}^{d \times d}$, we define $\tilde{\lambda}_{\min}(\mathbf{A})$ as (Horn & Johnson, 2012, Thm. 4.3.1):

$$\tilde{\lambda}_{\min}(\mathbf{A}) = \lambda_{\min}\left(\mathbf{A} + \epsilon_r \lambda_{\max}(\mathbf{A})\,I^{(d)}\right), \tag{110}$$

where $\epsilon_r = 0.05$ is typically taken and $I^{(d)}$ is the identity matrix. Ultimately, let us define the Euclidean Lipschitz constant $L_{i,\text{lip}}$ as:

$$L_i^{\text{PS}} = \sqrt{\left(\frac{\partial \text{PS}_i}{\partial A_i}\left(A_i^{(d)}, B_i^{(d)}\right)\right)^2 + \left(\frac{\partial \text{PS}_i}{\partial B_i}\left(A_i^{(d)}, B_i^{(d)}\right)\right)^2} = \frac{\sqrt{\left(\widehat{A}_i^{(d)}\right)^2 + \left(\widehat{B}_i^{(d)}\right)^2}}{\left(\widehat{A}_i^{(d)} + \widehat{B}_i^{(d)}\right)^2}, \tag{111}$$

which enables us to bound the finite-sample deviation of $\widehat{\text{PS}}_i^{(d)}$ with:

$$\left|\widehat{\text{PS}}_i^{(d)} - \text{PS}_i^{(d)}\right| \leq L_i^{\text{PS}}\sqrt{\varepsilon^{\text{PS}}\left(\widehat{A}_i^{(d)}\right) + \varepsilon^{\text{PS}}\left(\widehat{B}_i^{(d)}\right)}. \tag{112}$$

Finally, we employ the triangle inequality on both error sources (102) and (112) and obtain for $\delta_i^{\text{PS}} = \delta_{i,\boldsymbol{\mu}}^{\text{PS}} + \delta_{i,\boldsymbol{\Sigma}}^{\text{PS}}$, with $\delta_i^{\text{PS}} \in (0,1)$:

$$\mathbb{P}_{\boldsymbol{\pi}}\left\{\left|\widehat{\text{PS}}_i^{(d)} - \text{PS}_i\right| \leq \right. \tag{113}$$

$$\left. \frac{\widehat{B}_i^{(d)}\sqrt{\boldsymbol{r}_{i,i}^T \boldsymbol{S}_i^{-1} \boldsymbol{r}_{i,i}} + \widehat{A}_i^{(d)}\sqrt{\boldsymbol{r}_{i,j^*}^T \boldsymbol{S}_{j^*}^{-1} \boldsymbol{r}_{i,j^*}}}{\left(\widehat{A}_i^{(d)} + \widehat{B}_i^{(d)}\right)^2} + L_i^{\text{PS}}\sqrt{\varepsilon^{\text{PS}}\left(\widehat{A}_i^{(d)}\right) + \varepsilon^{\text{PS}}\left(\widehat{B}_i^{(d)}\right)}\right\} \geq 1 - \delta_i^{\text{PS}}.$$

We now analyze the obtained expression separately for the deterministic and probabilistic terms. In the former term, the two square-root terms are energies that leak into the truncated complement after regressing out the retained $d$ diffusion coordinates. Intuitively, $\boldsymbol{\Sigma}_j^{(d)}$ encodes the local anisotropy of cluster $j$ in the kept coordinates, $\boldsymbol{C}_j$ represents coupling of residual energy in the truncated block, and $\boldsymbol{\Sigma}_j^{(m)}$ is the spread that remains in the truncated block. Therefore, larger $\boldsymbol{S}_j$ down-weights complement deviations, reducing the bias, while $\boldsymbol{r}_{i,j}$ and $\boldsymbol{S}_j$ co-vary through the cross-covariance $\boldsymbol{C}_j$. Namely, increasing $\boldsymbol{C}_j$ shrinks both $\boldsymbol{r}_{i,j}$ and $\boldsymbol{S}_j$, while decreasing $\boldsymbol{C}_j$ does the opposite. The practical rule is to prevent tiny $\lambda_{\min}(\boldsymbol{S}_j)$, e.g., by promoting such directions into the kept set via a local choice of $d$, and shape distortions so the complement is predictable from the kept coordinates, keeping $\boldsymbol{r}_{i,j}$ small.

For the probabilistic part, its width reflects uncertainty in the empirical centroid and covariance of the attributed and nearest foreign clusters, where $\lambda_{\max}(\widehat{\boldsymbol{\Sigma}}_j^{(d)})$ and $n_{j,\text{eff}}$ determine the width primarily. Practically, correlated distortions shrink $n_{j,\text{eff}}$ and widen the bound, and a smaller spectral flatness ratio $r_j$ yields tighter matrix concentration. As expected, the probabilistic piece dominates the deterministic as shown in Figure 9, which underlines the importance of cluster construction and dependence control.

## E.2 THE PM MEASURE

As in the PS case, we start with the truncation error and assume access to cluster statistics. Let us reconsider (78)-(90), but with two adjustments. First, the cluster coordinates are centered around the cluster reference embedding and not the cluster mean. Given $m = N - d - 1$, we define:

$$\boldsymbol{\Delta}_{i,p}^{(d)} = \boldsymbol{\Psi}_t^{(d)}(\mathbf{x}_{i,p}) - \boldsymbol{\Psi}_t^{(d)}(\mathbf{x}_i) \in \mathbb{R}^d, \tag{114}$$

$$\boldsymbol{\Delta}_{i,p}^{(m)} = \boldsymbol{\Psi}_t^{(m)}(\mathbf{x}_{i,p}) - \boldsymbol{\Psi}_t^{(m)}(\mathbf{x}_i) \in \mathbb{R}^m. \tag{115}$$

Second, the cluster is now absent the reference embedding. Namely, the full $N - 1$-dimensional cluster is (83):

$$\tilde{\mathcal{C}}_i = \mathcal{C}_i \setminus \Psi_t(\mathbf{x}_i). \tag{116}$$

The cluster $\tilde{\mathcal{C}}_i$ has difference $\boldsymbol{\Delta}_{i,p} \in \mathbb{R}^{N-1}$ for every $p \in \{1, \ldots, N_p\}$ and covariance $\tilde{\boldsymbol{\Sigma}}_i \in \mathbb{R}^{(N-1)\times(N-1)}$ that hold (19):

$$\boldsymbol{\Delta}_{i,p} = \begin{bmatrix} \boldsymbol{\Delta}_{i,p}^{(d)} \\ \boldsymbol{\Delta}_{i,p}^{(m)} \end{bmatrix}, \quad \tilde{\boldsymbol{\Sigma}}_i = \begin{bmatrix} \tilde{\boldsymbol{\Sigma}}_i^{(d)} & \tilde{\boldsymbol{C}}_i \\ \tilde{\boldsymbol{C}}_i^T & \tilde{\boldsymbol{\Sigma}}_i^{(m)} \end{bmatrix}, \tag{117}$$

with $\tilde{\boldsymbol{\Sigma}}_i^{(m)} \in \mathbb{R}^{m \times m}$ and $\tilde{\boldsymbol{C}}_i \in \mathbb{R}^{d \times m}$ being:

$$\tilde{\boldsymbol{\Sigma}}_i^{(m)} = \frac{1}{\left|\tilde{\mathcal{C}}_i^{(m)}\right| - 1} \sum_{\boldsymbol{\psi} \in \tilde{\mathcal{C}}_i^{(m)}} \left(\boldsymbol{\psi} - \Psi_t^{(m)}(\mathbf{x}_i)\right) \left(\boldsymbol{\psi} - \Psi_t^{(m)}(\mathbf{x}_i)\right)^T, \tag{118}$$

$$\tilde{\boldsymbol{C}}_i = \frac{1}{\left|\tilde{\mathcal{C}}_i^{(m)}\right| - 1} \sum_{p=1}^{N_p} \left(\boldsymbol{\Psi}_t^{(d)}(\mathbf{x}_{i,p}) - \Psi_t^{(d)}(\mathbf{x}_i)\right) \left(\boldsymbol{\Psi}_t^{(m)}(\mathbf{x}_{i,p}) - \Psi_t^{(m)}(\mathbf{x}_i)\right)^T. \tag{119}$$

In $N - 1$ dimensions, the squared Mahalanobis distance from $\boldsymbol{\Psi}_t(\mathbf{x}_{i,p})$ to $\tilde{\mathcal{C}}_i$ is given by (16):

$$d_M^2\left(\boldsymbol{\Psi}_t(\mathbf{x}_{i,p}); \boldsymbol{\Psi}_t(\mathbf{x}_i), \tilde{\boldsymbol{\Sigma}}_i\right) = \boldsymbol{\Delta}_{i,p}^T\left(\tilde{\boldsymbol{\Sigma}}_i + \epsilon I^{(N-1)}\right)^{-1} \boldsymbol{\Delta}_{i,p}, \tag{120}$$

where as in (87), inversion has been empirically obtained with $\epsilon = 10^{-6}$ and the $N - 1$-dimensional identity matrix $I^{(N-1)}$. We again turn to the Schur complement (Horn & Johnson, 2012) and decompose (120):

$$d_M^2\left(\boldsymbol{\Psi}_t(\mathbf{x}_{i,p}); \boldsymbol{\Psi}_t(\mathbf{x}_i), \tilde{\boldsymbol{\Sigma}}_i\right) = \left(\boldsymbol{\Delta}_{i,p}^{(d)}\right)^T\left(\tilde{\boldsymbol{\Sigma}}_i^{(d)} + \epsilon I^{(d)}\right)^{-1} \boldsymbol{\Delta}_{i,p}^{(d)} + \boldsymbol{r}_{i,p}^T \boldsymbol{S}_i^{-1} \boldsymbol{r}_{i,p}, \tag{121}$$

with $\boldsymbol{r}_{i,p} \in \mathbb{R}^m$ and the Schur complement $\boldsymbol{S}_i \in \mathbb{R}^{m \times m}$ being:

$$\boldsymbol{r}_{i,p} = \boldsymbol{\Delta}_{i,p}^{(m)} - \tilde{\boldsymbol{C}}_i^T\left(\tilde{\boldsymbol{\Sigma}}_i^{(d)} + \epsilon I^{(d)}\right)^{-1} \boldsymbol{\Delta}_{i,p}^{(d)}, \tag{122}$$

$$\boldsymbol{S}_i = \tilde{\boldsymbol{\Sigma}}_i^{(m)} - \tilde{\boldsymbol{C}}_i^T\left(\tilde{\boldsymbol{\Sigma}}_i^{(d)} + \epsilon I^{(d)}\right)^{-1} \tilde{\boldsymbol{C}}_i. \tag{123}$$

Let us define the set of squared Mahalanobis distances of cluster $\tilde{\mathcal{C}}_i$ in dimension $N - 1$ as:

$$\mathcal{G}_i = \left\{d_M^2\left(\boldsymbol{\Psi}_t(\mathbf{x}_{i,p}); \boldsymbol{\Psi}_t(\mathbf{x}_i), \tilde{\boldsymbol{\Sigma}}_i\right) \,\Big|\, p = 1, \ldots, N_p\right\}, \tag{124}$$

in accordance to the truncated version of $\mathcal{G}_i^{(d)}$ in (20). By employing (121), for every $p \in \{1, \ldots, N_p\}$, we can bound the truncation error of the squared Mahalanobis distance as follows:

$$d_M^2\left(\boldsymbol{\Psi}_t(\mathbf{x}_{i,p}); \boldsymbol{\Psi}_t(\mathbf{x}_i), \tilde{\boldsymbol{\Sigma}}_i\right) - d_M^2\left(\boldsymbol{\Psi}_t^{(d)}(\mathbf{x}_{i,p}); \boldsymbol{\Psi}_t^{(d)}(\mathbf{x}_i), \tilde{\boldsymbol{\Sigma}}_i^{(d)}\right) = \tag{125}$$

$$\boldsymbol{r}_{i,p}^T \boldsymbol{S}_i^{-1} \boldsymbol{r}_{i,p} := \delta_{\mathcal{G}_i,p},$$

and the difference between the mean of the elements in $\mathcal{G}_i$ and $\mathcal{G}_i^{(d)}$ can be expressed as:

$$\mu_{\mathcal{G}_i} - \mu_{\mathcal{G}_i^{(d)}} = \frac{1}{|\mathcal{G}_i|} \sum_{g \in \mathcal{G}_i} g - \frac{1}{\left|\mathcal{G}_i^{(d)}\right|} \sum_{g \in \mathcal{G}_i^{(d)}} g = \frac{1}{N_p} \sum_{p=1}^{N_p} \boldsymbol{r}_{i,p}^T \boldsymbol{S}_i^{-1} \boldsymbol{r}_{i,p} = \tag{126}$$

$$\frac{1}{N_p} \sum_{p=1}^{N_p} \delta_{\mathcal{G}_i,p} := \delta_{\mathcal{G}_i,\mu}.$$

Similarly, we can express the deviation of the variance:

$$\sigma_{\mathcal{G}_i}^2 - \sigma_{\mathcal{G}_i^{(d)}}^2 = \frac{1}{|\mathcal{G}_i| - 1} \sum_{g \in \mathcal{G}_i} (g - \mu_{\mathcal{G}_i})^2 - \frac{1}{\left|\mathcal{G}_i^{(d)}\right| - 1} \sum_{g \in \mathcal{G}_i^{(d)}} \left(g - \mu_{\mathcal{G}_i^{(d)}}\right)^2, \tag{127}$$

and with (126) and the Cauchy-Schwartz inequality, we can obtain:

$$\left|\sigma_{\mathcal{G}_i}^2 - \sigma_{\mathcal{G}_i^{(d)}}^2\right| \le \frac{N_p}{N_p - 1} \left(2\delta_{\mathcal{G}_i,p}^{\max} \left(\sigma_{\mathcal{G}_i} + \sigma_{\mathcal{G}_i^{(d)}}\right) + \left(\delta_{\mathcal{G}_i,p}^{\max}\right)^2\right), \tag{128}$$

where $\delta_{\mathcal{G}_i,p}^{\max} = \max_p \delta_{\mathcal{G}_i,p}$. The Gamma-matching parameters in the truncated and full dimensions are (22):

$$k_i^{(d)} = \frac{\mu_{\mathcal{G}_i^{(d)}}^2}{\sigma_{\mathcal{G}_i^{(d)}}^2}, \quad k_i = \frac{\mu_{\mathcal{G}_i}^2}{\sigma_{\mathcal{G}_i}^2}, \tag{129}$$

$$\theta_i^{(d)} = \frac{\sigma_{\mathcal{G}_i^{(d)}}^2}{\mu_{\mathcal{G}_i^{(d)}}}, \quad \theta_i = \frac{\sigma_{\mathcal{G}_i}^2}{\mu_{\mathcal{G}_i}}, \tag{130}$$

and their deviations can be bounded by considering (126), (128):

$$\left|k_i - k_i^{(d)}\right| \le C_1 \delta_{\mathcal{G}_i,p}^{\max} \frac{N_p}{N_p - 1} \frac{\mu_{\mathcal{G}_i} + \mu_{\mathcal{G}_i^{(d)}}}{\sigma_{\mathcal{G}_i^{(d)}}^2} := \delta_{\mathcal{G}_i,k}, \tag{131}$$

$$\left|\theta_i - \theta_i^{(d)}\right| \le C_2 \delta_{\mathcal{G}_i,p}^{\max} \frac{N_p}{N_p - 1} \frac{\sigma_{\mathcal{G}_i}^2 + \sigma_{\mathcal{G}_i^{(d)}}^2}{\mu_{\mathcal{G}_i^{(d)}}^2} := \delta_{\mathcal{G}_i,\theta}, \tag{132}$$

with universal constants $C_1, C_2 > 0$. Let the squared Mahalanobis distance from the output embedding to the cluster be:

$$d_M^2\left(\boldsymbol{\Psi}_t^{(d)}(\hat{\mathbf{x}}_i); \boldsymbol{\Psi}_t^{(d)}(\mathbf{x}_i), \tilde{\boldsymbol{\Sigma}}_i^{(d)}\right) := a_i, \tag{133}$$

and employing (125) for the output embedding yields:

$$d_M^2\left(\boldsymbol{\Psi}_t(\hat{\mathbf{x}}_i); \boldsymbol{\Psi}_t(\mathbf{x}_i), \tilde{\boldsymbol{\Sigma}}_i\right) - d_M^2\left(\boldsymbol{\Psi}_t^{(d)}(\hat{\mathbf{x}}_i); \boldsymbol{\Psi}_t^{(d)}(\mathbf{x}_i), \tilde{\boldsymbol{\Sigma}}_i^{(d)}\right) = \boldsymbol{r}_{i,a}^T \boldsymbol{S}_i^{-1} \boldsymbol{r}_{i,a} := \delta_{\mathcal{G}_i,a}. \tag{134}$$

As in (23), the PM definition in dimension $N - 1$ can be expressed using the regularized upper incomplete gamma function $Q(k, x) = \Gamma(k, x)/\Gamma(k)$:

$$\text{PM}_i = Q\left(k_i, \frac{a_i}{\theta_i}\right). \tag{135}$$

Consider the truncation-induced ellipsoid:

$$\mathcal{B}_i = \left\{(k_i', \theta_i', a_i') : \left|k_i' - k_i^{(d)}\right| \le \delta_{\mathcal{G}_i,k}, \left|\theta_i' - \theta_i^{(d)}\right| \le \delta_{\mathcal{G}_i,\theta}, \left|a_i' - a_i^{(d)}\right| \le \delta_{\mathcal{G}_i,a}\right\}, \tag{136}$$

For $F(k, \theta, a) = Q(k, a/\theta)$, the gradient with the partial derivatives with respect to $k$, $\theta$ and $a$ is:

$$\nabla F(k, \theta, a) = \begin{pmatrix} \dfrac{1}{\Gamma(k)} \displaystyle\int_x^\infty t^{k-1} e^{-t} \ln t \, \mathrm{d}t - \psi(k) Q(k, x) \\[2ex] \dfrac{a}{\theta^2} \dfrac{x^{k-1} e^{-x}}{\Gamma(k)} \\[2ex] -\dfrac{1}{\theta} \dfrac{x^{k-1} e^{-x}}{\Gamma(k)} \end{pmatrix}, \tag{137}$$

where $x = a/\theta$ and $\psi(\cdot)$ is the digamma function. Since $\nabla F(k, \theta, a)$ is continuous and bounded on the compact set $\mathcal{B}_i$, we set:

$$L_{\mathcal{B}_i} = \sup_{(k,\theta,a) \in \mathcal{B}_i} \left\| \nabla F(k, \theta, a) \right\|_2 < \infty. \tag{138}$$

To yield the bound on the PM measure due to truncation, we notice that both $(k_i, \theta_i, a_i)$ and $\left(k_i^{(d)}, \theta_i^{(d)}, a_i^{(d)}\right)$ lie in $\mathcal{B}_i$, and apply the multivariate mean-value theorem to yield the following:

$$\left| \text{PM}_i - \text{PM}_i^{(d)} \right| \leq L_{\mathcal{B}_i} \left( \delta_{\mathcal{G}_i,k}^2 + \delta_{\mathcal{G}_i,\theta}^2 + \delta_{\mathcal{G}_i,a}^2 \right)^{1/2}. \tag{139}$$

However, this bound can be tightened. We notice that $Q(k, x)$ (23) is monotonically increasing in $k$ and decreasing in $x$, for $k, x > 0$ (137). We assume that on $\mathcal{B}_i$, and for all $(\theta, a) \in \mathcal{B}_i$, $\partial F/\partial k$ does not change signs, or otherwise we fallback to (139). Consequently, the maximal change of $Q(k, x)$ inside $\mathcal{B}_i$ is attained at one of its eight corners, and (139) can be tightened to this PM error radius:

$$\left| \text{PM}_i - \text{PM}_i^{(d)} \right| \leq \max_{(k_c,\theta_c,a_c) \in \partial \mathcal{B}_i} \left| Q\left(k_c, a_c/\theta_c\right) - Q\left(\hat{k}_i^{(d)}, \hat{a}_i^{(d)}/\hat{\theta}_i^{(d)}\right) \right|. \tag{140}$$

As in the PS case, we now analyze how the finite number of coordinates in a cluster leads to uncertainty in the PM evaluation. Let $R_i$ be the maximal squared Mahalanobis distance in $\mathcal{G}_i$, namely:

$$R_i = \max_{g \in \mathcal{G}_i} g. \tag{141}$$

Again, similarly to the PS case, we utilize the vector and matrix Bernstein (Vershynin, 2024, Props. 2.8.1, 4.7.1) and the dependent Hanson-Wright inequalities (Adamczak, 2014, Thm. 2.5). Let us consider the confidence parameters $\delta_{i,\mu}^{\text{PM}}, \delta_{i,\sigma}^{\text{PM}}, \delta_{i,a}^{\text{PM}} \in (0, 1/3)$, so with respective least probabilities of $1 - \delta_{i,\mu}^{\text{PM}}$, $1 - \delta_{i,\sigma}^{\text{PM}}$, $1 - \delta_{i,a}^{\text{PM}}$:

$$\left| \mu_{\mathcal{G}_i^{(d)}} - \hat{\mu}_{\mathcal{G}_i^{(d)}} \right| \leq \sqrt{\frac{2\hat{\sigma}_{\mathcal{G}_i^{(d)}}^2 \ln\left(2/\delta_{i,\mu}^{\text{PM}}\right)}{N_p}} + \frac{3R_i \ln\left(2/\delta_{i,\mu}^{\text{PM}}\right)}{N_p} := \Delta_{i,\mu}, \tag{142}$$

$$\left| \sigma_{\mathcal{G}_i^{(d)}} - \hat{\sigma}_{\mathcal{G}_i^{(d)}} \right| \leq \sqrt{\frac{2R_i^2 \ln\left(2/\delta_{i,\sigma}^{\text{PM}}\right)}{N_p}} + \frac{3R_i^2 \ln\left(2/\delta_{i,\sigma}^{\text{PM}}\right)}{N_p} := \Delta_{i,\sigma}, \tag{143}$$

$$\left| a_i - \hat{a}_i \right| \leq R_i \sqrt{\frac{\ln\left(2/\delta_{i,a}^{\text{PM}}\right)}{N_p}} := \Delta_{i,a}. \tag{144}$$

Recalling the definition of $k_i^{(d)}$, $\theta_i^{(d)}$ from (129), (130), since by design $\hat{\mu}_{\mathcal{G}_i^{(d)}}, \hat{\sigma}_{\mathcal{G}_i^{(d)}} > 0$, and since we empirically validate that $\Delta_{i,\mu} \ll \hat{\mu}_{\mathcal{G}_i^{(d)}}$, $\Delta_{i,\sigma} \ll \hat{\sigma}_{\mathcal{G}_i^{(d)}}$, we can apply the first-order Taylor expansions to $k_i^{(d)}$, $\theta_i^{(d)}$ around $\hat{k}_i^{(d)}$, $\hat{\theta}_i^{(d)}$, respectively. Apply the triangle inequality to it gives:

$$\left| k_i^{(d)} - \hat{k}_i^{(d)} \right| \leq \left| \frac{\partial k_i^{(d)}}{\partial \mu} \right| \Delta_{i,\mu} + \left| \frac{\partial k_i^{(d)}}{\partial \sigma} \right| \Delta_{i,\sigma} = \left| \frac{2\hat{\mu}_{\mathcal{G}_i^{(d)}}}{\hat{\sigma}_{\mathcal{G}_i^{(d)}}^2} \right| \Delta_{i,\mu} + \left| \frac{-2\hat{\mu}_{\mathcal{G}_i^{(d)}}^2}{\hat{\sigma}_{\mathcal{G}_i^{(d)}}^3} \right| \Delta_{i,\sigma} := \Delta_{i,k}, \tag{145}$$

$$\left| \theta_i^{(d)} - \hat{\theta}_i^{(d)} \right| \leq \left| \frac{\partial \theta_i^{(d)}}{\partial \mu} \right| \Delta_{i,\mu} + \left| \frac{\partial \theta_i^{(d)}}{\partial \sigma} \right| \Delta_{i,\sigma} = \left| \frac{-\hat{\sigma}_{\mathcal{G}_i^{(d)}}^2}{\hat{\mu}_{\mathcal{G}_i^{(d)}}^2} \right| \Delta_{i,\mu} + \left| \frac{2\hat{\sigma}_{\mathcal{G}_i^{(d)}}}{\hat{\mu}_{\mathcal{G}_i^{(d)}}} \right| \Delta_{i,\sigma} := \Delta_{i,\theta}. \tag{146}$$

Empirically, rarely $\Delta_{i,k}$, $\Delta_{i,\theta}$ or $\Delta_{i,a}$ become extremely loose. To avoid this behavior, we practically regularize the box by setting:

$$\Delta_{i,k} \rightarrow \min\left(\Delta_{i,k}, 0.5k_i^{(d)}\right), \tag{147}$$

$$\Delta_{i,\theta} \rightarrow \min\left(\Delta_{i,\theta}, 0.5\theta_i^{(d)}\right), \tag{148}$$

$$\Delta_{i,a} \rightarrow \min\left(\Delta_{i,a}, 0.5a_i^{(d)}\right). \tag{149}$$

Let us consider the local box of values:

$$\mathcal{B}_i^{\text{loc}} = \left\{ (k_i', \theta_i', a_i') : k_i' \in \left[ k_i^{(d)} \pm \Delta_{i,k} \right], \theta_i' \in \left[ \theta_i^{(d)} \pm \Delta_{i,\theta} \right], a_i' \in \left[ a_i^{(d)} \pm \Delta_{i,a} \right] \right\}. \qquad (150)$$

As discussed earlier, $Q(k,x)$ is monotonically increasing in $k$ and decreasing in $x$, for $k$, $x > 0$ (137). Consequently, the maximal change of $Q(k,x)$ inside $\mathcal{B}_i^{\text{loc}}$ is attained at one of its eight corners. Thus, the finite-sample error of the PM measure in dimension $d$ is bounded by:

$$\left| \text{PM}_i^{(d)} - \widehat{\text{PM}}_i^{(d)} \right| \leq \max_{(k_c, \theta_c, a_c) \in \partial \mathcal{B}_i^{\text{loc}}} \left| Q\left( k_c, a_c/\theta_c \right) - Q\left( \hat{k}_i^{(d)}, \hat{a}_i^{(d)}/\hat{\theta}_i^{(d)} \right) \right|. \qquad (151)$$

Ultimately, we combine the deterministic error radius with the probabilistic width. Let $\delta_i^{\text{PM}} = \delta_{i,\mu}^{\text{PM}} + \delta_{i,\sigma}^{\text{PM}} + \delta_{i,a}^{\text{PM}}$, which yields for $\delta_i^{\text{PM}} \in (0,1)$:

$$\mathbb{P}_{\boldsymbol{\pi}} \left\{ \left| \widehat{\text{PM}}_i^{(d)} - \text{PM}_i \right| \leq \right. \qquad (152)$$

$$\max_{(k_c, \theta_c, a_c) \in \partial \mathcal{B}_i} \left| Q\left( k_c, a_c/\theta_c \right) - Q\left( \hat{k}_i^{(d)}, \hat{a}_i^{(d)}/\hat{\theta}_i^{(d)} \right) \right| +$$

$$\left. \max_{(k_c, \theta_c, a_c) \in \partial \mathcal{B}_i^{\text{loc}}} \left| Q\left( k_c, a_c/\theta_c \right) - Q\left( \hat{k}_i^{(d)}, \hat{a}_i^{(d)}/\hat{\theta}_i^{(d)} \right) \right| \right\} \geq 1 - \delta_i^{\text{PM}}.$$

In the deterministic term, large cross-block coupling $\widetilde{C}_i$ or residual spread $\widetilde{\boldsymbol{\Sigma}}_i^{(m)}$ again directly inflate the error radius via the Schur complement.

In the probabilistic part, the local box $\mathcal{B}_i^{\text{loc}}$ aggregates two finite-sample pieces. The first is the uncertainty of the moment, with $\Delta_{i,\mu}$ and $\Delta_{i,\sigma}$ scale as $1/N_p$ but are amplified by the maximal radius of Mahalanobis within the cluster $R_i$. Heavy outliers increase $R_i$ and widen both bounds. The second is the uncertainty of the distance of the output, contributed by $\Delta_{i,a}$ which is also proportional to $R_i$ but scales by $1/\sqrt{N_p}$. Again, this emphasizes the importance of the design of distortions.

# F    ERROR RADIUS AND PROBABILISTIC CONFIDENCE BOUND OF THE PCC AND SRCC

In this Appendix, we propagate the frame-level error radius and probabilistic widths developed in Appendix E.1 and E.2 to the reported PCC and SRCC values.

We start by fixing a trial $l$, a source separation system $q$, and a time frame $f$. Let the indices of the active sources in frame $f$ be $\mathcal{S}_f^l$ and consider a source $i \in \mathcal{S}_f^l$. The observation of measure $\mathcal{P} \in \{\text{PS}, \text{PM}\}$, denoted $\hat{v}_{i,f}^{q,l,\mathcal{P}}$, can be decomposed as:

$$\hat{v}_{i,f}^{q,l,\mathcal{P}} = v_{i,f}^{q,l,\mathcal{P}} + \widetilde{\beta}_{i,f}^{q,l,\mathcal{P}} + \zeta_{i,f}^{q,l,\mathcal{P}}, \qquad (153)$$

where:

$$\widetilde{\beta}_{i,f}^{q,l,\mathcal{P}} = \beta_{i,f}^{q,l,\mathcal{P}} + \mu_{i,f}^{q,l,\mathcal{P}}, \qquad (154)$$

and $\beta_{i,f}^{q,l,\mathcal{P}}$ is an unknown deterministic bias with a provided radius $b_{i,f}^{q,l,\mathcal{P}}$, such that:

$$\left| \beta_{i,f}^{q,l,\mathcal{P}} \right| \leq b_{i,f}^{q,l,\mathcal{P}}, \qquad (155)$$

with $b_{i,f}^{q,l,\mathcal{P}}$ given by either (102) or (140). Regarding the probabilistic side, we define:

$$\zeta_{i,f}^{q,l,\mathcal{P}} = \varepsilon_{i,f}^{q,l,\mathcal{P}} - \mu_{i,f}^{q,l,\mathcal{P}}, \qquad (156)$$

where:

$$\varepsilon_{i,f}^{q,l,\mathcal{P}} = \hat{v}_{i,f}^{q,l,\mathcal{P}} - v_{i,f}^{q,l,\mathcal{P}}, \qquad (157)$$

and $\mathbb{E}_{\boldsymbol{\pi}}\left( \zeta_{i,f}^{q,l,\mathcal{P}} \right) = 0$. Thus, the two-sided probabilistic half-width $p_{i,f}^{q,l,\mathcal{P}} \geq 0$ can be interpreted as:

$$\mathbb{P}_{\boldsymbol{\pi}} \left( \left| \varepsilon_{i,f}^{q,l,\mathcal{P}} - \mu_{i,f}^{q,l,\mathcal{P}} \right| \leq p_{i,f}^{q,l,\mathcal{P}} \right) \geq 1 - \delta^{\mathcal{P}} := c^{\mathcal{P}}, \qquad (158)$$

with $\delta^{\mathcal{P}}$ and the probabilistic bounds defined in (112) and (151). We abbreviate $c^{\mathcal{P}}$ as $c$ from now on. Consider $z_{c^*}$ the normal quantile at level $c^* = (1 + c)/2$, so we calibrate the half-widths scale to be:

$$\sigma_{i,f}^{q,l,\mathcal{P}} = \frac{c}{z_{c^*}}, \tag{159}$$

with tails still reported back as half-widths at the original confidence $c$.

We now propagate these errors from frame to utterance level, based on the aggregations we introduced in (45) and (49). On average, experiments showed that frames more than $g = 4$ apart are effectively independent both for speech and music mixtures. Given the set $\mathcal{F}^l$ of time frames with two or more active sources, the standard Bartlett block-decimation (Bartlett, 1946) yields the conservative inflation:

$$\text{std}\left(\frac{1}{\mathcal{F}^l}\sum_{f=1}^{\mathcal{F}^l}\zeta_{i,f}^{q,l,\mathcal{P}}\right) \leq \frac{\sqrt{g+1}}{\sqrt{\mathcal{F}^l}}\left(\frac{1}{\mathcal{F}^l}\sum_{f=1}^{\mathcal{F}^l}\left(\sigma_{i,f}^{q,l,\mathcal{P}}\right)^2\right)^{1/2}. \tag{160}$$

Let the radius error and the $p$-level probabilistic half-width obtained at the utterance-level using average pooling equal, respectively:

$$b_{i,\text{average}}^{q,l,\mathcal{P}} = \frac{1}{\mathcal{F}^l}\sum_{f=1}^{\mathcal{F}^l}b_{i,f}^{q,l,\mathcal{P}}, \tag{161}$$

$$h_{i,\text{average}}^{q,l,\mathcal{P}} = z_{c^*}\frac{\sqrt{g+1}}{\sqrt{\mathcal{F}^l}}\left(\frac{1}{\mathcal{F}^l}\sum_{f=1}^{\mathcal{F}^l}\left(\sigma_{i,f}^{q,l,\mathcal{P}}\right)^2\right)^{1/2}. \tag{162}$$

For the PESQ-like aggregation, let us denote its aggregation function from (49) as:

$$s(u) = 0.999 + 4\left(1 + \exp(-1.3669\,u + 3.8224)\right)^{-1}. \tag{163}$$

Let $W$ be the window and $H$ the hop of frame used for aggregation, then $M^l$ is the maximal number of possible windows. By norm submultiplicativity and the mean-value theorem (Horn & Johnson, 2012, Sec. 5.6):

$$b_{i,\text{pesq}}^{q,l,\mathcal{P}} = \frac{C_{\text{OL}}}{\sqrt{M^l}}\left(\frac{1}{\mathcal{F}^l}\sum_{f=1}^{\mathcal{F}^l}\left(b_f^{q,l,\mathcal{P}}\right)^2\right)^{1/2}\frac{\partial s}{\partial u}, \tag{164}$$

$$h_{i,\text{pesq}}^{q,l,\mathcal{P}} = z_{c^*}\frac{C_{\text{OL}}}{\sqrt{M^l}}\left(\frac{1}{\mathcal{F}^l}\sum_{f=1}^{\mathcal{F}^l}\left(\sigma_f^{q,l,\mathcal{P}}\right)^2\right)^{1/2}\frac{\partial s}{\partial u}, \tag{165}$$

where $C_{\text{OL}} = \lceil W/H \rceil$ and by construction $\partial s/\partial u \leq 1.3669$ when evaluated at point $u$.

To translate utterance-level errors to source-based PCC and SRCC values, let the integration of utterance-level MOS ratings from every system $q \in \{1, \ldots, Q\}$ be:

$$\mathbf{v}_{i,\text{MOS}}^l = \left(v_{i,\text{MOS}}^{1,l}, \ldots, v_{i,\text{MOS}}^{Q,l}\right), \tag{166}$$

and similarly, denoting $\hat{v}_{i,\mathcal{A}}^{q,l,\mathcal{P}}$ as the estimated aggregated measure across systems, where $\mathcal{A}$ is either average or PESQ-like aggregation (§B.4), then its integration is given by:

$$\hat{\mathbf{v}}_{i,\mathcal{A}}^{l,\mathcal{P}} = \left(\hat{v}_{i,\mathcal{A}}^{1,l,\mathcal{P}}, \ldots, \hat{v}_{i,\mathcal{A}}^{Q,l,\mathcal{P}}\right). \tag{167}$$

For every vector $\mathbf{v}$, we denote its centered version by $\tilde{\mathbf{v}}$. Let us denote the PCC value between an observation vector $\mathbf{v}$ and a MOS vector $\mathbf{m}$ as $r^{\text{PCC}}(\mathbf{v}, \mathbf{m})$, according to (53) and (54). Its gradient with respect to $\mathbf{v}$ at point $\hat{\mathbf{v}}_{i,\mathcal{A}}^{l,\mathcal{P}}$ is (Benesty et al., 2009):

$$\frac{\partial r^{\text{PCC}}}{\partial \mathbf{v}}\bigg|_{\mathbf{v}=\hat{\mathbf{v}}_{i,\mathcal{A}}^{l,\mathcal{P}}} = \frac{\mathbf{v}_{i,\text{MOS}}^l}{\left\|\hat{\mathbf{v}}_{i,\mathcal{A}}^{l,\mathcal{P}}\right\|_2\left\|\mathbf{v}_{i,\text{MOS}}^l\right\|_2} - \frac{r^{\text{PCC}}\left(\hat{\mathbf{v}}_{i,\mathcal{A}}^{l,\mathcal{P}}, \mathbf{v}_{i,\text{MOS}}^l\right)}{\left\|\hat{\mathbf{v}}_{i,\mathcal{A}}^{l,\mathcal{P}}\right\|_2^2}\hat{\mathbf{v}}_{i,\mathcal{A}}^{l,\mathcal{P}}. \tag{168}$$

Consider $\mathbf{b}_{i,\mathcal{A}}^{l,\mathcal{P}}$ the utterance-level bias radii from (161) or (164) across all systems:

$$\mathbf{b}_{i,\mathcal{A}}^{l,\mathcal{P}} = \left( b_{i,\mathcal{A}}^{1,l,\mathcal{P}}, \dots, b_{i,\mathcal{A}}^{Q,l,\mathcal{P}} \right). \tag{169}$$

Then, the induced PCC bias can be bounded by:

$$b_{i,\mathcal{A}}^{l,\mathcal{P},\text{PCC}} \leq \left\| \frac{\partial r^{\text{PCC}}}{\partial \hat{\mathbf{v}}_{i,\mathcal{A}}^{l,\mathcal{P}}} \right\|_2 \left\| \tilde{\mathbf{b}}_{i,\mathcal{A}}^{l,\mathcal{P}} \right\|_2. \tag{170}$$

For the probabilistic half-width, we model independent Gaussian jitters across systems with scales fixed by the utterance half-widths. Consider the $Q$-dimensional Gaussian vector:

$$\boldsymbol{\eta} \sim \mathcal{N}\left( \mathbf{0}, \text{diag}\left( \left( \frac{h_{i,\mathcal{A}}^{1,l,\mathcal{P}}}{z_{c^*}} \right)^2, \dots, \left( \frac{h_{i,\mathcal{A}}^{Q,l,\mathcal{P}}}{z_{c^*}} \right)^2 \right) \right), \tag{171}$$

with $\mathbf{0} \in \mathbb{R}^Q$. Using the delta method, first-order error propagation gives:

$$h_{i,\mathcal{A}}^{l,\mathcal{P},\text{PCC}} = z_{c^*} \sqrt{ \left( \frac{\partial r^{\text{PCC}}}{\partial \hat{\mathbf{v}}_{i,\mathcal{A}}^{l,\mathcal{P}}} \right)^T \text{diag}\left( \left( \frac{h_{i,\mathcal{A}}^{1,l,\mathcal{P}}}{z_{c^*}} \right)^2, \dots, \left( \frac{h_{i,\mathcal{A}}^{Q,l,\mathcal{P}}}{z_{c^*}} \right)^2 \right) \left( \frac{\partial r^{\text{PCC}}}{\partial \hat{\mathbf{v}}_{i,\mathcal{A}}^{l,\mathcal{P}}} \right) }. \tag{172}$$

Turning to the SRCC, let $\rho^{\text{SRCC}}(\cdot,\cdot)$ denote Spearman's rank correlation between two vectors (Kendall & Gibbons, 1990), as defined in (55) and (56). Because ranks are piecewise-constant, a safe deterministic error radius is obtained by checking the two extreme bias orientations:

$$b_{i,\mathcal{A}}^{l,\mathcal{P},\text{SRCC}} = \max\Big( \left| \rho^{\text{SRCC}}\big(\hat{\mathbf{v}}_{i,\mathcal{A}}^{l,\mathcal{P}} + \mathbf{b}_{i,\mathcal{A}}^{l,\mathcal{P}}, \mathbf{v}_{i,\text{MOS}}^l\big) - \rho^{\text{SRCC}}\big(\hat{\mathbf{v}}_{i,\mathcal{A}}^{l,\mathcal{P}}, \mathbf{v}_{i,\text{MOS}}^l\big) \right|, \tag{173}$$
$$\left| \rho^{\text{SRCC}}\big(\hat{\mathbf{v}}_{i,\mathcal{A}}^{l,\mathcal{P}} - \mathbf{b}_{i,\mathcal{A}}^{l,\mathcal{P}}, \mathbf{v}_{i,\text{MOS}}^l\big) - \rho^{\text{SRCC}}\big(\hat{\mathbf{v}}_{i,\mathcal{A}}^{l,\mathcal{P}}, \mathbf{v}_{i,\text{MOS}}^l\big) \right| \Big).$$

For the probabilistic half-width we jitter $\hat{\mathbf{v}}_{i,\mathcal{A}}^{l,\mathcal{P}}$ with the same independent Gaussian model in (171) and report the empirical $c^*$ quantile from Monte Carlo of the following:

$$h_{i,\mathcal{A}}^{l,\mathcal{P},\text{SRCC}} = \text{Quantile}_{c^*}\left( \left| \rho^{\text{SRCC}}\big(\hat{\mathbf{v}}_{i,\mathcal{A}}^{l,\mathcal{P}} + \boldsymbol{\eta}, \mathbf{v}_{i,\text{MOS}}^l\big) - \rho^{\text{SRCC}}\big(\hat{\mathbf{v}}_{i,\mathcal{A}}^{l,\mathcal{P}}, \mathbf{v}_{i,\text{MOS}}^l\big) \right| \right), \tag{174}$$

where we used $10^4$ draws for estimation, in the spirit of quantile bootstrap (Tibshirani & Efron, 1993).

Lastly, we consider the error propagation across all trials and their sources in a given scenario, e.g., English mixtures. Let $\mathcal{L}$ denote the number of trials in a scenario, and for each trial $l \in \{1, \dots, \mathcal{L}\}$, assume the number of total speakers in the trial is $N_{\max}^l$ (57). The values we report average across all $\mathcal{L}$ trials and $N_{\max}^l$ speakers, following (58)-(61).

The deterministic error radius of the PCC and SRCC per scenario are respectively given by:

$$b^{\text{PCC}} = \frac{1}{\sum_{l=1}^{\mathcal{L}} N_{\max}^l} \sum_{l=1}^{\mathcal{L}} \sum_{i=1}^{N_{\max}^l} b_{i,\mathcal{A}}^{l,\mathcal{P},\text{PCC}}, \tag{175}$$

$$b^{\text{SRCC}} = \frac{1}{\sum_{l=1}^{\mathcal{L}} N_{\max}^l} \sum_{l=1}^{\mathcal{L}} \sum_{i=1}^{N_{\max}^l} b_{i,\mathcal{A}}^{l,\mathcal{P},\text{SRCC}}. \tag{176}$$

To yield the probabilistic term, we assume that within any fixed trial $l$, the pairwise correlation between the source jitters has been empirically estimated and is denoted by $\rho_l$, while jitters from different trials are independent. This assumption holds by the construction of our trials in every

scenario. Consequently, the $c$-level probabilistic half-width on the scenario mean equals:

$$h^{\text{PCC}} = \tag{177}$$
$$z_{c^*} \sqrt{\frac{1}{\left(\sum_{l=1}^{\mathcal{L}} N_{\max}^l\right)^2} \sum_{l=1}^{\mathcal{L}} \left( \sum_{i=1}^{N_{\max}^l} \left( \frac{h_{i,\mathcal{A}}^{l,\mathcal{P},\text{PCC}}}{z_{c^*}} \right)^2 + 2\rho_l \sum_{\substack{i,j=1\\i<j}}^{N_{\max}^l} \left( \frac{h_{i,\mathcal{A}}^{l,\mathcal{P},\text{PCC}}}{z_{c^*}} \right) \left( \frac{h_{j,\mathcal{A}}^{l,\mathcal{P},\text{PCC}}}{z_{c^*}} \right) \right)}.$$

$$h^{\text{SRCC}} = \tag{178}$$
$$z_{c^*} \sqrt{\frac{1}{\left(\sum_{l=1}^{\mathcal{L}} N_{\max}^l\right)^2} \sum_{l=1}^{\mathcal{L}} \left( \sum_{i=1}^{N_{\max}^l} \left( \frac{h_{i,\mathcal{A}}^{l,\mathcal{P},\text{SRCC}}}{z_{c^*}} \right)^2 + 2\rho_l \sum_{\substack{i,j=1\\i<j}}^{N_{\max}^l} \left( \frac{h_{i,\mathcal{A}}^{l,\mathcal{P},\text{SRCC}}}{z_{c^*}} \right) \left( \frac{h_{j,\mathcal{A}}^{l,\mathcal{P},\text{SRCC}}}{z_{c^*}} \right) \right)}.$$

Ultimately, for each scenario and each measure $\mathcal{P}$ that uses aggregation technique $\mathcal{A}$, we report the deterministic envelope and probabilistic half-width $b^{\text{PCC}}$ and $h^{\text{PCC}}$ for PCC values and $b^{\text{SRCC}}$ and $h^{\text{SRCC}}$ for SRCC values.

## G  Further Discussions

### G.1  Limitations

Our validation depends exclusively on the SEBASS database, the only public corpus that provides human ratings for source separation systems, which limits the diversity in acoustic and linguistic traits that multiple dataset usually carry together. Moreover, the listening tests in SEBASS ask the human raters a generic quality question, rather than questions that isolate leakage versus self-distortion. This design choice may attenuate the ground-truth sensitivity to the specific error modes that PS and PM are intended to disentangle, and can introduce a systematic bias that even multi-rater averaging cannot fully cancel. Another noticeable limitation of this research concerns the aggregation techniques we employ to convert frame-level to utterance-level scores. Since neither granular human ratings exist nor is there any documented data-driven mapping from granular to global human ratings, we limit the capability of the PS and PM measures by merely approximating aggregation functions.

On a single NVIDIA A6000 GPU paired with 32 CPU cores with 64 GB of memory, our implementation achieves a real-time factor of 1.2, e.g., when analyzing a 25 ms frame in 30 ms on average. While this enables offline evaluation and hyper-parameter sweeps, it falls short of strict real-time monitoring and may limit large-scale neural-architecture searches and limit using the PS and PM measures inside loss function during training sessions. Profiling reveals that the dominant bottlenecks are diffusion-map eigendecompositions and repeated Mahalanobis distance computations with per-frame covariance estimation for all distortions points in every cluster. We plan to introduce more efficient implementations as we maintain our code repository.

We also point out that in music mixtures, 0.5% of frames exhibit for the PM measure an error radius that exceeds 1, rendering these observations irrelevant. These cases should be ignored completely, and future work that focuses on the separation of music sources will further investigate this phenomenon.

### G.2  Positioning Our Work as a Catalyst

The absence of large, diverse datasets annotated with fine-grained human scores remains a critical gap in source separation research. We argue that introducing perceptually grounded measures is precisely what enables this gap to be closed. By releasing PS and PM as open-source tools, we provide the community with a foundation on which richer benchmark datasets can be built, rather than waiting for such datasets to exist before new measures are introduced. Their availability can catalyze the creation of corpora that include human annotations at both frame-level and utterance-level resolutions. Such resources would support systematic, fine-grained comparisons between objective measures and human perception, stimulate the development of new evaluation metrics and systems, and allow researchers to study the relationship between granular and global ratings, an aspect currently reduced to heuristic

aggregation. In this way, PS and PM act as a gateway toward more rigorous and perceptually aligned evaluation standards in source separation.

## H LLM USAGE

We used a large language model (LLM) as a general-purpose assistant in three ways:

1. Language polishing to improve clarity. Every word was read and proofed by the authors.
2. Exploration of literature. All cited literature was validated by the authors.
3. Coding assistance. All code was reviewed, rewritten as needed, and tested by the authors before use.

We did not delegate authorship decisions or scientific claims to the LLM. We manually verified all content, checked citations, and validated all results. No confidential or identity-revealing information was provided to the LLM, and use complied with dataset licenses and the ICLR Code of Ethics. We disclose this usage here as recommended by ICLR-2026. We also disclose LLM usage in the submission form. The authors take full responsibility for the submission.

