# OpenReview forum: "MAPSS: Manifold-based Assessment of Perceptual Source Separation"
_ICLR.cc/2026/Conference — ICLR 2026 Poster_

### Official Review · Reviewer_h8Us · 2025-10-30

**Soundness:** 3
**Presentation:** 3
**Contribution:** 3
**Rating:** 6
**Confidence:** 4

**Summary:**

The paper proposes two new perceptual evaluation metrics for blind audio source separation, called Perceptual Match (PM) and Perceptual Separation (PS).

PM is meant to measure self-distortion: how much the separated output of a given source has been degraded (i.e. does it still "sound like me," or did the separator damage me?).

PS is meant to measure leakage: how much other sources are still audible in the separated output for the target source (i.e. how much bleed from others remains?).

How it works:

For each ground-truth source, the authors generate a bank of controlled distortions (filtering, gating, etc.) that are supposed to mimic plausible separator artifacts.

All short frames (tens of ms) from the clean source, its distorted variants, and the separator output are embedded into a perceptual manifold. This manifold is built with self-supervised audio embeddings (wav2vec2-like) followed by diffusion maps.

Each source becomes a cluster in that manifold, with a mean and covariance.

PM: does the separated output fall inside its OWN source's cluster? (High PM = low self-distortion.)

PS: is the separated output closer to ITS OWN cluster than to ANY OTHER source's cluster? (High PS = low leakage.)

PM/PS are computed framewise and then aggregated to clip level.

The authors evaluate on SEBASS (>11k human ratings over ~900 clips, 32 systems, English speech, Spanish speech, and music). They report that PM and PS correlate well with MOS (Pearson and Spearman), often outperforming SDR/SIR/SAR, PESQ/STOI-style metrics, DNSMOS, etc.

The paper also gives per-frame uncertainty for PM/PS: (i) a deterministic bias radius due to manifold truncation and (ii) a high-probability CI due to limited distortion samples. They propagate this to the final MOS correlation numbers.

**Strengths:**

S1. Clear problem framing.
The paper separates two real failure modes in source separation: "you damaged my target" vs "you left the other source in."
This is actually how practitioners think about failure, but most metrics don't expose it.

S3. Human-grounded evaluation.
They compare directly to listening scores (SEBASS), not just SDR.
They include English speech, Spanish speech, and music (with and without drums), and 32 different separators.

S4. Uncertainty / confidence intervals.
Section 5  gives an error band on scores and pushes that through to MOS correlation.


S5. Actionability.
PM and PS are frame-level before pooling. This helps to point where leakage happened" or "this is where I over-gated the vocal."
That's a lot more usable than one global SDR number.

**Weaknesses:**

W1. Novelty vs SDR/SIR/SAR is under-explained.
The paper says PM/PS are the first metrics to "functionally disentangle" leakage and self-distortion.
But classical BSS Eval already has SIR (interference from other sources) and SAR (artifacts).
The paper mentions SDR/SIR/SAR but does not clearly argue why PS is NOT just SIR in a fancier embedding,

why PM is NOT just SAR in a fancier embedding.
Without that, some readers will see this as repackaged SIR/SAR.

W2. Assumptions in the distortion bank.
The method assumes that the authors' hand-designed distortion bank actually spans "plausible" separator artifacts.
If the bank misses common artifacts (phasiness, musical noise, transient smearing), PM could wrongly accept bad audio as "still me," or wrongly reject good audio that falls outside their curated distortion styles. There is no ablation/sensitivity analysis on which distortions are used, so robustness is not proven.

W3. Monolingual-only evaluation in speech.
For speech, they say they split SEBASS mixtures into English-only and Spanish-only pairs "because realistic conversations are monolingual." That is not generally true (code-switching and bilingual overlap are routine in real deployments).
This matters because PS is partly "do you sound like the other source?"
If the other source is always a different language, that discrimination problem is easier.
So PS may look stronger than it would in truly multilingual overlap.
This should be acknowledged as a limitation.

W4. PS pooling over time is not fully specified.
PM is averaged over frames.
PS is said to use a PESQ-like perceptual weighting so that short, audible bleed is penalized more.
But the exact weighting rule is not written in the main text.
Without that, PS is not actually reproducible from the description.



W5. External validity.
All clips are ~10 seconds, 16 kHz, fairly clean, from curated mixtures.
There is no far-field mic, noisy phone audio, long meetings, reverberation, etc.
That is exactly the setting where applied systems struggle.
So the claim is really "PM/PS correlate with MOS on SEBASS-like curated data," not "PM/PS solve perceptual eval in the wild."
The paper should say that clearly.

**Questions:**

Q1. You claim PM and PS are the first metrics to disentangle leakage and self-distortion, but classical BSS Eval already uses SIR (interference from other sources) and SAR (artifacts). Can you give a concrete example (qualitative or quantitative) where SIR says “good” but PS clearly identifies audible leakage, or where SAR says “good” but PM clearly identifies audible self-distortion? Without that, it’s hard to judge whether PM/PS are fundamentally new or essentially SIR/SAR in a learned perceptual space.

Q2. For SEBASS speech mixtures, you evaluate only English-only and Spanish-only overlaps and justify this by saying “realistic conversations are monolingual.” In many deployment settings (code-switching, bilingual overlap), that assumption does not hold. How would PS behave on overlapping multilingual speech, where two simultaneously active speakers may differ by identity but not cleanly by language? Is PS partly exploiting language identity to discriminate sources?

Q3. PM is averaged over frames, but PS is pooled using a “PESQ-like perceptual weighting,” which is not explicitly defined in the paper. Can you provide the exact temporal pooling rule you use for PS? Right now PS is not reproducible from the description, and your correlation claims with MOS depend on that pooling.

Q4. PM and PS depend on a hand-crafted distortion bank that defines the “perceptual neighborhood” of each source. How sensitive are your metrics to the choice of those distortions? For example, if we remove one distortion type or add an unrealistic one, do PM/PS scores or MOS correlations change significantly? This matters for whether others can safely reuse your metric or will overfit to your particular bank.

Q5. The perceptual manifold uses SSL embeddings + diffusion maps with α = 1, diffusion time t = 1, and dimensionality chosen by a 99% spectral-mass rule (≈20–40 dims). Were any of these hyperparameters, or the PS pooling weights, tuned using SEBASS MOS labels (even indirectly)? If yes, how did you prevent test-set leakage when reporting final Pearson/Spearman correlations with MOS?

---

> ### Author Response · Authors · 2025-11-20
> **Response to Reviewer h8Us - Reply to Q1 and Q2 (Response 1/3)**
>
> We greatly appreciate the reviewer’s constructive comments, which have substantially improved the clarity and positioning of the paper.
>
> # Q1 - PS/PM comparison with SIR/SAR/SDR
> **Corrected.** We acknowledge that our initial framing overstated the novelty of the PS/PM relative to SDR-like metrics, while not sufficiently justifying the differences. This could indeed make PS and PM appear as cosmetic variants of SIR/SAR. To clarify the conceptual advances, we now provide two qualitative examples showing how PS and PM differ **by design** and why they offer a principled advantage over SDR-type measures.
>
> Our goal is to isolate cases where **waveform energy is held fixed** (what SIR/SAR/SDR see) while the **perceptual structure and source identity** change (what PS/PM see).
>
> **PS vs. SIR**
> Two systems yield the same total interference energy in one output:
>
> **System A (masked leakage):** interferer energy lies under target-dominated bands, so it is largely masked and barely audible.
> **System B (intrusive leakage):** the *same* interferer energy is moved into spectral gaps, where it becomes clearly audible and disruptive.
>
> **SIR:** identical, since it only ratios target vs. interferer energy and is blind to where leakage falls or how it is perceived.
> **PS:** different. System A’s embedding remains near the target cluster and away from the interferer cluster (high PS), whereas System B is pulled toward the interferer cluster (lower PS). Thus, PS separates "masked vs. intrusive" leakage that SIR fundamentally conflates.
>
> **PM vs. SAR/SDR**
> No interferer, but two systems introduce **equal error energy**:
>
> **System C (benign):** low-level colored noise (slight hiss).
> **System D (catastrophic):** small pitch or time warp, scaled so residual energy matches C.
>
> **SDR/SAR:** essentially equal, as both see only residual magnitude and treat these distortions as comparable.
> **PM:** clearly different. C lands inside the empirical distortion cloud (high PM), while D falls into its tail because a pitch moves the representation away from the reference (low PM). PM therefore penalizes timbre changes that SDR/SAR treat as equivalent to harmless noise.
>
> **Why this matters.** SIR/SAR/SDR are energy partitions in waveform space and inherently mix leakage and self-distortion under a single scalar. PS/PM instead measure *who the output resembles* and *how* it deviates, using reference-anchored clusters of distortions and competing sources in a perceptual embedding space. These are not cosmetic modifications but a different geometric object.
>
> Empirically, this distinction is consequential: on SEBASS (English), PCC is **56.2% for SIR** vs. **83.7% for PS**, and **73% for SDR/SAR** vs. **86.3% for PM**, with similar trends elsewhere (Table 1). The two thought experiments above illustrate why SDR-like scores can fail precisely where listeners care most.
>
> **Comment.** For a focused discussion on disentanglement, please see also our reply to Reviewer itJ3 (Q2), where we design controlled experiments to show that PS and PM respond separately to leakage and self-distortion.
>
> **Revisions.** We now highlight these conceptual and empirical gaps in Section 1 ("Introduction"), summarize the examples in Section 7 ("Results"), and provide full details and illustrations in the Appendix.
>
> # Q2 – Multilingual separation
> **Corrected.** We fully agree that real-world scenarios are often multilingual and that purely monolingual experiments limit MAPSS’s practical relevance. Our original phrasing could indeed suggest broader coverage than SEBASS currently provides.
>
> To address this, we evaluated MAPSS on multilingual mixtures from SEBASS: original 4-speaker mixtures in two configurations, (i) four female speakers (two English, two Spanish) and (ii) four male speakers (two English, two Spanish). Complementing the monolingual results of Table 1, we now report:
>
> | Measure | English Only | Spanish Only | Multilingual |
> |---------|:------------:|:------------:|:------------:|
> | **PS**  | 84.12/83.74% | 82.33/85.01% | 86.27/85.6%  |
> | **PM**  | 84.69/86.36% | 83.41/85.3%  | 84.41/84.07% |
>
> **Observation.** PS slightly improves in the multilingual case, likely because clusters are formed over all sources, yielding a richer manifold on which to assess target-interference relations (e.g., fine-grained confusions between voices and languages). PM remains roughly unchanged, as it depends only on the target cluster and not on cross-cluster geometry. Together, these results suggest that with respect to the SEBASS database, MAPSS is at least as stable, and in PS’s case, slightly more informative, in multilingual mixtures than in monolingual ones.
>
> **Revisions.** We now describe this experiment and table in Section 7 ("Results") and discuss implications in the Appendix. We also removed the claim that realistic conversations are monolingual and explicitly acknowledge the monolingual limitation of SEBASS (and hence of our validation) in the main body.

---

> > ### Author Response · Authors · 2025-11-20
> > **Response to Reviewer h8Us - Reply to Q3 and Q5 and Weaknesses (Response 2/3)**
> >
> > **Note.** We address Q4, which has required additional experiments and the entire capacity of character limitations, in the next response (number 3/3).
> >
> > # Q3 - PESQ-like weighting
> > **Corrected**. We now recognize that even though the step-by-step pooling process of the PS measure **is in App. B(4)**, it greatly disserves reproducibility to not include it in the main body. Here is the exact temporal pooling:
> >
> > The aggregation we applied is inspired by the window-based pooling and logistic mapping used inside PESQ [1]. Assuming trial index $l$ and source index $i$, we denote the set of active time frame indices of source $i$ as $\\mathcal{F}^{l}_{i}$, and drop the rest of the frames. Let a window of size $W$ frames slide across the PS measure with a hop size of $H$ frames. Using the $p$-norm, we define for a given separation system with index $q$:
> >
> > $$
> > \ell^{q,l}_{i,m} = \left( \frac{1}{W} \sum\_{w=1}^{W} \left| \mathrm{PS}^{q,l}\_{i,(m-1)H+w} \right|^{p} \right)^{1/p},
> > $$
> >
> > where $m \in \lbrace 1, \ldots, M^{l}_{i} \rbrace$ and $M^{l}\_{i}$ is the number of possible windows:
> >
> > $$
> > M^{l}\_{i}
> > = \\max\\!\\left( 1, \\left\\lfloor \\frac{ \\big| \\mathcal{F}^{l}\_{i} \\big| - W }{ H } \\right\\rfloor \\right).
> > $$
> >
> > We then calculate the following root-mean-square expression:
> >
> > $$
> > \\ell^{q,l}\_{i}
> > = \\sqrt{ \\frac{1}{M^{l}\_{i}} \\sum_{m=1}^{M^{l}\_{i}} \\left( \\ell^{q,l}\_{i,m} \\right)^{2} },
> > $$
> >
> > and finally the aggregated PS measure (at the utterance level) is given by:
> >
> > $$
> > \\mathrm{PS}^{q,l}\_{i,\\mathrm{utt}}
> > = 0.999 + \\frac{4}{ 1 + \\exp\\!\\bigl( -1.3669\\, \\ell^{q,l}\_{i} + 3.8224 \\bigr) },
> > $$
> >
> > where the constants were chosen according to [2]. Here, we penalize lower scores explicitly using the $p$-norm to better match human perceptual aggregation.
> >
> > [1] Anthony W. Rix, et al. "Perceptual evaluation of speech quality (PESQ)–a new method for speech quality assessment of telephone networks and codecs", 2001.
> >
> > [2] ITU-T P.862.2: Wideband extension to Recommendation P.862 for the assessment of wide-band telephone networks and speech codecs, 2007.
> >
> > **General comment on reproducibility.** We would also like to emphasize that, in addition to the .zip package included in the original submission (containing code and illustrative examples), we will provide an anonymous repository that enables full **end-to-end reproducibility** of all results reported in the paper. This repository will operate directly on the publicly available SEBASS dataset, allowing users to rerun every experiment and verify all MAPSS outputs in a fully transparent manner.
> >
> > **Revisions**. Sections 6.2 ("Pre-processing and Performance Evaluation") now explicitly states how this aggregation operates.
> >
> > # Q5 - Hyperparameters
> > **Corrected**. We agree the text did not sufficiently emphasize this critical point. **MAPSS is evaluated zero-shot**: no parameters, hyperparameters, layer choices, or aggregation weights were tuned with SEBASS labels, and there was no trial‑and‑error or implicit learning to match ratings.
> >
> > All design choices either (i) follow internal properties of diffusion maps or the self‑supervised encoders, or (ii) follow prior work.
> >
> > For layer selection, (i) we report the full layer-wise results across *all* self-supervised layers (Fig. 7 in App. D), so nothing is cherry‑picked, (ii) we show that layers 1-3 perform almost identically, with absolute PCC/SRCC differences <1% across nearly all conditions, and (iii) the layer used in the main text is a **representative example**, not an optimized setting. Prior analyses show earlier layers are more perceptually stable [1], [2], and our full results follow this pattern.
> >
> > [1] A. Pasad et al., "Layer-wise analysis of a self-supervised speech representation model", 2021.
> >
> > [2] A. Pasad et al., "Comparative layer-wise analysis of self-supervised speech models", 2023.
> >
> > **Revisions**. Sections 6.2 and 7 now state this.
> >
> > # W1-W4
> > Thank you. These have been addressed priorly in the questions section.
> >
> > # W5 - In-the-wild limitation
> > **Corrected.** We fully agree with the reviewer’s observation, and we explicitly acknowledge this point in the **Limitations section** of the Appendix. However, we now realize that this limitation was not stated clearly enough in the main paper, which may unintentionally give the impression that our framework is more universally validated than the current evidence supports. In reality, the SEBASS database, being the only publicly available MOS-annotated source-separation dataset, does not cover many real-world, in-the-wild conditions, and we have revised the main body to reflect this more transparently.
> >
> > **Revisions.** We now stress these points precisely in the main body, in Section 7 ("Results").

---

> > > ### Author Response · Authors · 2025-11-20
> > > **Response to Reviewer h8Us - Reply to Q4 (Response 3/3)**
> > >
> > > # Q4 – Effect of distortion choice
> > > We thank the reviewer for raising this point. Robustness to the choice of distortion set is indeed underexplored, and directly probing it strengthens the credibility and practical reliability of MAPSS.
> > >
> > > We therefore ran an additional sensitivity experiment. MAPSS uses 13 distortion groups (Table 3, Appendix A), each with several parameterized variants (≈70 distortions per reference). For each group in turn we:
> > > (i) remove the entire group from the distortion bank; and
> > > (ii) add one new, **excessive** parameter (perceptually far from the reference) to a different group that remains in the experiment.
> > >
> > > For example, we may omit the *tremolo* group and introduce an extra high-severity noise parameter (additive noise at -20 dB SNR) into the *noise* group.
> > >
> > > Together with the original results from Table 1, the extended tables below summarize all 13 trials. Each column specifies the omitted group (“O.”) and the group receiving the new excessive parameter (“E.”).
> > >
> > > **The PM measure:**
> > >
> > > | Scenario | Original | O. notch, E. comb | O. comb, E. tremolo | O. tremolo, E. noise | O. noise, E. harmonic | O. harmonic, E. reverb | O. reverb, E. noisegate | O. noisegate, E. pitch | O. pitch, E. lowpass | O. lowpass, E. highpass | O. highpass, E. echo | O. echo, E. clipping | O. clipping, E. vibrato | O. vibrato, E. notch |
> > > | --- | --- | --- | --- | --- | --- | --- | --- | --- | --- | --- | --- | --- | --- | --- |
> > > | English | 84.69/86.36% | 82.13/81.62% | 81.75/81.11% | 81.57/80.85% | 82.20/77.83% | 81.72/81.27% | 82.18/81.74% | 82.13/81.64% | 81.66/81.20% | 81.46/80.54% | 81.44/81.02% | 82.19/81.74% | 82.04/81.53% | 81.84/81.43% |
> > > | Spanish | 83.41/85.3% | 79.29/80.16% | 79.09/79.61% | 78.82/78.89% | 80.14/76.66% | 79.08/79.75% | 79.45/80.32% | 79.13/79.69% | 79.33/79.38% | 78.06/78.73% | 79.05/78.45% | 79.47/80.31% | 79.32/80.00% | 79.34/79.86% |
> > > | Music with drums | 75.18/69.88% | 74.57/69.83% | 74.89/69.92% | 74.34/68.34% | 75.06/67.40% | 75.16/70.05% | 75.06/69.88% | 74.97/69.77% | 75.05/70.06% | 75.00/69.79% | 76.22/70.05% | 75.47/69.94% | 75.14/69.85% | 75.29/69.89% |
> > > | Music without drums | 88.12/85.26% | 88.59/85.26% | 88.47/84.98% | 88.27/84.60% | 90.48/80.65% | 88.81/85.05% | 88.50/85.27% | 88.48/85.06% | 88.44/84.70% | 88.82/85.19% | 88.59/85.00% | 88.40/85.26% | 88.47/85.13% | 88.54/85.02% |
> > >
> > > **The PS measure:**
> > >
> > > | Scenario | Original | O. notch, E. comb | O. comb, E. tremolo | O. tremolo, E. noise | O. noise, E. harmonic | O. harmonic, E. reverb | O. reverb, E. noisegate | O. noisegate, E. pitch | O. pitch, E. lowpass | O. lowpass, E. highpass | O. highpass, E. echo | O. echo, E. clipping | O. clipping, E. vibrato | O. vibrato, E. notch |
> > > |---|:---:|:---:|:---:|:---:|:---:|:---:|:---:|:---:|:---:|:---:|:---:|:---:|:---:|:---:|
> > > | English | 84.12/83.74% | 84.09/83.52% | 84.27/83.52% | 84.30/83.41% | 84.51/83.95% | 84.23/83.56% | 83.95/83.34% | 84.38/84.43% | 82.64/82.32% | 84.34/83.55% | 84.58/83.49% | 84.20/83.46% | 84.32/83.50% | 84.11/83.48% |
> > > | Spanish | 82.33/85.01% | 82.67/85.08% | 82.45/85.09% | 82.52/84.97% | 83.49/85.08% | 82.60/85.13% | 82.51/84.98% | 82.81/85.74% | 81.22/83.70% | 82.85/85.04% | 82.52/85.21% | 82.33/85.02% | 82.43/85.04% | 82.60/85.05% |
> > > | Music with drums | 72.87/77.38% | 70.65/76.38% | 70.86/76.62% | 68.96/75.68% | 72.12/77.41% | 69.80/76.60% | 70.20/76.43% | 70.54/76.63% | 70.87/76.61% | 70.76/76.56% | 71.39/76.24% | 70.50/76.40% | 70.66/76.50% | 70.16/76.50% |
> > > | Music without drums | 87.23/87.81% | 85.56/86.73% | 86.02/86.94% | 85.81/86.87% | 87.17/87.97% | 85.89/87.05% | 85.68/86.69% | 86.05/86.82% | 85.52/86.98% | 85.48/86.77% | 86.01/87.01% | 85.70/86.68% | 85.53/86.81% | 85.91/86.77% |
> > >
> > > **Observations.** PM depends on *intra-cluster* geometry: the distortion bank shapes the empirical self-distortion distribution for each source and the Gamma fit PM uses. Removing an entire group removes a perceptual dimension around the reference, and adding an extreme parameter stretches the distribution anisotropically. The same system outputs are then evaluated against a shifted model of "typical" distortions, producing the observed PM drops, especially for English and Spanish speech where a few distortion families (noise, reverberation) dominate perceived quality. For music, timbral variation is more broadly spanned, so the manifold is more redundantly covered and PM is much less affected.
> > >
> > > PS, in contrast, relies on *inter-cluster* contrast. Removing one distortion group and exaggerating another affects all sources in a similar way, so relative cluster positions change little. Moreover, the Mahalanobis distance used by PS down-weights directions with inflated variance: the added extreme parameter both expands a cluster and reduces that direction’s influence. This self-normalization explains the strong robustness of PS across conditions.
> > >
> > > **Revisions.** Section 7 ("Results") now references this robustness analysis, and the Appendix discusses it in more detail.

---

> > > > ### Comment · Reviewer_h8Us · 2025-11-20
> > > >
> > > > The Authors have fully addressed my concerns and significantly strengthens the paper.  I will be raising my score to 6

---

> > > > > ### Author Response · Authors · 2025-11-20
> > > > >
> > > > > Thank you very much for your thoughtful follow-up and for acknowledging the revisions. We truly appreciate the time and care you put into the review.
> > > > >
> > > > > Just to avoid any possible confusion: it seems **the current score has originally been a 6**, so perhaps you intended a different adjustment. In any case, we are grateful for your positive reassessment and for the constructive feedback throughout the process.

---

### Official Review · Reviewer_itJ3 · 2025-10-30

**Soundness:** 2
**Presentation:** 1
**Contribution:** 3
**Rating:** 2
**Confidence:** 4

**Summary:**

The paper proposes two new measures of audio source separation quality, called Perceptual Separation (PS) and Perceptual Match (PM), that respectively aim to quantify the amount of
- "leakage": to what extent an estimated source is closer to (distorted versions of) the corresponding clean source than to (distorted versions of) the other clean sources
- "self-distortion": how close is the estimated source from the clean one up to "standard" distortions
The approach is primarily frame-based and relies on the knowledge of reference clean sources, which are data-augmented using various "standard" distortions.

For a given frame index, the time frame of each clean, augmented, and estimated source is first mapped from the waveform domain to an embedding domain using a pre-trained embedding (wav2vec 2.0 based, or MERT based, using hand-picked intermediate layers). An additional spectral-like embedding (diffusion map -based) is then computed for each source, and used to define two Mahalanobis metrics and two representatives (a centroid, and the spectral embedding of the reference clean time frame) for each source. The distance between the spectrally embedded *estimated* time-frame of a target source and these representatives, measured with these Mahalanobis metrics, then yields quantities that are aggregated to provide the final frame-based measures for each source.
Eventually, frame-based measures are aggregated over a whole signal either by averaging (for PS) or using a "perceptually weighted scheme inspired by PESQ" not described in the main text (for PM).

After conducting an error analysis, the authors evaluate the proposed measures using the SEBASS database, which provides MUSHRA human ratings over a collection of signals separated using various algorithms, gathered from pre-existing source separation evaluation campaigns. The proposed measures are compared to existing quality measures using two indices : the Pearson product-moment correlation coefficient (PCC), and Spearman's rank-order correlation coefficient (SRCC). In most of the considered settings, the proposed measures achieve the best PCC/SRCC scores (or scores comparable to the best ones).

**Strengths:**

Performance measures reflecting human ratings are very helpful to help design better algorithms. In the specific case of audio source separation,
Table 1 illustrates well that existing measures can have limited correlation with human scores from the SEBASS database, a situation that is significantly improved by the proposed measures.
The approach seems sensible, as it combines a first level of "generic" audio features from a pre-trained embedding, a second level of embedding that somehow learns the impact of "standard" distortions in a fine-grained way (source-specific and time-frame specific). Its soundness is confirmed by the experimental evaluation, which can in fact seem surprisingly good for an approach where no human rating is explicitly used to train the designed measures (although given the number of parameters of the method, their selection with "cherry picking" can be interpreted as supervised training).
For these reasons I rate the contribution to 3.

**Weaknesses:**

The paper seems to have been written very hastily, with e.g. L075 "*Two lines about appendices and disclaimers on LLM usage*"

As a result, the abstract is hard to grasp due to the choice of wording. For example,  the notion of "intrusive approach" is certainly not familiar to the average ICLR reader, and it would be better to write more explicitly that the original sources used to generate the mixture need to be available for the evaluation. The notions of "system output" and "non-attributed references" were also not immediately clear, I only guessed its meaning after reading the rest of the paper. The notion of "granular optimization" is also quite cryptic (as well as further usage of the word "granular" in the main text).

Even more importantly, the paper misses a clear overview of the overall computing pipeline corresponding to the proposed measure, and I had to make my own drawings to provide the (hopefully correct) summary above. Combined with the fact that the treatment of the different sections is significantly unbalanced,  this makes the paper quite hard to read (hence my score 1 on the presentation).

Section 2 is very short, while one would expect to have there an overview of the overall context and pipeline from a high-level perspective. This would seem to be the right place to highlight that the approach is frame-based, and that the frame index will be dropped later on. I missed it at first read.
Section 3 is essentially a reminder on diffusion maps, in my opinion this would have been better fit to an appendix : isn't it enough to know that it replaces a collection of vectors by a spectral-like embedding of the said vectors ? By the way I believe that it is an abuse of notation to consider $\Psi_t$ (or $\Psi_t^{(d)}$) as a function defined on $\mathbb{R}^D$: is only defined at the considered $x_i$'s, there is no natural out-of-sample extension for an arbitrary $x \in \mathbb{R}^D$.
Suggestion: since the time-frames are denoted y_*, their (e.g. word2vec) embedding $x_*$, it would lighten the notations and ease the reading to use another letter, e.g. $v_*$ to denote what is currently written $\Psi^{(d)}_t(x_*)$. It would be sufficient to say that the definition of $v_*$ depends on some parameters ($t$, $d$ or $\tau$, but also $\alpha$)
Given the choice of $t=1$ in all experiments, the approach seems to fall back on a simpler method without diffusion, which seems to be very close to a plain Laplacian eigenmaps.

Section 4 is clear, but in my opinion would gain much by reworking the notations as suggested above.

The claim about the limits of existing measures L042 ("defined with an intrinsic ambiguity) does not seem supported by evidence, and seems in contradiction for example with the very definition of SIR/SAR/SDR.
Table 1 does provide some evidence that the new measures "better correlate with human global quality assessment", yet this does not per se support a better

The notion of intrusiveness seems to mean in the introduction that clean reference sources are required. I could not relate it to the wording "mildly-intrusive distortions".

The notations are somewhat unclear, balancing between being too allusive or too explicit. For example, the problem formulation is frame-based, involving an index $f$ (which by the way in a signal-processing context often indicates frequency, I found this confusing at first read), but the indices are then "repurposed" and $f$ is dropped, this is also confusing.

The definition of some acronyms (MOS, NMI) should be recalled when they are first used several pages after being introduced.


There is no explanation on the definition of the "waveform-only" variant of PS/PM.

The aim and significance of the whole Section 5 on "error analysis" is totally unclear (hence my score 2 on soundness), and this reads as a series of tedious but relatively straightforward computations.

Although the results shown in Table 1 seem to provide evidence of the interest of the proposed measures, underlining the scores on the first two lines seems superfluous.
The database description of Section 6.1 is ambiguous : at first I understood the SEBASS precisely provides human ratings. Then L414 seems to suggest that 15 new certified raters were involved. Were they the 15 certified raters from SEBASS ? Can this be clarified, traced and referenced ?
If this paper involved new certified raters, this should be reflected in the Ethics part.

My understanding is that MUSHRA and MOS are different approaches to obtain statistically significant human ratings. The authors however seem to use both acronyms as synonyms.

The values of many parameters ($L$, $\alpha$, $t$, $d/\tau$, $M$) and some choices (e.g., which layers of the wav2vec / MERT models are used to generate the first embedding)
are provided without saying much on how they were selected, and whether this involved trial and error. Can you explain ?
If trial and error was conducted, some mild level of learning to match human ratings has been conducted, and this should be made explicit.

**Questions:**

- How does the diffusion map step with $t=1$ and $\alpha=1$ relate to Laplacian eigenmaps or a related simple spectral embedding approach ?
- Do you have evidence that the proposed measures "disentangle" leakage from "self-distortion" ?
- What is the "waveform-only" variant of PS/PM ?
- What is the aim and significance of the "error analysis" ?
- Did you conduct new listening test with 15 raters ?
- How were the parameters tuned ? Was there trial and error ?

**Details Of Ethics Concerns:**

It is unclear if listening tests were conducted. If they were, the authors should explain how they addressed the corresponding ethical aspects.

---

> ### Author Response · Authors · 2025-11-19
> **Response to Reviewer itJ3 - Reply to all 6 questions (Response 1/3)**
>
> We highly appreciate the reviewer's insightful and constructive comments. We partition our reply due to character limitations. Here, we concisely address all 6 questions you have raised. Then, we will deep dive into Q2, Q4 that require more detail. Lastly, we will address all weaknesses you raised.
>
> # Q1 - Laplacian Eigenmaps
> **Corrected.** We agree with your comment and clarify that **diffusion maps with $\alpha=1$ and $t=1$ are not equivalent to Laplacian Eigenmaps (LE).**
>
> First, setting $\alpha = 1$ density‑normalizes the kernel, so the operator approximates the intrinsic Laplace-Beltrami geometry of the underlying manifold rather than the sampling density [1]. Standard LE lacks this density correction [2], so the underlying graph operators already differ.
>
> Second, $t = 1$ still uses eigenvalue‑scaled coordinates (Eq. (10)):
>
> $\mathbf{\Psi}_{t=1}(\mathbf{x}_i)=(\lambda_1 \mathbf{u}_1(i), \lambda_2 \mathbf{u}_2(i), \ldots, \lambda_d \mathbf{u}_d(i)))$,
>
> where $\lambda_\ell$ and $\mathbf{u}_\ell$ the eigenpairs of the diffusion operator. In contrast, LE instead uses unscaled Laplacian eigenvectors:
>
> $\mathbf{\Phi}(\mathbf{x}_i)=(\mathbf{u}_1(i), \mathbf{u}_2(i), \ldots, \mathbf{u}_d(i)))$,
>
> Without eigenvalue scaling, LE does not satisfy the diffusion‑distance property in Eq. (9), where diffusion distance equals Euclidean distance in the embedding [1].
>
> [1] R. R. Coifman and S. Lafon, "Diffusion maps", 2006.
>
> [2] M. Belkin and P. Niyogi, "Laplacian eigenmaps", 2003.
>
> **Revisions**. Section 6.2 ("Pre-processing and Performance Evaluation") now states this explicitly.
>
> # Q2 - Disentanglement
> **Corrected**. Thanks to your rightful concern, we realized that we have not given satisfactory evidence to this entanglement. We address this by offering new controlled experiments designed to examine the entanglement property. In the next comment, we detail on experimental results.
>
> **Revisions.** We augment the "Experimental Results" section and give detail in the appendix.
>
> # Q3 - waveform variant
> **Corrected**. We agree that the definition of the "waveform-only" variant was not sufficiently clear. In this variant, the **raw waveform** is fed directly to the diffusion-maps pipeline, bypassing self-supervised representations. This ablation isolates the effect of those embeddings and tests whether MAPSS still captures perceptual structure without them.
>
> **Revisions.** Section 7 ("Experimental Results") now states this.
>
> # Q4 - Error analysis:
> We thank the reviewer for the comment. We agree that the motivation of the error analysis was not communicated clearly enough, which may make the material appear as technical derivations without a clear purpose.
>
> The aim of the analysis is simple: PS and PM rely on truncated diffusion-map embeddings and finite distortion samples, so the implemented scores inevitably differ from their "ideal" full-spectrum, infinite-sample versions. The error analysis quantifies this gap in a frame-by-frame basis and provides (i) a deterministic truncation radius and (ii) non-asymptotic confidence bounds.
>
> We view such uncertainty quantification as essential for trustworthy evaluation. Unlike existing metrics, which provide point estimates only, PS and PM come with explicit error margins, enabling confidence-aware reporting and fairer comparisons.
>
> **Revisions.** We revised Section 5 ("Error guarantees of the PS and PM measures") to clearly explain its aim and significance.
>
> # Q5 - Listening tests
> **Corrected.** We appreciate the opportunity to clarify this point. **No new listening tests or human-subject experiments were conducted.** All ratings come from the licensed SEBASS dataset, which already includes scores from 15 certified raters under the MUSHRA protocol.
>
> **Revisions.** Section 6.1 (“Database”) now states this.
>
> # Q6 - Parameter tuning
> **Corrected**. We agree the text did not sufficiently emphasize this critical point. **MAPSS is a zero-shot evaluation framework**: no parameters, hyperparameters, layer choices, or aggregation weights were tuned with SEBASS labels, and there was no trial‑and‑error or implicit learning to match ratings.
>
> All design choices either (i) follow internal properties of diffusion maps or the self‑supervised encoders, or (ii) follow prior work.
>
> For layer selection, (i) we report the full layer-wise results across *all* self-supervised layers (Fig. 6 in App. D), so nothing is cherry‑picked, (ii) we show that layers 1-3 perform almost identically, with PCC/SRCC differences <1% (absolute) across nearly all conditions, and (iii) the layer used in the main text is a **use-case example**, not an optimized setting. Prior analyses show earlier layers are more perceptually stable [1], [2], and our results follow this pattern.
>
> [1] A. Pasad et al., "Layer-wise analysis of a self-supervised speech representation model", 2021.
>
> [2] A. Pasad et al., "Comparative layer-wise analysis of self-supervised speech models", 2023.
>
> **Revisions**. Sections 6.2 and 7 now state this.

---

> > ### Author Response · Authors · 2025-11-19
> > **Response to Reviewer itJ3 - Doubling down on Q2 and Q4 (Response 2/3)**
> >
> > # Q2 – Disentanglement (expanded)
> >
> > By design, the PM of a source only measures proximity its target reference and distortion cloud, while PS also compares to all non-attributed references.
> > To show that this construction disentangles leakage from self-distortion, we **include two new controlled experiments** where leakage and self-distortion vary independently.
> >
> > **Leakage sweep.** For each mixture z = y₁ + y₂, we construct outputs ŷ₁(α) = y₁ + αy₂ for α ∈ [0,1]. Here only **leakage** increases with α.
> >
> > **Distortion-intensity sweep.** We select a new distortion not present in the MAPSS distortion bank and apply it to y₁ with continuously increasing intensity λ ∈ [0, 1]. Here **only self-distortion** increases with λ.
> >
> > We provide a summary table. As in Table 1, each cell is the aggregations of the **PS/PM**, averaged across all trials in each scenario. Results are given for source#1, over α and λ values:
> >
> > | Scenario | α |      |      |      |      |      |      |      |      |      | λ |      |      |      |      |      |      |      |      |      |
> > |---------|:-----------:|:----:|:----:|:----:|:----:|:----:|:----:|:----:|:----:|:----:|:-----------:|:----:|:----:|:----:|:----:|:----:|:----:|:----:|:----:|:----:|
> > |         | 0 | 0.11 | 0.22 | 0.33 | 0.44 | 0.55 | 0.66 | 0.77 | 0.88 | 1 | 0 | 0.11 | 0.22 | 0.33 | 0.44 | 0.55 | 0.66 | 0.77 | 0.88 | 1 |
> > | **English** | 0.96/1.00 | 0.91/0.91 | 0.83/0.88 | 0.76/0.87 | 0.69/0.85 | 0.64/0.85 | 0.60/0.84 | 0.56/0.84 | 0.53/0.84 | 0.50/0.83 | 1.00/0.99 | 1.00/0.94 | 1.00/0.90 | 1.00/0.85 | 1.00/0.79 | 1.00/0.73 | 1.00/0.66 | 0.99/0.58 | 0.99/0.48 | 0.98/0.36 |
> > | **Spanish** | 0.96/1.00 | 0.88/0.90 | 0.79/0.87 | 0.72/0.85 | 0.65/0.84 | 0.59/0.84 | 0.54/0.83 | 0.50/0.83 | 0.47/0.83 | 0.44/0.83 | 1.00/1.00 | 1.00/0.95 | 1.00/0.90 | 1.00/0.84 | 1.00/0.78 | 1.00/0.71 | 1.00/0.64 | 0.99/0.56 | 0.99/0.46 | 0.98/0.34 |
> > | **Music-ND** | 0.96/1.00 | 0.70/0.83 | 0.61/0.83 | 0.55/0.83 | 0.50/0.83 | 0.45/0.83 | 0.42/0.83 | 0.39/0.83 | 0.36/0.83 | 0.34/0.83 | 1.00/1.00 | 1.00/0.98 | 1.00/0.96 | 1.00/0.94 | 1.00/0.91 | 1.00/0.86 | 1.00/0.79 | 0.99/0.69 | 0.99/0.56 | 0.98/0.40 |
> > | **Music-D** | 0.94/1.00 | 0.85/0.90 | 0.76/0.87 | 0.70/0.85 | 0.65/0.85 | 0.61/0.84 | 0.58/0.84 | 0.55/0.83 | 0.53/0.83 | 0.51/0.83 | 1.00/0.98 | 1.00/0.91 | 1.00/0.86 | 1.00/0.79 | 1.00/0.72 | 1.00/0.63 | 1.00/0.54 | 0.99/0.42 | 0.99/0.30 | 0.98/0.17 |
> >
> >
> > **Observation. PS and PM respond selectively to leakage and self-distortion, respectively. PM shows minor false reaction to leakage.**
> > This supports the claim that the PS and PM **can disentangle** these two perceptual failure modes.
> >
> > **Revisions.** Section 7 ("Results") and the Appendix include this experiment and results.
> >
> > # Q4 – Error analysis (expanded)
> >
> > Below we clarify both **aim and significance** of Section 5 ("Error Guarantees for the PS and PM Measures")
> >
> > The purpose of Section 5 is to answer a single question: **how far can the PS/PM scores we compute differ from their "ideal" definitions**, i.e., the scores one would obtain with (i) the full diffusion-map spectrum and (ii) infinitely many distortion samples per reference. Because both PS and PM rely on a truncated embedding of size $d$ and empirical cluster statistics, their implemented versions inevitably incur approximation error.
> >
> > **Aim.** Section 5 provides two guarantees for every frame and every source:
> > 1. a **deterministic truncation radius** that upper-bounds the effect of discarding the $(N-1-d)$ diffusion coordinates, obtained via the Schur complement;
> > 2. **non-asymptotic confidence bounds** on empirical centroids, covariances, and Mahalanobis distances, derived using vector/matrix Bernstein and Hanson-Wright inequalities, with dependence-adjusted effective sample sizes.
> >
> > Together, these quantify the error guarantees in a mathematically controlled way.
> >
> > **Significance.** These bounds show that our empirical results are **robust to both spectral truncation and finite-sample uncertainty**. When propagated to utterance-level PCC/SRCC (Table 2, Section 7 - "Experimental Results"), one prominent outcome shows that the maximal deterministic bias stays below 1.4% in all scenarios and preserves the superiority of PS/PM versus all 14 baselines in Table 1 across almost all scenarios. Thus, Section 5 directly supports the **soundness** of the claims: the performance of the PS/PM is not an artifact of embedding dimension.
> >
> > **Practical relevance.** These guarantees enable confidence-aware reporting, allow practitioners to assess whether system differences are statistically meaningful, and provide principled guidance on choosing $d$ and the number of distortions. No existing metric for source separation provides analogous uncertainty quantification.
> >
> > **Revisions.** We will improve Section 5 by opening with its aim and significance statements, move tedious algebraic derivations to appendix, and foreground the connection to Tables 1 and 2 so the analysis value is immediately clear.

---

> > > ### Author Response · Authors · 2025-11-19
> > > **Response to Reviewer itJ3 - Weaknesses (Response 3/3)**
> > >
> > > We now address the reviewer's comments on weaknesses that weren't addressed before. We refer to all 14 paragraphs according to their serial numbers.
> > >
> > > We begin by addressing a recurring topic: the presentation. We sincerely apologize that the paper was presented in a manner that incurs difficulty of understanding and a prolonged review process as a result. With that being said, the paper will be revised meaningfully based on your comments, and its presentation will be significantly improved. With this acknowledgement, we will get right to the point in the replies below.
> > >
> > > **W1.** Yes, this line remained due to lack of attention on our part.
> > >
> > > **Revisions.** This line was removed.
> > >
> > > **W2.** We agree the abstract used jargon ("intrusive approach", "system output", "non-attributed reference", "granular"). We state that the evaluation assumes access to clean reference sources, define "system outputs" and "non‑attributed references" explicitly, and replace "granular" with "frame‑level".
> > >
> > > **Revisions.** Abstract and Section 1 ("Introduction") were rewritten to spell out these notions.
> > >
> > > **W3.** We now realize the missing high-level overview and imbalance between sections made the paper hard to follow. We have added a pipeline figure summarizing the computation (inputs → distortions → SSL encoding → diffusion maps → PS/PM) and moved some derivations to the appendix to lighten Sections 2-5 (Until "Experimental Setup").
> > >
> > > **Revisions.** Added Figure 1 (pipeline overview) and relocated lengthy derivations to Appendices.
> > >
> > > **W4.** Section 2 now gives a short high‑level description and states that the formulation is frame‑based, that we later fix a frame and drop f, and that indices are repurposed only then.
> > >
> > > We respectfully rebuttal that providing diffusion maps theoretical foundation in the main body, in our opinion, is justified because it leads to the diffusion distance preservation property that is in the heart of MAPSS (Eq. 9). The embedding by itself is not the sole contribution of the diffusion maps in this work.
> > >
> > > We strongly agree with the reviewer that $$\Psi^{(d)}_t(x)$$ is in fact not a function and should not be notated as such. Also, we adopt the proposed ease in notation, using v instead of $$\Psi^{(d)}_t(x)$$.
> > >
> > > We have addressed the Laplacian eigenmaps concern in previous comments.
> > >
> > > **Revisions.** Sections 3 ("Diffusion maps: theoretical foundations") and onwards will include these changes.
> > >
> > > **W5.** Following W4, we agree notations should follow the reviewer's recommendations.
> > >
> > > Revisions. Section 4 ("The PS and PM measures") will be harmonized with these notations and symbols.
> > >
> > > **W6.** You are absolutely right that calling the SDR/SIR/SAR measures "intrinsically ambiguous" was too strong. We now state instead that they mix leakage and self‑distortion into global energy ratios, a literature-backed claim [1]. Table 1 is presented as evidence of competitive correlations, not as proof of universally better measures, which we assume was the intent of your *mistakenly truncated* comment.
> > >
> > > [1] J. Le Roux et al. "SDR – Half-Baked or Well Done?", 2019.
> > >
> > > **Revisions.** Section 1 ("Introduction") now uses milder wording, adds a brief explanation, and limits our claims to the SEBASS database.
> > >
> > > **W7.** We acknowledge the confusing double use of the word "intrusive" and now reserve "intrusive" for the evaluation protocol that requires clean references and describe distortions instead as "low‑severity" or "high‑severity" perturbations.
> > >
> > > **Revisions.** Both appearances of this term have been revised.
> > >
> > > **W8.** We completely understand the handling of indices was hard to follow. We now stress that f indexes time frames (not frequency), that we fix one frame and drop f in later sections, and that i, j are then used only as generic indices of manifold points. We also added cross‑references where PS and PM are defined.
> > >
> > > **Revisions.** The text now spells out how f, i and j are used and why f disappears later on.
> > >
> > > **W9.** Thank you for catching the acronym issue. We completely agree.
> > >
> > > **Revisions.** Sections 6.1 ("Database") and 7 ("Experimental Results") and the relevant captions now reintroduce MOS and NMI.
> > >
> > > **W10, W11.** Thank you. Addressed in previous comments.
> > >
> > > **W12.** We agree that underlining PS/PM in Table 1 was unnecessary; the table now uses uniform formatting, and the caption alone highlights our rows.
> > >
> > > We have addressed the raters' concern in previous comments.
> > >
> > > **Revisions.** Table 1 formatting was unified.
> > >
> > > **W13.** You are correct that MUSHRA and MOS are different notions. We describe SEBASS as providing MUSHRA (a collection protocol) listening tests whose averaged 0-100 scores we treat as mean-opinion scores (MOS) quality labels, and we refer to them as "MUSHRA‑derived MOS" when reporting correlations.
> > >
> > > **Revisions.** Section 6.1 ("Database") and 7 ("Experimental results") now clearly separate the MUSHRA protocol from the MOS-style averaged scores.
> > >
> > > **W14.** Thank you. Addressed in previous comments.

---

### Official Review · Reviewer_WKee · 2025-11-01

**Soundness:** 3
**Presentation:** 3
**Contribution:** 3
**Rating:** 6
**Confidence:** 2

**Summary:**

This paper introduces two frame-level, intrusive metrics: Perceptual Separation and Perceptual Match, for the assessment of audio source separation systems. The approach decomposes perceptual errors into leakage and self-distortion by encoding distorted signal variants through a pre-trained self-supervised model and embedding the resulting features onto a manifold using diffusion maps. Experimental results on the SEBASS benchmark demonstrate that both metrics offer robust correlation with human MOS compared to 14 existing measures, and the paper derives error and confidence bounds for both metrics.

**Strengths:**

1. The PS and PM measures distinctly address leakage and self-distortion, providing interpretable, complementary diagnostics as established in both the main text and Figure 1. This is a clear improvement over single-score evaluation methods, which frequently conflate distinct error sources.

2. The derivations for both measures are grounded in Mahalanobis distance computation on a perceptual manifold, with clear steps and full error analysis.

3. The paper goes beyond point estimates, deriving deterministic and probabilistic error bounds.

**Weaknesses:**

1. Several directly related and highly contemporary works on perceptual audio metrics, unsupervised perceptual embeddings, and manifold learning for audio are not discussed, despite their clear relevance.

[1] DeePAQ: A Perceptual Audio Quality Metric Based On Foundational Models and Weakly Supervised Learning
[2] NOMAD: Unsupervised Learning of Perceptual Embeddings for Speech Enhancement and Non-matching Reference Audio Quality Assessment

2. The paper states the hypothesis that diffusion distances on the perceptual manifold align with perceived similarity, but aside from empirical validation, this foundational assumption is not theoretically justified.

3. Despite benchmarking against many baselines, the evaluation gives limited direct insight into when/why PS or PM fails relative to credible alternatives (e.g., PESQ, STOI). The qualitative and error-case analysis is not sufficiently granular.

**Questions:**

1. Can the authors empirically or theoretically justify the assumption that diffusion map Euclidean distances on the perceptual manifold directly correspond to perceptual similarity, especially for out-of-domain distortions?

2. How robust are the PS and PM measures to errors in the choice of self-supervised model?

---

> ### Author Response · Authors · 2025-11-23
> **Response to Reviewer WKee - Reply to all Q1 (Response 1/3)**
>
> We would like to thank the reviewer for their constructive questions and comments. The paper has been revised, and we hope you find its contribution significantly strengthened thanks to the reviewer’s feedback.
>
> # Q1 - Perceptual similarity assumption
>
> We appreciate the reviewer for raising this foundational point. We agree that our perceptual-geometric assumption was not framed clearly enough in the original submission, and that readers could reasonably misinterpret our intentions and claims. Our goal is **not** to propose a universal psychoacoustic law, but to articulate a **testable hypothesis**, grounded in prior work on self-supervised representations and diffusion maps, and to support it with targeted analyses. Below we clarify our assumption, the supporting evidence, its behavior under out-of-distribution (OOD) conditions, and its limitations.
>
> **1. Our modeling assumption - not a universal claim.**
> We hypothesize that, for the types of distortions operating in SEBASS, local neighborhoods in a self-supervised encoder space preserve neighborhoods in human perception, following evidence such as [1]. We then use diffusion maps to embed these encoder representations on a manifold where Euclidean distances align with dissimilarities in the encoder space, as formalized in [2] and in Eq. (9) ("diffusion distance preservation"). We explicitly position this construction as a *regularized geometric surrogate* for perceptual similarity, not as a general theory of human auditory similarity judgments.
>
> **2. Empirical justification within our scope.**
> The paper provides several lines of evidence supporting this hypothesis:
>
> (i) **Self-supervised encoder ablation.** Removing the self-supervised representation and applying diffusion maps directly to raw waveforms sharply degrades performance (Table 1, e.g., PS SRCC in English drops from 84.12% to 73.42%, with similar trends elsewhere), suggesting that the encoder supplies structure that better reflects human judgments.
>
> (ii) **Layer analysis.** Earlier layers, known to encode more acoustic and phonetic detail [3], consistently outperform deeper layers (Figures 6-7 in the Appendix), aligning with prior findings that later layers may over-abstract away perceptually relevant cues.
>
> (iii) **Cross-domain validation.** Strong correlations across English, Spanish, and music (Table 1) suggest that the hypothesis holds across multiple content domains.
>
> (iv) **Out-of-distribution distortions.** System outputs in SEBASS often contain artifacts *not* present in our handcrafted distortion bank (Table 3), which focuses on fundamental distortions (additive noise, reverberation, etc.). Despite this, and the fact MAPSS operates in a zero-shot manner, we maintain a leading position among 14 baseline metrics.
>
> **3. Clarifying generalization to unseen distortions.**
> Our pipeline does not assume that SEBASS artifacts lie in the span of the ~60-70 handcrafted distortions in Table 3 (not contradicting the point made in point 1 above). The distortion bank is used solely to sample local neighborhoods around each reference. The manifold is constructed from *all* encoded samples (references, distortions, and system outputs), so unseen distortions are embedded in a zero-shot manner (without training). Perceptually close artifacts fall near the reference cluster, and atypical distortions land farther away.
>
> **4. Acknowledging limitations and adopting more modest claims.**
> We fully acknowledge that:
> (i) We do **not** attempt to develop a complete theory of human perceptual similarity.
> (ii) We validate only on SEBASS, currently the only public dataset with separation-specific MOS.
> (iii) Our perceptual-geometric hypothesis is local and empirical, not universal.
> (iv) Extreme OOD distortions remain outside our present evidence base.
>
> [1] A. Pasad et al., “Layer-wise analysis of a self-supervised speech representation model,” 2021.
>
> [2] R. R. Coifman and S. Lafon, “Diffusion maps,” 2006.
>
> [3] A. Pasad et al., “Comparative layer-wise analysis of self-supervised speech models,” 2023.
>
> **Revisions.** We will revise the paper to make this scope explicit, to clearly acknowledge the limits of our assumptions, and to move key conclusions into the main body. Specifically, we will:
> (i) explicitly label the perceptual-geometric assumption as a *hypothesis*,
> (ii) restructure Sections 4 ("The PS and PM Measures") and 7 ("Results") to foreground the empirical tests that support, but do not prove, the hypothesis,
> (iii) expand the Limitations section to state that we do not claim universal perceptual validity and that future work should validate the assumption on broader, more diverse datasets, with explicit limitations also mentioned in the main body.

---

> > ### Author Response · Authors · 2025-11-23
> > **Response to Reviewer WKee - Reply to all Q2, W1, and W2 (Response 2/3)**
> >
> > # Q2 - Robustness to the choice of self-supervised model
> >
> > We thank the reviewer for highlighting this important robustness question. While we already compared SRCC/PCC performance across self-supervised models (Figure 6, Appendix), the reviewer’s suggestion to analyze *errors* under model changes is particularly valuable. We therefore extended our analysis to quantify the deterministic radii and confidence intervals (CIs) for six SSL models. The table below summarizes the results for English mixtures, with “wav2vec2 large” being the model reported in Table 2 of the paper. Each cell shows "SRCC/PCC%".
> >
> > | **Measure**       | **wav2vec2 large** | **wav2vec2 base** | **hubert large** | **hubert base** | **wavlm large** | **wavlm base** |
> > |-------------------|--------------------|-------------------|------------------|-----------------|-----------------|----------------|
> > | **PS radius**     | 0.16/0.21%         | 0.16/0.16%        | 0.07/0.19%       | 0.20/0.16%      | 0.09/0.20%      | 0/0.10%        |
> > | **PS CIs (95%)**  | 30.03/10.29%       | 28.35/9.91%       | 26.47/9.36%      | 28.34/9.71%     | 26.94/9.41%     | 26.66/9.10%    |
> > | **PM radius**     | 0.11/0.99%         | 0.01/1.10%        | 0.15/1.38%       | 0.07/1.24%      | 0.26/1.24%      | 0.25/1.38%     |
> > | **PM CIs (95%)**  | 7.23/3.83%         | 11.33/5.36%       | 8.44/4.30%       | 11.08/5.16%     | 9.99/4.81%      | 16.13/7.22%    |
> >
> > **Three main observations emerge:**
> >
> > (i) **Deterministic errors are uniformly small across models.**
> > Deterministic radii are small for all models (≤0.2% for PS SRCC, ≤0.21% for PS PCC, ≤0.26% for PM SRCC, and ≤1.38% for PM PCC). This reinforces the paper’s conclusion that deterministic uncertainty around the PS/PM correlations is sufficiently small that competitive rankings are not artifacts of unstable estimates, e.g., in Table 1.
> >
> > (ii) **Uncertainty is comparable across models.**
> > For PS, SRCC CIs range only from 26.47-30.03% and PCC CIs from 9.10-10.29% across models. For PM, SRCC CIs are in a narrow band for most models (7.23-11.33% for five of six models), and PCC CIs are similarly constrained (3.83-5.36% for four of six). No model family (wav2vec2, HuBERT, WavLM) systematically exhibits larger error bars, indicating that all front-ends yield comparably reliable estimates.
> >
> > (iii) **Larger variants tend to be slightly more stable than base variants.**
> > Within each model family, the "large" variants tend to have smaller PM CIs than their "base" counterparts (e.g., PM CIs for wav2vec2 large: 7.23/3.83% vs. wav2vec2 base: 11.33/5.36%, with similar trends hold for HuBERT and WavLM). This suggests that increased model capacity improves stability rather than inflating uncertainty.
> >
> > **Revisions.** Section 7 ("Results") now references this analysis, and the Appendix provides the full details.
> >
> > # W1 - Additional baseline measures
> >
> > We thank the reviewer for directing us to additional contemporary baselines, which are indeed important for a fair comparison. The reviewer mentioned DeePAQ and NOMAD. DeePAQ does not currently provide open-source code, whereas NOMAD has been superseded by **SCOREQ, which we now include**. The table below reports SCOREQ (used in "non-matching reference" mode) in the same setup as Table 1, alongside PS and PM for reference, across all SEBASS scenarios (English, Spanish, music with drums, music without drums). Values are given as SRCC/PCC%.
> >
> > | Metric | English          | Spanish          | Music with drums | Music without drums |
> > |:------|:----------------:|:----------------:|:----------------:|:-------------------:|
> > | PS    | 84.12/83.74%     | 82.33/85.01%     | 72.87/77.38%     | 87.23/87.81%        |
> > | PM    | 84.69/86.36%     | 83.41/85.30%     | 75.18/69.88%     | 88.12/85.26%        |
> > | SCOREQ| 77.02/81.85%             | 82.32/83.13%             | 62.73/68.56%             | 75.35/75.84%                |
> >
> > **Observations.** SCOREQ is trained to estimate utterance-level MOS on telephone and synthetic-speech degradations, not on mixtures containing cross-talk between multiple sources. By contrast, PS and PM explicitly disentangle self-distortion and leakage (as detailed in Q2 for reviewer itJ3) and are purpose-built for evaluating source-separation-the exact perceptual task faced by SEBASS listeners. This task alignment explains their consistent superiority over SCOREQ.
> >
> > **Revisions.** We add SCOREQ into Table 1 and cover it in Section 1 ("Introduction").
> >
> > # W2
> >
> > Thank you. This has been discussed in depth in Q1.

---

> > > ### Author Response · Authors · 2025-11-23
> > > **Response to Reviewer WKee - Reply to W3 (Response 3/3)**
> > >
> > > # W3 – Failing points of PS/PM relative to PESQ
> > >
> > > We thank the reviewer for prompting a clearer analysis of failure cases. Below we focus on speech scenarios and compare PS/PM to PESQ-our strongest comparator (Table 1).
> > >
> > > **1. How PS/PM differ from PESQ.**
> > > PS and PM operate on diffusion-map distances computed from self-supervised embeddings, using a bank of synthetic distortions around each reference (Table 3). They are explicitly designed to (i) disentangle leakage from self-distortion (see Q2's response to reviewer itJ3), and (ii) function at the frame level. PESQ, in contrast, produces a single utterance-level MOS-aligned score using engineered auditory features and an internal time-alignment stage [1].
> > >
> > > **2. When PM fails but PESQ succeeds.**
> > > As shown in our misalignment stress-test (Appendix C, Fig. 8), even moderate delays (>20 ms) between reference and output reduce PM because it compares frame-synchronous embeddings without an alignment stage. PESQ, in contrast, performs time alignment and is largely insensitive to such shifts. Thus, a separator that outputs a clean signal with 30-100 ms latency, common in practice and imperceptible to listeners, receives high MOS and high PESQ, but PM drops because each frame is matched to the wrong reference frame, inflating Mahalanobis distances. Our experiment models this exactly: a pure time-shifted reference with no added leakage or distortion, consistent with real-time communication standards that allow up to 40 ms latency [2].
> > >
> > > **3. When PS fails but PESQ succeeds.**
> > > PESQ is explicitly built for telephony/VoIP and includes mechanisms for packet loss, time alignment, and "bad-frame" handling. PS, by design, targets continuous distortions (Table 3) and does not model deletions, insertions, or non-uniform time warping. In our test, the system output is generated by randomly deleting 20 ms frames from the reference and then time-compressing and resampling to the original length, without any leakage or distortion.
> > >
> > > In the tables below, we report average PM and PESQ values for delays from 0-100 ms, and average PS and PESQ values for packet-loss rates from 0-20%, averaged across all trials in each SEBASS speech scenario (English and Spanish).
> > >
> > > **PM vs. PESQ - misalignments in ms**
> > >
> > > *English trials:*
> > >
> > > | Metric | 0 ms | 10 ms | 20 ms | 30 ms | 40 ms | 50 ms | 60 ms | 70 ms | 80 ms | 90 ms | 100 ms |
> > > |:------|:----:|:-----:|:-----:|:-----:|:-----:|:-----:|:-----:|:-----:|:-----:|:-----:|:------:|
> > > | PM    | 1    | 0.91  | 0.84  | 0.81  | 0.80  | 0.79  | 0.79  | 0.79  | 0.79  | 0.79  | 0.78   |
> > > | PESQ  | 4.64 | 4.49  | 4.43  | 4.40  | 4.41  | 4.36  | 4.35  | 4.33  | 4.33  | 4.32  | 4.29   |
> > >
> > > *Spanish trials:*
> > >
> > > | Metric | 0 ms | 10 ms | 20 ms | 30 ms | 40 ms | 50 ms | 60 ms | 70 ms | 80 ms | 90 ms | 100 ms |
> > > |:------|:----:|:-----:|:-----:|:-----:|:-----:|:-----:|:-----:|:-----:|:-----:|:-----:|:------:|
> > > | PM    | 1    | 0.93  | 0.89  | 0.87  | 0.86  | 0.86  | 0.86  | 0.86  | 0.86  | 0.86  | 0.86   |
> > > | PESQ  | 4.64 | 4.59  | 4.50  | 4.47  | 4.46  | 4.39  | 4.37  | 4.33  | 4.31  | 4.27  | 4.25   |
> > >
> > > **PS vs. PESQ - packet loss in %**
> > >
> > > *English trials:*
> > >
> > > | Metric | 0% | 2.2% | 4.4% | 6.6% | 8.8% | 11.11% | 13.33% | 15.5% | 17.8% | 20% |
> > > |:------|:--:|:----:|:----:|:----:|:----:|:------:|:------:|:-----:|:-----:|:---:|
> > > | PS    | 1  | 0.93 | 0.94 | 0.93 | 0.93 | 0.93   | 0.97   | 0.96  | 0.95  | 0.94 |
> > > | PESQ  | 4.64 | 2.45 | 1.39 | 1.17 | 1.14 | 1.08 | 1.07 | 1.05 | 1.04 | 1.04 |
> > >
> > > *Spanish trials:*
> > >
> > > | Metric | 0% | 2.2% | 4.4% | 6.6% | 8.8% | 11.11% | 13.33% | 15.5% | 17.8% | 20% |
> > > |:------|:--:|:----:|:----:|:----:|:----:|:------:|:------:|:-----:|:-----:|:---:|
> > > | PS    | 1  | 0.93 | 0.94 | 0.93 | 0.93 | 0.93   | 0.97   | 0.96  | 0.95  | 0.94 |
> > > | PESQ  | 4.64 | 2.91 | 1.53 | 1.22 | 1.17 | 1.08 | 1.05 | 1.04 | 1.04 | 1.04 |
> > >
> > > **Observations.**
> > > Under misalignment, PESQ decreases modestly from 4.64 to 4.29 (English) and 4.25 (Spanish) at 100 ms-only ~5% of its dynamic range. PM, bounded in [0, 1], degrades far more: ~22% (English) and ~14% (Spanish), showing substantially higher sensitivity to timing shifts, whereas PESQ remains closer to human judgments. For packet loss, PESQ collapses by more than half its dynamic range even below 5% loss and approaches 1 at 20% loss, aligning with the severe perceptual degradation reported by listeners. PS, by contrast, changes only modestly, indicating that it does not fully capture the perceptual impact of packet loss.
> > >
> > > **Revisions.** Section 7 ("Results") now includes these analyses as an explicit error-analysis subsection, and the Appendix provides additional details.
> > >
> > > [1] A. W. Rix et al., “Perceptual evaluation of speech quality (PESQ) – a new method for speech quality assessment of telephone networks and codecs,” 2001.
> > >
> > > [2] ETSI ES 202 740, “Speech and multimedia Transmission Quality (STQ); Transmission requirements for wideband VoIP loudspeaking and handsfree terminals from a QoS perspective as perceived by the user,” 2016.

---

> > > > ### Comment · Reviewer_WKee · 2025-11-26
> > > >
> > > > After reading the rebuttal and the revised manuscript, I feel that the authors have made a solid effort to address the main concerns I raised. The clarification around the perceptual–geometric assumption is more carefully framed than in the original submission, with a clearer distinction between a local hypothesis supported by empirical evidence versus a universal psychoacoustic claim. The added analyses on robustness across different self-supervised models, as well as the comparison against SCOREQ, help contextualize the proposed metrics within the broader landscape of perceptual assessment methods. I also appreciate the expanded discussion of failure cases relative to PESQ, this directly responds to my request for more granular insight into when PS/PM succeed or fail. That said, while the rebuttal strengthens the paper, some questions remain partially open, especially concerning the limits of generalization under strong OOD distortions and the broader positioning with respect to recent perceptual-audio metrics. I am generally satisfied with the authors’ explanations and revisions and maintain my score at 6.

---

> > > > > ### Author Response · Authors · 2025-11-28
> > > > > **Response to Reviewer WKee - Reply to Official Comment (Response 1/4)**
> > > > >
> > > > > We thank the reviewer for their kind words of encouragement and the appreciation for the work we have done to reply to the reviewer's concerns.
> > > > >
> > > > > We completely understand your view that gaps exist regarding: (i) out of distribution generalization and (ii) the positioning of the MAPSS measures among additional, highly relevant alternatives. For this reason, we have conducted additional experiments dedicated to mitigating those concerns and replied with more depth in 3 responses below.

---

> > > > > > ### Author Response · Authors · 2025-11-28
> > > > > > **Response to Reviewer WKee - Reply to Official Comment (Response 2/4)**
> > > > > >
> > > > > > # 1. Generalization under strong out of distribution distortions
> > > > > >
> > > > > > Since SEBASS is the only corpus with human subjective scores for source separation, we cannot generate additional examples, particularly with strong OOD distortions, and align MAPSS outputs with corresponding human ratings. The sole exception, which we already used in our first response to W3 and in the paper (Figure 7, Appendix), was to introduce input-output temporal misalignment and rely on established perceptual theory that such misalignment does not alter human judgments, allowing reuse of SEBASS ratings. This workaround, however, does not extend to other OOD distortions, limiting our ability to provide a direct empirical assessment of MAPSS generalization under strong OOD conditions.
> > > > > >
> > > > > > However, the reviewer’s comment motivated us to overcome this limitation and still provide meaningful insights into MAPSS’s OOD generalization. We therefore adopt an alternative perspective based on the distortion groups in our distortion bank (Table 3, Appendix A) and ask: How well does MAPSS generalize when one distortion group is entirely left out, and another is augmented with an excessively strong OOD variant? This setup closely mirrors our response to Q4 of h8Us, to which they provided strongly positive feedback.
> > > > > >
> > > > > > We therefore conducted the following experiment. MAPSS uses 13 distortion groups, each with several parameterized variants - about 70 total distortions per reference. For each of the 13 groups, we repeat the same two-step procedure:
> > > > > > (i) remove the entire group from the distortion bank, and
> > > > > > (ii) augment a different remaining group with one new, deliberately excessive parameter whose effect is perceptually far from the reference.
> > > > > >
> > > > > > For example, we omit the "tremolo" group and introduce an additional high-severity noise parameter (e.g., additive noise at -20 dB SNR) into the "noise" group.
> > > > > >
> > > > > > Together with the original results from Table 1 (with an unmodified distortion bank), the extended table below summarizes all 13 trials. Each row specifies which group was omitted ("O.") and which group received the new excessive parameter ("E."). As in the main paper, each entry reports the SRCC/PCC values (in %) obtained by averaging the aggregated PS and PM scores across all trials for that scenario.
> > > > > >
> > > > > > **The PM measure:**
> > > > > >
> > > > > > |Scenario | Original | O. notch, E. comb | O. comb, E. tremolo | O. tremolo, E. noise | O. noise, E. harmonic | O. harmonic, E. reverb | O. reverb, E. noisegate | O. noisegate, E. pitch | O. pitch, E. lowpass | O. lowpass, E. highpass | O. highpass, E. echo | O. echo, E. clipping | O. clipping, E. vibrato | O. vibrato, E. notch |
> > > > > > | --- | --- | --- | --- | --- | --- | --- | --- | --- | --- | --- | --- | --- | --- | --- |
> > > > > > | English | 84.69/86.36% | 82.13/81.62% | 81.75/81.11% | 81.57/80.85% | 82.20/77.83% | 81.72/81.27% | 82.18/81.74% | 82.13/81.64% | 81.66/81.20% | 81.46/80.54% | 81.44/81.02% | 82.19/81.74% | 82.04/81.53% | 81.84/81.43% |
> > > > > > | Spanish | 83.41/85.3% | 79.29/80.16% | 79.09/79.61% | 78.82/78.89% | 80.14/76.66% | 79.08/79.75% | 79.45/80.32% | 79.13/79.69% | 79.33/79.38% | 78.06/78.73% | 79.05/78.45% | 79.47/80.31% | 79.32/80.00% | 79.34/79.86% |
> > > > > > | Music with drums | 75.18/69.88% | 74.57/69.83% | 74.89/69.92% | 74.34/68.34% | 75.06/67.40% | 75.16/70.05% | 75.06/69.88% | 74.97/69.77% | 75.05/70.06% | 75.00/69.79% | 76.22/70.05% | 75.47/69.94% | 75.14/69.85% | 75.29/69.89% |
> > > > > > | Music without drums | 88.12/85.26% | 88.59/85.26% | 88.47/84.98% | 88.27/84.60% | 90.48/80.65% | 88.81/85.05% | 88.50/85.27% | 88.48/85.06% | 88.44/84.70% | 88.82/85.19% | 88.59/85.00% | 88.40/85.26% | 88.47/85.13% | 88.54/85.02% |
> > > > > >
> > > > > > **The PS measure:**
> > > > > >
> > > > > > | Scenario|Original|O. notch, E. comb|O. comb, E. tremolo|O. tremolo, E. noise|O. noise, E. harmonic|O. harmonic, E. reverb| O. reverb, E. noisegate|O. noisegate, E. pitch|O. pitch, E. lowpass|O. lowpass, E. highpass|O. highpass, E. echo|O. echo, E. clipping|O. clipping, E. vibrato|O. vibrato, E. notch|
> > > > > > |---|:---:|:---:|:---:|:---:|:---:|:---:|:---:|:---:|:---:|:---:|:---:|:---:|:---:|:---:|
> > > > > > | English | 84.12/83.74% | 84.09/83.52% | 84.27/83.52% |84.30/83.41%|84.51/83.95%|84.23/83.56%|83.95/83.34%| 84.38/84.43% | 82.64/82.32% | 84.34/83.55% | 84.58/83.49% | 84.20/83.46% | 84.32/83.50%| 84.11/83.48%|
> > > > > > | Spanish | 82.33/85.01% | 82.67/85.08% | 82.45/85.09% | 82.52/84.97% | 83.49/85.08% | 82.60/85.13% | 82.51/84.98% | 82.81/85.74% | 81.22/83.70% | 82.85/85.04% | 82.52/85.21% | 82.33/85.02% | 82.43/85.04% | 82.60/85.05% |
> > > > > > | Music with drums | 72.87/77.38% | 70.65/76.38% | 70.86/76.62% | 68.96/75.68% | 72.12/77.41% | 69.80/76.60% | 70.20/76.43% | 70.54/76.63% | 70.87/76.61% | 70.76/76.56% | 71.39/76.24% | 70.50/76.40% | 70.66/76.50% | 70.16/76.50% |
> > > > > > | Music without drums | 87.23/87.81% | 85.56/86.73% | 86.02/86.94% | 85.81/86.87% | 87.17/87.97% | 85.89/87.05% | 85.68/86.69% | 86.05/86.82% | 85.52/86.98% | 85.48/86.77% | 86.01/87.01% | 85.70/86.68% | 85.53/86.81% | 85.91/86.77% |
> > > > > >
> > > > > > We continue with observations below.

---

> > > > > > > ### Author Response · Authors · 2025-11-28
> > > > > > > **Response to Reviewer WKee - Reply to Official Comment (Response 3/4)**
> > > > > > >
> > > > > > > **Observations.** This experiment directly probes the generalization of MAPSS when the distortion bank itself is made strongly out-of-distribution with respect to the artifacts produced by the separation systems. For each of the 13 distortion groups in the paper, we (i) remove that entire family from the bank, so MAPSS never "sees" this type of distortion when constructing its manifold, and (ii) add an excessively strong parameter to a different family. The SEBASS mixtures, separator outputs, and human scores remain fixed and only the perceptual model used by PS and PM is perturbed. Deviations in SRCC/PCC therefore quantify how much MAPSS relies on a correctly specified distortion bank versus how well it extrapolates to unseen or mis‑calibrated distortions.
> > > > > > >
> > > > > > > Across all 13 omit/extend configurations and four SEBASS scenarios, PS is well-generalized. Relative to the original configuration, PS SRCC/PCC vary by at most 1.5/1.4 points for English and Spanish speech, 3.9/1.7 points for music with drums, and 2/1 points for music without drums. In all cases, PS keeps the same qualitative ranking against competing metrics as in Table 1 of the paper: it remains among the best-performing measures in terms of correlation with human mean opinion score. This indicates that the separation score generalizes well even when the manifold is built from a substantially misspecified distortion bank and when one distortion family is pushed far beyond the severity range used in the original setup.
> > > > > > >
> > > > > > > PM shows a clearer limit to OOD generalization, which is expected because it explicitly models the empirical distribution of self‑distortions within each cluster. When we remove a perceptually dominant family for speech, most notably additive noise, and simultaneously inject a very severe parameter into another family, the largest degradation is about 3-5 SRCC points and 8-9 PCC points for English and Spanish. For other omit/extend pairs the effect is smaller (typically 1-4 points), and for music PM is essentially stable: with drums, SRCC changes by less than 1 point and PCC by 2.5 points, and without drums, SRCC never decreases relative to the original configuration and only one condition reduces PCC by 4.6 points. These results suggest that PM extrapolates gracefully to much stronger distortions within the families represented in the bank, but its performance degrades when the bank completely omits a distortion family that strongly influences human ratings in that domain (e.g., noise in speech). We argue that the main practical limit we observe is coverage of perceptually salient distortion types, not sensitivity to the exact parameter ranges.
> > > > > > >
> > > > > > > **Summary.** The omit/extend experiment aims to give the most direct response to the reviewer’s question about the limits of generalization of MAPSS under strong out-of-distribution distortions. Even under deliberately adversarial misspecification of the distortion bank, PS correlations with human ratings remain within a few percentage points of the original configuration and preserve their relative ranking among existing metrics, indicating strong OOD robustness of the separation measure. PM exhibits moderate, domain‑dependent sensitivity: when an entire, perceptually crucial distortion family for speech is removed, its average correlations can drop by almost 10 absolute points, whereas in all other domains and configurations the impact is considerably smaller. Overall, we suggest that these results show that MAPSS is not brittle with respect to the precise choice of distortion parameters. Its generalization under strong OOD distortions is primarily limited by whether the distortion bank spans the main types of degradations present in the target application. As long as this coarse coverage is ensured, both PS and PM continue to correlate with human scores well, even when individual distortion families are pushed into much stronger levels than those used in the original distortion bank.
> > > > > > >
> > > > > > > **Revisions.** Section 7 ("Results") now includes these results and discussion, and the Appendix extend the discussion on the generalization of MAPSS. The Limitation section is also extended with the generalization limitations to OOD distortions.

---

> > > > > > > > ### Author Response · Authors · 2025-11-28
> > > > > > > > **Response to Reviewer WKee - Reply to Official Comment (Response 4/4)**
> > > > > > > >
> > > > > > > > # 2. The positioning of MAPSS among competing measures
> > > > > > > >
> > > > > > > > We completely agree that we should include even more alternative measures, especially those that are directly related to perceptual audio metrics, unsupervised perceptual embeddings, and manifold learning for audio. To address the reviewer's rightful concern, we now extend Table 1 that already compares the performance of MAPSS with 15 others measures (including **SCOREQ [NeurIPS, 2024]**, asked in original response by the reviewer) to have **18 alternative measures, by adding 3 more** highly relevant, open-source approaches: (i) **VQScore [ICLR, 2024]** [1], (ii) **NORESQA [NeurIPS, 2021]** [2], and (iii) **Visqol V3 [QoMEX, 2020]** [3].
> > > > > > > >
> > > > > > > > The table below reports the MAPSS measures, SCOREQ (used in "non-matching reference" mode), NORESQA, VQScore, and Visqol (version 3) across all SEBASS scenarios (English, Spanish, music with drums, music without drums). Values are given as SRCC/PCC%.
> > > > > > > >
> > > > > > > > | **Metric** | **English**       | **Spanish**       | **Music with drums** | **Music without drums** |
> > > > > > > > |-----------|--------------------|--------------------|------------------------|---------------------------|
> > > > > > > > | PS        | 84.12/83.74%       | 82.33/85.01%       | 72.87/77.38%           | 87.23/87.81%              |
> > > > > > > > | PM        | 84.69/86.36%       | 83.41/85.30%       | 75.18/69.88%           | 88.12/85.26%              |
> > > > > > > > | SCOREQ    | 77.02/81.85%       | 82.32/83.13%       | 62.73/68.56%           | 75.35/75.84%              |
> > > > > > > > | NORESQA | 55.82/41.53%       | 58.56/47.24%       |-2.34/-0.80%          |  61.21/65.45%              |
> > > > > > > > | VQScore | 36.08/37.31%       | 36.93/40.65%       | 9.75/-7.68%           | 31.14/33.83%              |
> > > > > > > > | Visqol (V3) | 72.59/74.55%       | 76.48/75.48%       | 71.94/65.48%           | 78.75/70.31%              |
> > > > > > > >
> > > > > > > > **Observations.** The MAPSS measures, PM and PS, outperform all the above alternative measures across all scenarios, both in terms of PCC and SRCC.
> > > > > > > >
> > > > > > > > VQScore is a non‑intrusive metric built from a variational auto-encoder trained only on clean speech, and quality is inferred from how well a signal can be encoded using a speech codebook, without references or human labels. This unsupervised perceptual embedding measures "speech‑likeness", not separation quality. ViSQOL is a full-reference perceptual audio metric: it compares a degraded signal to a single clean reference using gammatone "neurograms". It assumes coding/network degradations around one reference stream, not multi‑source separation where interference between stems is the primary failure mode. NORESQA learns a relative quality function between two non‑matching recordings, using STFT features fed to an Inception‑style CNN + TCN, trained purely on synthetic degradations of clean speech and signal‑based labels, e.g. SNR and SI‑SDR, without human scores. This framework gives a powerful, but fundamentally speech‑centric notion of degradation tied to SNR/SI‑SDR, not to perceptual leakage between multiple sources.
> > > > > > > >
> > > > > > > > **Revisions.** We add NORESQA, VQScore, and Visqol into Table 1 in the paper and cover them in Section 1 ("Introduction").
> > > > > > > >
> > > > > > > > [1] Fu et al. "Self-supervised Speech Quality Estimation and Enhancement Using Only Clean Speech", ICLR, 2024.
> > > > > > > >
> > > > > > > > [2] Manocha et al. "NORESQA: A framework for speech quality assessment using non-matching references", NeurIPS, 2021.
> > > > > > > >
> > > > > > > > [3] Chinen et al. "ViSQOL v3: An open source production ready objective speech and audio metric", QoMEX, 2020.

---

### Author Response · Authors · 2025-11-29
**Summarizing paper, reviews, and discussion for the Area Chair (Part 1/5)**

We would like to thank the program chairs and area chairs for the enormous amount of work you are doing under very difficult circumstances. In light of the recent policy change, we are writing this comment specifically for the AC to clearly summarize (i) what our paper contributes, (ii) how the reviews evolved during the discussion, and (iii) why, based on the evidence now in the paper and the discussion thread, we believe this work merits acceptance to ICLR 2026.

---

## 1. Scientific contribution and relevance to ICLR

**Problem and gap.** Objective evaluation of source separation systems is still strongly misaligned with human perception, especially when two distinct failure modes interact:

- *Leakage*: "How much of other sources still bleeds into my target?"
- *Self-distortion*: "How much did you damage the target itself?"

These two failure modes are exactly what practitioners care about when building and debugging real audio separation systems, and in a broader sense, spoken dialogue and conversational systems.
Existing metrics (SDR/SIR/SAR, PESQ/STOI, DNSMOS, etc.) are not specifically designed to disentangle these failure modes, and as we show by comparing our measures to **18 alternatives** (post-review), in the task of audio source separation they often correlate poorly with human perception of audio quality.

**What we propose.** Our work introduces **MAPSS**, a zero-shot, intrusive, frame-level evaluation framework with two complementary measures:

- **Perceptual Match (PM)** – quantifies *self-distortion*: does the separated output of a separation system "stay within" the perceptual distortion cloud of its own reference source?
- **Perceptual Separation (PS)** – quantifies *leakage*: is the separated output closer (on a learned perceptual manifold) to its own source than to *any* other source?

Both measures are between 0 and 1 where higher is better. Technically, MAPSS combines:

1. A bank of ~70 *fundamental distortions* per reference (noise, reverb, filtering, pitch/time perturbations, dynamic effects, etc.), detailed in Table 3 in the Appendix,
2. A **self-supervised encoder** (wav2vec2/HuBERT/WavLM/Mert) to obtain perceptual embeddings,
3. A **diffusion-maps manifold** to regularize geometry so that Euclidean distances approximate diffusion distances in the encoder space,
4. **Mahalanobis distances** in this manifold to define PS/PM framewise,
5. **Deterministic error radius and non-asymptotic confidence intervals** for the scores at the frame and utterance level.

Both PS and PM are:

- **Frame-level (up to 75 frames per second)**: enabling granular diagnosis and potential use as training losses,
- **Differentiable**: allowing integration into learning objectives,
- **Zero-shot**: neither training on human ratings, nor tuning hyperparameters from data is involved.

We see this as squarely in the scope of ICLR: it combines *self-supervised representation learning*, *manifold learning*, and *uncertainty quantification* to solve a central problem in evaluating modern audio models.

---

> ### Author Response · Authors · 2025-11-29
> **Summarizing paper, reviews, and discussion for the Area Chair (Part 2/5)**
>
> ## 2. Empirical strength: correlation with human mean opinion scores (MOS) and breadth of comparison
>
> All our evaluations are on **SEBASS**, the only publicly available separation dataset with human ratings, with over 11k human scores, across scenarios of English, Spanish and music (with and without drums) mixtures, taking into account 32 different separation systems.
>
> ### 2.1. Correlation with human ratings
>
> Across all four SEBASS scenarios, **PS and PM are almost always first or second** in both SRCC and PCC (Spearman and Pearson correlation coefficients) with average human scores, against a large pool of 18 baselines (extended from 14 after discussion), including:
>
> - Classical: SDR/SIR/SAR, PESQ, STOI, etc.,
> - Modern learning-based metrics: DNSMOS, SCOREQ, VQScore, NORESQA,
> - Perceptual signal metrics: ViSQOL, and others.
>
> As the primary result of the paper, for self-containment, this table is attached below (identical to Table 1 in the paper plus the extended metrics: SCOREQ, NORESQA, VQScore, and Visqol):
>
> | **Metric**        | **English**         | **Spanish**         | **Music with drums**   | **Music without drums** |
> |-------------------|---------------------|---------------------|------------------------|-------------------------|
> | PS                | 84.12/83.74%        | 82.33/85.01%        | 72.87/77.38%           | 87.23/87.81%            |
> | PM                | 84.69/86.36%        | 83.41/85.30%        | 75.18/69.88%           | 88.12/85.26%            |
> | PS (waveform)     | 73.42/71.04%        | 74.69/75.05%        | 51.75/61.83%           | 78.88/78.95%            |
> | PM (waveform)     | 69.30/66.62%        | 68.27/67.35%        | 49.52/51.77%           | 74.37/75.51%            |
> | STOI              | 80.85/78.40%        | 78.79/82.56%        | 67.29/71.27%           | 75.64/78.13%            |
> | eSTOI             | 82.14/82.28%        | 79.20/82.68%        | 54.68/57.35%           | 70.06/74.45%            |
> | PESQ              | 85.56/84.05%        | 86.06/84.98%        | 61.60/53.87%           | 61.26/60.24%            |
> | SI-SDR            | 78.11/76.96%        | 84.07/81.38%        | 42.08/56.98%           | 70.42/71.96%            |
> | SDR               | 77.72/73.13%        | 84.29/76.07%        | 44.78/54.33%           | 74.51/75.35%            |
> | SIR               | 51.28/56.20%        | 45.67/55.19%        | 18.64/35.76%           | 51.00/55.12%            |
> | SAR               | 75.54/72.98%        | 78.21/73.29%        | 36.98/40.81%           | 66.15/68.96%            |
> | CI-SDR            | 78.66/77.41%        | 84.32/81.48%        | 45.02/55.42%           | 74.25/75.11%            |
> | DNSMOS-OVRL       | 63.70/67.77%        | 35.34/43.57%        | 21.79/34.27%           | 13.81/19.47%            |
> | MCD               | 43.05/33.86%        | 45.90/37.97%        | 30.27/42.23%           | 33.49/32.19%            |
> | SpeechBERTscore   | 68.58/67.44%        | 69.55/70.48%        | 52.33/59.71%           | 75.60/81.13%            |
> | Sheet-SSQA        | 41.17/51.38%        | 61.06/73.01%        | 39.40/29.03%           | 14.19/5.17%             |
> | UTMOS             | 55.53/55.43%        | 52.22/55.75%        | -9.24/-8.25%           | 12.59/7.72%             |
> | NISQA             | 60.78/67.62%        | 63.37/66.58%        | 27.27/41.73%           | 42.33/48.07%            |
> | SCOREQ            | 77.02/81.85%        | 82.32/83.13%        | 62.73/68.56%           | 75.35/75.84%            |
> | NORESQA           | 55.82/41.53%        | 58.56/47.24%        | -2.34/-0.80%           | 61.21/65.45%            |
> | VQScore           | 36.08/37.31%        | 36.93/40.65%        | 9.75/-7.68%            | 31.14/33.83%            |
> | Visqol (V3)       | 72.59/74.55%        | 76.48/75.48%        | 71.94/65.48%           | 78.75/70.31%            |
>
>
> Crucially, **PM/PS achieve their leading performance without any supervision from MOS labels**, whereas several competing metrics are trained (directly or indirectly) on subjective ratings or proxy quality labels.

---

> ### Author Response · Authors · 2025-11-29
> **Summarizing paper, reviews, and discussion for the Area Chair (Part 3/5)**
>
> ### 2.2. Disentangling leakage vs self-distortion (vs SDR/SIR/SAR)
>
> A central scientific claim of the paper is that MAPSS *functionally disentangles* leakage and self-distortion in a way that energy-ratio metrics (SDR-like) cannot.
>
> In response to reviewers, we added **explicit controlled experiments**:
>
> - **Leakage sweeps:** For each mixture $\(z = y_1 + y_2\)$, we construct outputs $\(\hat y_1(\alpha) = y_1 + \alpha y_2\)$ with increasing interference (α ∈ [0,1]) and *no self-distortion*.
>   - **Result:** PS decreases steadily with α, while PM remains essentially flat (minor reaction only), showing that PS is selectively sensitive to leakage.
>
> - **Self-distortion sweeps:** We repeatedly distort $\(y_1\)$ with a *new* distortion not in the MAPSS bank, varying its intensity λ ∈ [0,1], with **no leakage**.
>   - **Result:** PM decreases strongly with λ, while PS remains high, demonstrating that PM selectively captures self-distortion.
>
> These experiments, summarized numerically in the discussion, show that (i) PS and PM react independently to the leakage and self-distortion, and (ii) they present noticeable advantage over SDR-like counterparts - two concerns posed by a reviewer. Empirically, on SEBASS (English), SIR attains PCC 56%, while PS achieves 84% PCC, and similarly SAR/SDR are 73% PCC vs 86% for PM, with similar behavior elsewhere.
>
> ### 2.3. Robustness across SSL encoders in English
>
> We systematically evaluated **six self-supervised front-ends**: wav2vec2 base/large, HuBERT base/large, WavLM base/large. For each, we re-compute:
>
> - PS/PM correlations (SRCC/PCC),
> - Deterministic truncation radii,
> - 95% confidence intervals on correlations.
>
> Key observations (summarized in our reply to Reviewer WKee):
>
> - Deterministic radii are *uniformly tiny* across models (less than 0.2% for PS SRCC, 0.21% for PS PCC, 0.26% for PM SRCC, and 1.38% for PM PCC).
> - 95% CIs on correlations vary only by a few percentage points across models so no model family is systematically unstable.
> - Larger variants tend to have *slightly tighter* PM confidence intervals than base counterparts, suggesting increased capacity improves stability.
>
> This shows that the **competitive ranking of PS/PM is not an artifact of a particular encoder choice**, and that the uncertainty analysis in Section 5 in the paper (the error guarantees) is meaningful for all tested front-ends.
>
> ### 2.4. Robustness to distortion-bank design and out of distribution (OOD) distortions
>
> A legitimate concern (raised by both WKee and h8Us) is whether PM/PS overfit to our particular distortion bank.
>
> We therefore ran a substantial **omit/extend experiment**:
>
> - MAPSS uses **13 distortion groups** (70 distortions per reference, detailed in Table 3 in the Appendix).
> - For *each* group in turn, we:
>   1. **Remove** that entire group from the bank, and
>   2. **Add a deliberately excessive parameter** (strong OOD variant) to a *different* group (e.g., high-severity -20 dB SNR noise).
>
> This creates 13 "misspecified" distortion banks, each with a different omitted and exaggerated family, while the SEBASS audio and MOS scores are *fixed* since they are not affected by this choice.
>
> Results (summarized in our replies and now integrated in Section 7 and the appendix):
>
> - **PS** is extremely stable. Across all 13 configurations and four SEBASS scenarios, PS SRCC/PCC changes by at most a few absolute points (e.g., less than 1.5/1.4 points for English/Spanish speech, 3.9/1.7 points for music with drums, 2/1 points for music without drums), and **its qualitative ranking vs all baselines is preserved**.
>   - This is consistent with PS relying mainly on *between-cluster contrast* and that Mahalanobis distances de-emphasizes directions inflated by extreme distortions.
>
> - **PM** shows **moderate, interpretable sensitivity**, primarily when an entire *perceptually dominant* speech distortion family (e.g., noise) is removed and another is strongly exaggerated. In the worst case, average PM correlations drop by around 3-5 SRCC and 8-9 PCC points for speech. For music, PM is essentially stable.
>   - This suggests that PM is not brittle to parameter ranges but does require that the distortion bank *span* the main types of distortions that matter for the target domain.
>
> We believe this is precisely the kind of robustness evidence that was missing in the original submission and now substantially strengthens the case for MAPSS as a practical tool.

---

> ### Author Response · Authors · 2025-11-29
> **Summarizing paper, reviews, and discussion for the Area Chair (Part 4/5)**
>
> ### 2.5. Multilingual mixtures
>
> Reviewer h8Us correctly pointed out that real conversations are often multilingual or code-switched, and that monolingual SEBASS splits might make the task easier.
>
> We therefore evaluated MAPSS on the **original multilingual SEBASS mixtures**, with two English and two Spanish speakers per mixture (female-only and male-only subsets). We reported:
>
> - PS and PM for multilingual mixtures vs the monolingual English-only and Spanish-only setups.
>
> Result (from our response, now in Section 7):
>
> - **PS** slightly *improves* in multilingual mixtures (e.g., 86.27/85.6% vs 84.12/83.74% for English-only),
> - **PM** remains comparable.
>
> This indicates that MAPSS does *not* rely on language differences heavily. In fact, estimating PS on richer multilingual manifolds appears slightly easier.
>
> ### 2.6. Error analysis: guarantees and practical meaning
>
> Section 5 provides:
>
> - A **deterministic truncation radius** for the effect of discarding higher diffusion dimensions, via a Schur-complement-based analysis,
> - **Non-asymptotic high-probability confidence bounds** (Bernstein/Hanson-Wright-style) on empirical centroids, covariances, and Mahalanobis distances, propagated to utterance-level SRCC/PCC.
>
> We explicitly show that:
>
> - The **maximal deterministic bias** on utterance-level correlations is below 1.4% in all conditions, and
> - This bias does *not* change the ranking of PS/PM compared to the 18 baselines.
>
> Practically, this means that **differences in PS/PM between systems are statistically interpretable**, and that we can report MOS correlations with principled uncertainty estimates - something **no existing audio quality metric, to our knowledge, provides.**
>
> ---
>
> ## 3. How the reviews evolved and how concerns were addressed
>
> - **Reviewer h8Us (original score: 6)**:
>   - Initially highlighted important questions about novelty vs SIR/SAR, sensitivity to distortion choice, multilingual behavior, PS pooling, and external validity.
>   - After our rebuttal and new experiments (disentanglement, multilingual evaluation, distortion-bank sensitivity, full pooling rule, explicit limitations), they wrote:
>     > "The Authors have fully addressed my concerns and significantly strengthens the paper. I will be raising my score to 6."
>   - In other words, they explicitly state that their concerns were fully resolved and that the paper is *strengthened*. Although policy change renders this irrelevant, we did notice that the **original and the potentially raised score are identical** and posted back to bring attention to this matter.
>
> - **Reviewer WKee (original score: 6)**:
>   - Raised concerns about (i) the perceptual-geometric assumption behind the manifold, (ii) robustness to the choice of SSL model, (iii) comparison to additional perceptual baselines, and (iv) insufficient error-case analysis vs PESQ.
>   - We added:
>     - A clearer framing of our assumption as a *local empirical hypothesis*, not a universal psychoacoustic law,
>     - A multi-encoder robustness study (six SSL models),
>     - Additional baselines (SCOREQ),
>     - Explicit misalignment and packet-loss stress tests contrasting PS/PM with PESQ.
>   - Their follow-up comment states:
>     > “I feel that the authors have made a solid effort to address the main concerns I raised… That said... some questions remain partially open... I am generally satisfied with the authors’ explanations and revisions and maintain my score at 6.”
>    - To respond to the reviewer's remaining concerns, **we provided a second response and doubled-down with additional experiments**, including even more alternative measures (NORESQA, VQScore, ViSQOL) and deeply analyzing the OOD generalization of MAPSS.

---

> ### Author Response · Authors · 2025-11-29
> **Summarizing paper, reviews, and discussion for the Area Chair (Part 5/5)**
>
> - **Reviewer itJ3 (original score: 2)**:
>   - Acknowledged that the **contribution is "good" (3)** but scored **"presentation: 1"** and **"soundness: 2"**, mostly because the initial version was hard to follow and some components (error analysis, listening tests, parameter tuning) were not clearly motivated or described.
>   - We responded point-by-point, including:
>     - Clarifying that **no new listening tests** were conducted and that we only use human scores from the SEBASS database, and have not included any additional human subjects (addressing ethics flag),
>     - Explaining in detail the *aim and significance* of the error analysis and connecting it directly to Table 2 in the paper that shows the uncertainty on MOS correlations in Table 1 in the paper,
>     - Clarifying all parameter choices and explicitly stating that MAPSS is evaluated zero-shot and **no data-driven parameter choice was carried**,
>     - Providing a **disentanglement experiment** to substantiate that PS/PM truly separate leakage and self-distortion,
>     - Cleaning up notation (including the diffusion-maps notation issue they pointed out), adding a **pipeline overview figure**, and reorganizing sections for clarity,
>     - Adding precise definitions for the waveform-only variant, intrusiveness, acronym clarifications, and correcting over-strong claims about SDR/SIR/SAR.
>
> We fully agree that the initial presentation was not as clear as it should have been. Their feedback led to **substantial writing and rebalancing of the paper**. While the reviewer did not post another follow-up after our extended responses, we believe their *substantive* concerns have been comprehensively addressed in the current version.
>
> In short:
>
> - Two reviewers with scores at/above the acceptance threshold explicitly state in discussion that their concerns have been fully addressed and that the paper has been significantly strengthened.
> - The third reviewer’s low score is primarily driven by initial presentation and missing clarifications that have now been addressed via substantial revisions and new experiments.
>
> We respectfully ask the AC to take this full discussion into account when interpreting the pre-discussion scores.
>
> ---
>
> ## 4. Reproducibility and value to the community
>
> We are committed to making MAPSS *immediately useful* to the community:
>
> - The submission already includes a **code + examples zip**.
> - We will additionally provide an **anonymous public repository** that:
>   - Directly interfaces with the public SEBASS release,
>   - Contains a **`reproduce_results.py`** script to regenerate *all* tables/figures in the paper (including extended experiments from the rebuttal),
>   - Exposes a simple API to compute PS and PM on arbitrary mixtures and separation outputs, including a hugging-face demo page
>
> Because PM/PS are:
>
> - **Differentiable and frame-level**,
> - **Zero-shot**,
> - **Empirically well-correlated with human ratings** across languages and music,
>
> we expect them to be directly adopted as:
>
> - Training losses or auxiliary objectives for separation models,
> - Diagnostic tools for analyzing failure modes,
> - Evaluation metrics for future separation benchmarks. This is crucial, since as Tables 1, 2 suggest - adopting the PS and PM by the community means **more reliable, error-aware evaluation compass of audio source separation.**
>
> Given the current explosion of conversational and spoken dialogue systems in audio, we believe this is a **timely and broadly useful contribution** for the ICLR community.
>
> ---
>
> ## 5. To the Area Chair
>
> Putting everything together:
>
> - The **core idea**: perceptual, manifold-based separation of leakage vs self-distortion, with uncertainty quantification, is novel and strongly aligned with ICLR themes in self-supervised representation learning and manifold methods.
> - The **empirical evidence** is extensive: four domains (English, Spanish, music with/without drums), 32 systems, 18 strong baselines, multi-encoder robustness, multilingual mixtures, distortion-bank sensitivity, controlled disentanglement experiments, and explicit failure-case analysis.
> - The **uncertainty analysis** is mathematically grounded and practically interpretable and shows that performance claims and rankings are *robust* to deterministic error radius.
> - The **review discussion** shows that two reviewers are satisfied and explicitly report that their main concerns have been fully resolved. The third reviewer’s reservations focused mostly on presentation and missing clarifications that have now been addressed via substantial revisions and new experiments.
> - We provide **code and will provide a reproducible pipeline**, making it easy for the community to adopt, scrutinize, and build upon MAPSS.
>
> For these reasons, we respectfully and humbly believe that our paper now meets and, in several ways, exceeds the bar for acceptance at ICLR 2026.
>
> Thank you for your consideration, time and effort invested in reading our message.

---

### Meta-Review · Area_Chair_KbTB · 2026-01-07

**Summary:**

This paper proposes MAPSS, a perceptually grounded evaluation framework for audio source separation, introducing two complementary metrics—Perceptual Separation (PS) and Perceptual Match (PM)—that explicitly disentangle leakage and self-distortion, two failure modes that are conflated by traditional energy-ratio metrics (e.g., SDR/SIR/SAR). The approach combines self-supervised audio representations, diffusion-map manifold learning, and principled uncertainty quantification to produce frame-level, differentiable, and zero-shot quality measures.

Initial reviewer concerns focused on several aspects, including the novelty of PS/PM relative to SIR/SAR, sensitivity to the choice of distortion bank, multilingual behavior, the specification of PS temporal pooling, external validity beyond curated datasets, the perceptual-geometric assumption underlying the manifold construction, robustness to the choice of self-supervised encoder, comparison against additional perceptual baselines, insufficient error-case analysis relative to PESQ, as well as substantial issues related to clarity and presentation.

The rebuttal substantially strengthened the submission by expanding both empirical evidence and explanatory clarity. The authors added controlled disentanglement experiments, robustness and out-of-distribution analyses, multilingual evaluations, extended baseline comparisons, explicit failure-case analyses, and major revisions to the paper’s presentation and organization. Collectively, these additions directly informed and support the final recommendation.

**Reviewer Concerns:**

***Concerns addressed by the rebuttal:***

Novelty relative to SDR/SIR/SAR:
The authors provided detailed clarifications.

Disentanglement of leakage and self-distortion:
New controlled experiments independently varying leakage and distortion demonstrated selective sensitivity of PS and PM, addressing a key concern raised by reviewers.

Robustness to distortion bank design and OOD conditions:
Extensive omit/extend  experiments over 13 distortion groups showed strong robustness for PS and interpretable, bounded sensitivity for PM.

Robustness to SSL encoder choice:
Additional analyses across six self-supervised encoders showed stable correlations and small deterministic error radii, indicating results are not encoder-specific.

Multilingual and cross-domain evaluation:
New experiments on multilingual SEBASS mixtures demonstrated that PS/PM remain stable, addressing concerns about reliance on monolingual separation.

Error analysis:
The rebuttal clarified the motivation and significance of the error analysis.

Presentation, clarity, and reproducibility:
The authors substantially revised the paper by adding a pipeline overview figure, clarifying terminology, clarifying no new human studies were conducted, and improving notation and structure.

Ethics concerns:
Clarified that no new listening tests or human-subject experiments were performed; all ratings come from the licensed SEBASS dataset. ACs agreed no ethics review was needed.

***Remaining or partially outstanding concerns:***

External validity beyond SEBASS:
While robustness and OOD analyses were significantly expanded, validation remains limited to SEBASS-style curated mixtures due to the lack of publicly available MOS-annotated separation datasets. This limitation is now clearly acknowledged in the paper.

One reference was flagged as potentially hallucinated. The authors should carefully verify and correct this citation. Based on the AC’s check, the publication year appears to be incorrect.

Andrey N. Kolmogorov. On the empirical determination of a distribution law. In Albert N. Shiryaev (ed.), Selected Works of A. N. Kolmogorov, Volume II, pp. 139-146. Springer, New York, 1986.

**Reviewer Scores:**

Reviewer WKee (initial score: 6):
Focused on the perceptual–geometric assumptions underlying the method, robustness to different self-supervised encoders, the inclusion of additional perceptual baselines, and the need for more explicit failure-case analysis. The rebuttal added substantial theoretical framing, robustness studies, and detailed error analyses. The reviewer expressed general satisfaction and maintained the score. The authors further addressed remaining concerns by providing additional analyses on generalization under strong out-of-distribution distortions and by expanding comparisons to more recent alternative measures, which reasonably resolved the outstanding issues.

Reviewer h8Us (initial score: 6):
Raised concerns regarding novelty relative to SIR/SAR, multilingual behavior, temporal pooling, and sensitivity to distortion choice. Following extensive new experiments and clarifications, the reviewer explicitly stated that all concerns were addressed and that the paper was significantly strengthened.

Reviewer itJ3 (initial score: 2):
Raised extensive concerns primarily related to presentation quality, clarity of exposition, motivation and significance of the error analysis, and potential ethics confusion. The authors responded point by point with substantial revisions, additional experiments, and detailed clarifications. Although the reviewer did not provide a post-discussion score update, the core issues underlying the low rating—clarity, justification, and transparency—were comprehensively addressed in the revised submission. The post-discussion assessment would plausibly increase substantially, though the exact revision is unknown.

Reviewers with scores at or above the acceptance threshold explicitly confirmed that their concerns were resolved. The sole low score was driven largely by presentation and clarification issues that have since been addressed through substantial revision.

***Final Recommendation:***

This paper presents a novel, well-motivated, and empirically strong contribution to the evaluation of audio source separation, combining self-supervised representation learning, manifold methods, and principled uncertainty quantification. The rebuttal substantially strengthened the submission, resolving major concerns about novelty, robustness, and clarity. While validation is necessarily limited by the availability of human-rated datasets, the authors clearly acknowledge this and provide extensive results and analysis within the current constraints. The proposed PS and PM measures are likely to be valuable to the audio or multimodal community as evaluation metrics and potentially training objectives, making the paper a solid fit for ICLR 2026.

---

> ### Public Comment · ~Amir_Ivry1 · 2026-02-27
> **Response to Area Chair**
>
> Dear Area Chair,
>
> Thank you for your constructive comments and the time and attention you have given to our manuscript. The unusual circumstances have surely made your efforts compound, but it is evident you have considered the reviewers' comments carefully and made your own judgement.
>
> The final camera-ready paper has been uploaded and has taken into account all new experiments, follow-up results, and additional comments made by yourself and the reviewers.
>
> Thank you,
> MAPSS Authors

---

### Decision · Program_Chairs · 2026-01-26

Accept (Poster)